**Journal of Clinical and Translational Research 2021; 7(5): 666-681**

Journal homepage: http://www.jctres.com/en/home

# Evidence for a connection between coronavirus disease-19 and exposure to radiofrequency radiation from wireless communications including 5G

Beverly Rubik[1,2]*, Robert R. Brown[3]

[1]*Department of Mind-Body Medicine, College of Integrative Medicine and Health Sciences, Saybrook University, Pasadena CA, USA, [2]Institute for Frontier Science, Oakland, CA, USA, [3]Department of Radiology, Hamot Hospital, University of Pittsburgh Medical Center, Erie, PA; Radiology Partners, Phoenix, AZ, USA*

## ARTICLE INFO

*Article history:*

*Keywords:*
COVID-19
coronavirus
coronavirus disease-19
severe acute respiratory syndrome
coronavirus 2
electromagnetic stress
electromagnetic fields
environmental factor
microwave
millimeter wave
pandemic
public health
radio frequency
radiofrequency
wireless

*\*Corresponding author:*
Beverly Rubik
College of Integrative Medicine and Health
Sciences, Saybrook University, Pasadena CA;
Institute for Frontier Science, Oakland, CA,
USA. E-mail: brubik@earthlink.net

## ABSTRACT

**Background and Aim:** Coronavirus disease (COVID-19) public health policy has focused on the severe acute respiratory syndrome coronavirus 2 (SARS-CoV-2) virus and its effects on human health while environmental factors have been largely ignored. In considering the epidemiological triad (agent-host-environment) applicable to all disease, we investigated a possible environmental factor in the COVID-19 pandemic: ambient radiofrequency radiation from wireless communication systems including microwaves and millimeter waves. SARS-CoV-2, the virus that caused the COVID-19 pandemic, surfaced in Wuhan, China shortly after the implementation of city-wide (fifth generation [5G] of wireless communications radiation [WCR]), and rapidly spread globally, initially demonstrating a statistical correlation to international communities with recently established 5G networks. In this study, we examined the peer-reviewed scientific literature on the detrimental bioeffects of WCR and identified several mechanisms by which WCR may have contributed to the COVID-19 pandemic as a toxic environmental cofactor. By crossing boundaries between the disciplines of biophysics and pathophysiology, we present evidence that WCR may: (1) cause morphologic changes in erythrocytes including echinocyte and rouleaux formation that can contribute to hypercoagulation; (2) impair microcirculation and reduce erythrocyte and hemoglobin levels exacerbating hypoxia; (3) amplify immune system dysfunction, including immunosuppression, autoimmunity, and hyperinflammation; (4) increase cellular oxidative stress and the production of free radicals resulting in vascular injury and organ damage; (5) increase intracellular $Ca^{2+}$ essential for viral entry, replication, and release, in addition to promoting pro-inflammatory pathways; and (6) worsen heart arrhythmias and cardiac disorders.

**Relevance for Patients:** In short, WCR has become a ubiquitous environmental stressor that we propose may have contributed to adverse health outcomes of patients infected with SARS-CoV-2 and increased the severity of the COVID-19 pandemic. Therefore, we recommend that all people, particularly those suffering from SARS-CoV-2 infection, reduce their exposure to WCR as much as reasonably achievable until further research better clarifies the systemic health effects associated with chronic WCR exposure.

## 1. Introduction

### 1.1. Background

Coronavirus disease 2019 (COVID-19) has been the focus of international public health policy since 2020. Despite unprecedented public health protocols to quell the pandemic, the number of COVID-19 cases continues to rise. We propose a reassessment of our public health strategies.

DOI: http://dx.doi.org/10.18053/jctres.07.202105.007

According to the Center for Disease Control and Prevention (CDC), the simplest model of disease causation is the epidemiological triad consisting of three interactive factors: the agent (pathogen), the environment, and the health status of the host [1]. Extensive research is being done on the agent, severe acute respiratory syndrome coronavirus 2 (SARS-CoV-2). Risk factors that make a host more likely to succumb to the disease have been elucidated. However, environmental factors have not been sufficiently explored. In this paper, we investigated the role of wireless communication radiation (WCR), a widespread environmental stressor.

We explore the scientific evidence suggesting a possible relationship between COVID-19 and radiofrequency radiation related to wireless communications technology including fifth generation (5G) of wireless communications technology, henceforth referred to as WCR. WCR has already been recognized as a form of environmental pollution and physiological stressor [2]. Assessing the potentially detrimental health effects of WCR may be crucial to develop an effective, rational public health policy that may help expedite eradication of the COVID-19 pandemic. In addition, because we are on the verge of worldwide 5G deployment, it is critical to consider the possible damaging health effects of WCR before the public is potentially harmed.

5G is a protocol that will use high frequency bands and extensive bandwidths of the electromagnetic spectrum in the vast radiofrequency range from 600 MHz to nearly 100 GHz, which includes millimeter waves (>20 GHz), in addition to the currently used third generation (3G) and fourth generation (4G) long-term evolution (LTE) microwave bands. 5G frequency spectrum allocations differ from country to country. Focused pulsed beams of radiation will emit from new base stations and phased array antennas placed close to buildings whenever persons access the 5G network. Because these high frequencies are strongly absorbed by the atmosphere and especially during rain, a transmitter's range is limited to 300 meters. Therefore, 5G requires base stations and antennas to be much more closely spaced than previous generations. Plus, satellites in space will emit 5G bands globally to create a wireless worldwide web. The new system therefore requires significant densification of 4G infrastructure as well as new 5G antennas that may dramatically increase the population's WCR exposure both inside structures and outdoors. Approximately 100,000 emitting satellites are planned to be launched into orbit. This infrastructure will significantly alter the world's electromagnetic environment to unprecedented levels and may cause unknown consequences to the entire biosphere, including humans. The new infrastructure will service the new 5G devices, including 5G mobile phones, routers, computers, tablets, self-driving vehicles, machine-to-machine communications, and the Internet of Things.

The global industry standard for 5G is set by the 3G Partnership Project (3GPP), which is an umbrella term for several organizations developing standard protocols for mobile telecommunications. The 5G standard specifies all key aspects of the technology, including frequency spectrum allocation, beam-forming, beam steering, multiplexing multiple in, multiple out schemes, as well as modulation schemes, among others. 5G will utilize from 64 to 256 antennas at short distances to serve virtually simultaneously a large number of devices within a cell. The latest finalized 5G standard, Release 16, is codified in the 3GPP published Technical Report TR 21.916 and may be downloaded from the 3GPP server at https://www.3gpp.org/specifications. Engineers claim that 5G will offer performance up to 10 times that of current 4G networks [3].

COVID-19 began in Wuhan, China in December 2019, shortly after city-wide 5G had "gone live," that is, become an operational system, on October 31, 2019. COVID-19 outbreaks soon followed in other areas where 5G had also been at least partially implemented, including South Korea, Northern Italy, New York City, Seattle, and Southern California. In May 2020, Mordachev [4] reported a statistically significant correlation between the intensity of radiofrequency radiation and the mortality from SARS-CoV-2 in 31 countries throughout the world. During the first pandemic wave in the United States, COVID-19 attributed cases and deaths were statistically higher in states and major cities with 5G infrastructure as compared with states and cities that did not yet have this technology [5].

There is a large body of peer reviewed literature, since before World War II, on the biological effects of WCR that impact many aspects of our health. In examining this literature, we found intersections between the pathophysiology of SARS-CoV-2 and detrimental bioeffects of WCR exposure. Here, we present the evidence suggesting that WCR has been a possible contributing factor exacerbating COVID-19.

### 1.2. Overview on COVID-19

The clinical presentation of COVID-19 has proven to be highly variable, with a wide range of symptoms and variability from case to case. According to the CDC, early disease symptoms may include sore throat, headache, fever, cough, chills, among others. More severe symptoms including shortness of breath, high fever, and severe fatigue may occur in a later stage. The neurological sequela of taste and smell loss has also been described.

Ing *et al.* [6] determined 80% of those affected have mild symptoms or none, but older populations and those with comorbidities, such as hypertension, diabetes, and obesity, have a greater risk for severe disease [7]. Acute respiratory distress syndrome (ARDS) can rapidly occur [8] and cause severe shortness of breath as endothelial cells lining blood vessels and epithelial cells lining airways lose their integrity, and protein rich fluid leaks into adjacent air sacs. COVID-19 can cause insufficient oxygen levels (hypoxia) that have been seen in up to 80% of intensive care unit (ICU) patients [9] exhibiting respiratory distress. Decreased oxygenation and elevated carbon dioxide levels in patients' blood have been observed, although the etiology for these findings remains unclear.

Massive oxidative damage to the lungs has been observed in areas of airspace opacification documented on chest radiographs and computed tomography (CT) scans in patients with SARS-CoV-2 pneumonia [10]. This cellular stress may indicate a biochemical rather than a viral etiology [11].

DOI: http://dx.doi.org/10.18053/jctres.07.202105.007

Because disseminated virus can attach itself to cells containing an angiotensin-converting enzyme 2 (ACE2) receptor; it can spread and damage organs and soft tissues throughout the body, including the lungs, heart, intestines, kidneys, blood vessels, fat, testes, and ovaries, among others. The disease can increase systemic inflammation and induce a hypercoagulable state. Without anticoagulation, intravascular blood clots can be devastating [12].

In COVID-19 patients referred to as "long-haulers," symptoms can wax and wane for months [13]. Shortness of breath, fatigue, joint pain, and chest pain can become persistent symptoms. Post-infectious brain fog, cardiac arrhythmia, and new onset hypertension have also been described. Long-term chronic complications of COVID-19 are being defined as epidemiological data are collected over time.

As our understanding of COVID-19 continues to evolve, environmental factors, particularly those of wireless communication electromagnetic fields, remain unexplored variables that may be contributing to the disease including its severity in some patients. Next, we summarize the bioeffects of WCR exposure from the peer reviewed scientific literature published over decades.

### 1.3. Overview on bioeffects of WCR exposure

Organisms are electrochemical beings. Low-level WCR from devices, including mobile telephony base antennas, wireless network protocols utilized for the local networking of devices and internet access, trademarked as Wi-Fi (officially IEEE 802.11b Direct Sequence protocol; IEEE, Institute of Electrical and Electronic Engineers) by the Wi-Fi alliance, and mobile phones, among others, may disrupt regulation of numerous physiological functions. Non-thermal bioeffects (below the power density that causes tissue heating) from very low-level WCR exposure have been reported in numerous peer-reviewed scientific publications at power densities below the International Commission on Non-Ionizing Radiation Protection (ICNIRP) exposure guidelines [14]. Low-level WCR has been found to impact the organism at all levels of organization, from the molecular to the cellular, physiological, behavioral, and psychological levels. Moreover, it has been shown to cause systemic detrimental health effects including increased cancer risk [15], endocrine changes [16], increased free radical production [17], deoxyribonucleic acid (DNA) damage [18], changes to the reproductive system [19], learning and memory defects [20], and neurological disorders [21]. Having evolved within Earth's extremely low-level natural radiofrequency background, organisms lack the ability to adapt to heightened levels of unnatural radiation of wireless communications technology with digital modulation that includes short intense pulses (bursts).

The peer-reviewed world scientific literature has documented evidence for detrimental bioeffects from WCR exposure including 5G frequencies over several decades. The Soviet and Eastern European literature from 1960 to 1970s demonstrates significant biological effects, even at exposure levels more

than 1000 times below 1 mW/cm$^2$, the current guideline for maximum public exposure in the US. Eastern studies on animal and human subjects were performed at low exposure levels (<1 mW/cm$^2$) for long durations (typically months). Adverse bioeffects from WCR exposure levels below 0.001 mW/cm$^2$ have also been documented in the Western literature. Damage to human sperm viability including DNA fragmentation by internet-connected laptop computers at power densities from 0.0005 to 0.001 mW/cm$^2$ has been reported [22]. Chronic human exposure to 0.000006 – 0.00001 mW/cm$^2$ produced significant changes in human stress hormones following a mobile phone base station installation [23]. Human exposures to cell phone radiation at 0.00001 – 0.00005 mW/cm$^2$ resulted in complaints of headache, neurological problems, sleep problems, and concentration problems, corresponding to "microwave sickness" [24,25]. The effects of WCR on prenatal development in mice placed near an "antenna park" exposed to power densities from 0.000168 to 0.001053 mW/cm$^2$ showed a progressive decrease in the number of newborns and ended in irreversible infertility [26]. Most US research has been performed over short durations of weeks or less. In recent years, there have been few long-term studies on animals or humans.

Illness from WCR exposure has been documented since the early use of radar. Prolonged exposure to microwaves and millimeter waves from radar was associated with various disorders termed "radio-wave sickness" decades ago by Russian scientists. A wide variety of bioeffects from nonthermal power densities of WCR were reported by Soviet research groups since the 1960s. A bibliography of over 3700 references on the reported biological effects in the world scientific literature was published in 1972 (revised 1976) by the US Naval Medical Research Institute [27,28]. Several relevant Russian studies are summarized as follows. Research on *Escherichia coli* bacteria cultures show power density windows for microwave resonance effects for 51.755 GHz stimulation of bacterial growth, observed at extremely low power densities of $10^{-13}$ mW/cm$^2$ [29], illustrating an extremely low level bioeffect. More recently Russian studies confirmed earlier results of Soviet research groups on the effects of 2.45 GHz at 0.5 mW/cm$^2$ on rats (30 days exposure for 7 h/day), demonstrating the formation of antibodies to the brain (autoimmune response) and stress reactions [30]. In a long-term (1 – 4 year) study comparing children who use mobile phones to a control group, functional changes, including greater fatigue, decreased voluntary attention, and weakening of semantic memory, among other adverse psychophysiological changes, were reported [31]. Key Russian research reports that underlie the scientific basis for Soviet and Russian WCR exposure guidelines to protect the public, which are much lower than the US guidelines, have been summarized [32].

By comparison to the exposure levels employed in these studies, we measured the ambient level of WCR from 100 MHz to 8 GHz in downtown San Francisco, California in December, 2020, and found an average power density of 0.0002 mW/cm$^2$. This level is from the superposition of multiple WCR devices. It is approximately $2 \times 10^{10}$ times above the natural background.

DOI: http://dx.doi.org/10.18053/jctres.07.202105.007

Pulsed radio-frequency radiation such as WCR exhibits substantially different bioeffects, both qualitatively and quantitatively (generally more pronounced) compared to continuous waves at similar time-averaged power densities [33-36]. The specific interaction mechanisms are not well understood. All types of wireless communications employ extremely low frequency (ELFs) in the modulation of the radiofrequency carrier signals, typically pulses to increase the capacity of information transmitted. This combination of radiofrequency radiation with ELF modulation(s) is generally more bioactive, as it is surmised that organisms cannot readily adapt to such rapidly changing wave forms [37-40]. Therefore, the presence of ELF components of radiofrequency waves from pulsing or other modulations must be considered in studies on the bioeffects of WCR. Unfortunately, the reporting of such modulations has been unreliable, especially in older studies [41].

The BioInitiative Report [42], authored by 29 experts from ten countries, and updated in 2020, provides a scholarly contemporary summary of the literature on the bioeffects and health consequences from WCR exposure, including a compendium of supporting research. Recent reviews have been published [43-46]. Two comprehensive reviews on the bioeffects of millimeter waves report that even short-term exposures produce marked bioeffects [47,48].

## 2. Methods

An ongoing literature study of the unfolding pathophysiology of SARS-CoV-2 was performed. To investigate a possible connection to bioeffects from WCR exposure, we examined over 250 peer-reviewed research reports from 1969 to 2021, including reviews and studies on cells, animals, and humans. We included the world literature in English and Russian reports translated to English, on radio frequencies from 600 MHz to 90 GHz, the carrier wave spectrum of WCR (2G to 5G inclusive), with particular emphasis on nonthermal, low power densities ($<1$ mW/cm$^2$), and long-term exposures. The following search terms were used in queries in MEDLINE® and the Defense Technical Information Center (https://discover.dtic.mil) to find relevant study reports: radiofrequency radiation, microwave, millimeter wave, radar, MHz, GHz, blood, red blood cell, erythrocyte, hemoglobin, hemodynamic, oxygen, hypoxia, vascular, inflammation, pro-inflammatory, immune, lymphocyte, T cell, cytokine, intracellular calcium, sympathetic function, arrhythmia, heart, cardiovascular, oxidative stress, glutathione, reactive oxygen species (ROS), COVID-19, virus, and SARS-CoV-2. Occupational studies on WCR exposed workers were included in the study. Our approach is akin to Literature-Related Discovery, in which two concepts that have heretofore not been linked are explored in the literature searches to look for linkage(s) to produce novel, interesting, plausible, and intelligible knowledge, that is, potential discovery [49]. From analysis of these studies in comparison with new information unfolding on the pathophysiology of SARS-CoV-2, we identified several ways in which adverse bioeffects of WCR exposure intersect with COVID-19 manifestations and organized our findings into five categories.

**Table 1.** Bioeffects of Wireless Communication Radiation (WCR) exposure in relation to COVID-19 manifestations and their progression

| Wireless communications radiation (WCR) exposure bioeffects | COVID-19 manifestations |
|---|---|
| **Blood changes** | **Blood changes** |
| Short-term: rouleaux, echinocytes | Rouleaux, echinocytes |
| Long-term: reduced blood clotting time, reduced hemoglobin, hemodynamic disorders | Hemoglobin effects; vascular effects |
| | → Reduced hemoglobin in severe disease; autoimmune hemolytic anemia; hypoxemia and hypoxia |
| | → Endothelial injury; impaired microcirculation; hypercoagulation; disseminated intravascular coagulopathy (DIC); pulmonary embolism; stroke |
| **Oxidative stress** | **Oxidative stress** |
| Glutathione level decrease; free radicals and lipid peroxide increase; superoxide dismutase activity decrease; oxidative injury in tissues and organs | Glutathione level decrease; free radical increase and damage; apoptosis |
| | → Oxidative injury; organ damage in severe disease |
| **Immune system disruption and activation** | **Immune system disruption and activation** |
| Immune suppression in some studies; immune hyperactivation in other studies | Decreased production of T-lymphocytes; elevated inflammatory biomarkers. |
| Long-term: suppression of T-lymphocytes; inflammatory biomarkers increased; autoimmunity; organ injury | → Immune hyperactivation and inflammation; cytokine storm in severe disease; cytokine-induced hypo-perfusion with resulting hypoxia; organ injury; organ failure |
| **Increased intracellular calcium** | **Increased intracellular calcium** |
| From activation of voltage-gated calcium channels on cell membranes, with numerous secondary effects | → Increased virus entry, replication, and release |
| | → Increased NF-κB, pro-inflammatory processes, coagulation, and thrombosis |
| **Cardiac effects** | **Cardiac effects** |
| Up-regulation of sympathetic nervous system; palpitations and arrhythmias | Arrhythmias |
| | → Myocarditis; myocardial ischemia; cardiac injury; cardiac failure |

Supportive evidence including study details and citations are provided in the text under each subject heading, i.e., blood changes, oxidative stress, etc.

DOI: http://dx.doi.org/10.18053/jctres.07.202105.007

# 3. Results

Table 1 lists the manifestations common to COVID-19 including disease progression and the corresponding adverse bioeffects from WCR exposure. Although these effects are delineated into categories — blood changes, oxidative stress, immune system disruption and activation, increased intracellular calcium ($Ca^{2+}$), and cardiac effects — it must be emphasized that these effects are not independent of each other. For example, blood clotting and inflammation have overlapping mechanisms, and oxidative stress is implicated in erythrocyte morphological changes as well as in hypercoagulation, inflammation, and organ damage.

## 3.1. Blood changes

WCR exposure can cause morphologic changes in blood readily seen through phase contrast or dark-field microscopy of live peripheral blood samples. In 2013, Havas observed erythrocyte aggregation including rouleaux (rolls of stacked red blood cells) in live peripheral blood samples following 10 min human exposure to a 2.4 GHz cordless phone [50]. Although not peer reviewed, one of us (Rubik) investigated the effect of 4G LTE mobile phone radiation on the peripheral blood of ten human subjects, each of whom had been exposed to cell phone radiation for two consecutive 45-min intervals [51]. Two types of effects were observed: increased stickiness and clumping of red blood cells with rouleaux formation, and subsequent formation of echinocytes (spiky red blood cells). Red blood cell clumping and aggregation are known to be actively involved in blood clotting [52]. The prevalence of this phenomenon on exposure to WCR in the human population has not yet been determined. Larger controlled studies should be performed to further investigate this phenomenon.

Similar red blood cell changes have been described in peripheral blood of COVID-19 patients [53]. Rouleaux formation has been observed in 1/3 of COVID-19 patients, whereas spherocytes and echinocyte formation is more variable. Spike protein engagement with ACE2 receptors on cells lining the blood vessels can lead to endothelial damage, even when isolated [54]. Rouleaux formation, particularly in the setting of underlying endothelial damage, can clog the microcirculation, impeding oxygen transport, contributing to hypoxia, and increasing the risk of thrombosis [52]. Thrombogenesis associated with SARS-CoV-2 infection may also be caused by direct viral binding to ACE2 receptors on platelets [55].

Additional blood effects have been observed in both humans and animals exposed to WCR. In 1977, a Russian study reported that rodents irradiated with 5 – 8 mm waves (60 – 37 GHz) at 1 mW/cm$^2$ for 15 min/day over 60 days developed hemodynamic disorders, suppressed red blood cell formation, reduced hemoglobin, and an inhibition of oxygen utilization (oxidative phosphorylation by the mitochondria) [56]. In 1978, a 3-year Russian study on 72 engineers exposed to millimeter wave generators emitting at 1 mW/cm$^2$ or less showed a decrease in their hemoglobin levels and red blood cell counts, and a tendency toward hypercoagulation, whereas a control group showed no changes [57]. Such deleterious hematologic effects from WCR exposure may also contribute to the development of hypoxia and blood clotting observed in COVID-19 patients.

It has been proposed that the SARS-CoV-2 virus attacks erythrocytes and causes degradation of hemoglobin [11]. Viral proteins may attack the 1-beta chain of hemoglobin and capture the porphyrin, along with other proteins from the virus catalyzing the dissociation of iron from heme [58]. In principle this would reduce the number of functional erythrocytes and cause the release of free iron ions that could cause oxidative stress, tissue damage, and hypoxia. With hemoglobin partially destroyed and lung tissue damaged by inflammation, patients would be less able to exchange carbon dioxide ($CO_2$) and oxygen ($O_2$), and would become oxygen depleted. In fact, some COVID-19 patients show reduced hemoglobin levels, measuring 7.1 g/L and even as low as 5.9 g/L in severe cases [59]. Clinical studies of almost 100 patients from Wuhan revealed that the hemoglobin levels in the blood of most patients infected with SARS-CoV-2 are significantly lowered resulting in compromised delivery of oxygen to tissues and organs [60]. In a meta-analysis of four studies with a total of 1210 patients and 224 with severe disease, hemoglobin values were reduced in COVID-19 patients with severe disease compared to those with milder forms [59]. In another study on 601 COVID-19 patients, 14.7% of anemic COVID-19 ICU patients and 9% of non-ICU COVID-19 patients had autoimmune hemolytic anemia [61]. In patients with severe COVID-19 disease, decreased hemoglobin along with elevated erythrocyte sedimentation rate (ESR), C-reactive protein, lactate dehydrogenase, albumin [62], serum ferritin [63], and low oxygen saturation [64] provide additional support for this hypothesis. In addition, packed red blood cell transfusion may promote recovery of COVID-19 patients with acute respiratory failure [65].

In short, both WCR exposure and COVID-19 may cause deleterious effects on red blood cells and reduced hemoglobin levels contributing to hypoxia in COVID-19. Endothelial injury may further contribute to hypoxia and many of the vascular complications seen in COVID-19 [66] that are discussed in the next section.

## 3.2. Oxidative stress

Oxidative stress is a non-specific pathological condition reflecting an imbalance between an increased production of ROS and an inability of the organism to detoxify the ROS or to repair the damage they cause to biomolecules and tissues [67]. Oxidative stress can disrupt cell signaling, cause the formation of stress proteins, and generate highly reactive free radicals, which can cause DNA and cell membrane damage.

SARS-CoV-2 inhibits intrinsic pathways designed to reduce ROS levels, thereby increasing morbidity. Immune dysregulation, that is, the upregulation of interleukin (IL)-6 and tumor necrosis factor $\alpha$ (TNF-$\alpha$) [68] and suppression of interferon (IFN) $\alpha$ and IFN $\beta$ [69] have been identified in the cytokine storm accompanying severe COVID-19 infections and generates oxidative stress [10]. Oxidative stress and mitochondrial dysfunction may further perpetuate the cytokine storm, worsening tissue damage, and increasing the risk of severe illness and death.

DOI: http://dx.doi.org/10.18053/jctres.07.202105.007

Similarly low-level WCR generates ROS in cells that cause oxidative damage. In fact, oxidative stress is considered to be one of the primary mechanisms in which WCR exposure causes cellular damage. Among 100 currently available peer-reviewed studies investigating oxidative effects of low-intensity WCR, 93 of these studies confirmed that WCR induces oxidative effects in biological systems [17]. WCR is an oxidative agent with a high pathogenic potential especially when exposure is continuous [70].

Oxidative stress is also an accepted mechanism causing endothelial damage [71]. This may manifest in patients with severe COVID-19 in addition to increasing the risk for blood clot formation and worsening hypoxemia [10]. Low levels of glutathione, the master antioxidant, have been observed in a small group of COVID-19 patients, with the lowest level found in the most severe cases [72]. The finding of low glutathione levels in these patients further supports oxidative stress as a component of this disease [72]. In fact, glutathione, the major source of sulfhydryl-based antioxidant activity in the human body, may be pivotal in COVID-19 [73]. Glutathione deficiency has been proposed as the most likely cause of serious manifestations in COVID-19 [72]. The most common co-morbidities, hypertension [74]; obesity [75]; diabetes [76]; and chronic obstructive pulmonary disease [74] support the concept that pre-existing conditions causing low levels of glutathione may work synergistically to create the "perfect storm" for both the respiratory and vascular complications of severe infection. Another paper citing two cases of COVID-19 pneumonia treated successfully with intravenous glutathione also supports this hypothesis [77].

Many studies report oxidative stress in humans exposed to WCR. Peraica *et al.* [78] found diminished blood levels of glutathione in workers exposed to WCR from radar equipment (0.01 mW/cm$^2$ – 10 mW/cm$^2$; 1.5 – 10.9 GHz). Garaj-Vrhovac *et al.* [79] studied bioeffects following exposure to non-thermal pulsed microwaves from marine radar (3 GHz, 5.5 GHz, and 9.4 GHz) and reported reduced glutathione levels and increased malondialdehyde (marker for oxidative stress) in an occupationally exposed group [79]. Blood plasma of individuals residing near mobile phone base stations showed significantly reduced glutathione, catalase, and superoxide dismutase levels over unexposed controls [80]. In a study on human exposure to WCR from mobile phones, increased blood levels of lipid peroxide were reported, while enzymatic activities of superoxide dismutase and glutathione peroxidase in the red blood cells decreased, indicating oxidative stress [81].

In a study on rats exposed to 2450 MHz (wireless router frequency), oxidative stress was implicated in causing red blood cell lysis (hemolysis) [82]. In another study, rats exposed to 945 MHz (base station frequency) at 0.367 mW/cm$^2$ for 7 h/day, over 8 days, demonstrated low glutathione levels and increased malondialdehyde and superoxide dismutase enzyme activity, hallmarks for oxidative stress [83]. In a long-term controlled study on rats exposed to 900 MHz (mobile phone frequency) at 0.0782 mW/cm$^2$ for 2 h/day for 10 months, there was a significant increase in malondialdehyde and total oxidant status over controls [84]. In another long-term controlled study on rats exposed to two mobile phone frequencies, 1800 MHz and 2100

MHz, at power densities 0.04 – 0.127 mW/cm$^2$ for 2 h/day over 7 months, significant alterations in oxidant-antioxidant parameters, DNA strand breaks, and oxidative DNA damage were found [85].

There is a correlation between oxidative stress and thrombogenesis [86]. ROS can cause endothelial dysfunction and cellular damage. The endothelial lining of the vascular system contains ACE2 receptors that are targeted by SARS-CoV-2. The resulting endotheliitis can cause luminal narrowing and result in diminished blood flow to downstream structures. Thrombi in arterial structures can further obstruct blood flow causing ischemia and/or infarcts in involved organs, including pulmonary emboli and strokes. Abnormal blood coagulation leading to micro-emboli was a recognized complication early in the history of COVID-19 [87]. Out of 184 ICU COVID-19 patients, 31% showed thrombotic complications [88]. Cardiovascular clotting events are a common cause of COVID-19 deaths [12]. Pulmonary embolism, disseminated intravascular coagulation (DIC), liver, cardiac, and renal failure have all been observed in COVID-19 patients [89].

Patients with the highest cardiovascular risk factors in COVID-19 includ males, the elderly, diabetics, and obese and hypertensive patients. However, increased incidence of strokes in younger patients with COVID-19 has also been described [90].

Oxidative stress is caused by WCR exposure and is known to be implicated in cardiovascular disease. Ubiquitous environmental exposure to WCR may contribute to cardiovascular disease by creating a chronic state of oxidative stress [91]. This would lead to oxidative damage to cellular constituents and alter signal transduction pathways. In addition, pulse-modulated WCR can cause oxidative injury in liver, lung, testis, and heart tissues mediated by lipid peroxidation, increased levels of nitric oxides, and suppression of the antioxidant defense mechanism [92].

In summary, oxidative stress is a major component in the pathophysiology of COVID-19 as well as in cellular damage caused by WCR exposure.

### 3.3. Immune system disruption and activation

When SARS-CoV-2 first infects the human body, it attacks cells lining the nose, throat, and upper airway harboring ACE2 receptors. Once the virus gains access to a host cell through one of its spike proteins, which are the multiple protuberances projecting from the viral envelope that bind to ACE2 receptors, it converts the cell into a virus self-replicating entity.

In response to COVID-19 infection, both an immediate systemic innate immune response as well as a delayed adaptive response has been shown to occur [93]. The virus can also cause a dysregulation of the immune response, particularly in the decreased production of T-lymphocytes. [94]. Severe cases tend to have lower lymphocyte counts, higher leukocyte counts and neutrophil-lymphocyte ratios, as well as lower percentages of monocytes, eosinophils, and basophils [94]. Severe cases of COVID-19 show the greatest impairment in T-lymphocytes.

In comparison, low-level WCR studies on laboratory animals also show impaired immune function [95]. Findings

include physical alterations in immune cells, a degradation of immunological responses, inflammation, and tissue damage. Baranski [96] exposed guinea pigs and rabbits to continuous or pulse-modulated 3000 MHz microwaves at an average power density of 3.5 mW/cm$^2$ for 3 h/day over 3 months and found nonthermal changes in lymphocyte counts, abnormalities in nuclear structure, and mitosis in the erythroblastic cell series in the bone marrow and in lymphoid cells in lymph nodes and spleen. Other investigators have shown diminished T-lymphocytes or suppressed immune function in animals exposed to WCR. Rabbits exposed to 2.1 GHz at 5mW/cm$^2$ for 3 h/day, 6 days/week, for 3 months, showed suppression of T-lymphocytes [97]. Rats exposed to 2.45 GHz and 9.7 GHz for 2 h/day, 7 days/week, for 21 months showed a significant decrease in the levels of lymphocytes and an increase in mortality at 25 months in the irradiated group [98]. Lymphocytes harvested from rabbits irradiated with 2.45 GHz for 23 h/day for 6 months show a significant suppression in immune response to a mitogen [99].

In 2009, Johansson conducted a literature review, which included the 2007 Bioinitiative Report. He concluded that electromagnetic fields (EMF) exposure, including WCR, can disturb the immune system and cause allergic and inflammatory responses at exposure levels significantly less than current national and international safety limits and raise the risk for systemic disease [100]. A review conducted by Szmigielski in 2013 concluded that weak RF/microwave fields, such as those emitted by mobile phones, can affect various immune functions both *in vitro* and *in vivo* [101]. Although the effects are historically somewhat inconsistent, most research studies document alterations in the number and activity of immune cells from RF exposure. In general, short-term exposure to weak microwave radiation may temporarily stimulate an innate or adaptive immune response, but prolonged irradiation inhibits those same functions.

In the acute phase of COVID-19 infection, blood tests demonstrate elevated ESR, C-reactive protein, and other elevated inflammatory markers [102], typical of an innate immune response. Rapid viral replication can cause death of epithelial and endothelial cells and result in leaky blood vessels and pro-inflammatory cytokine release [103]. Cytokines, proteins, peptides, and proteoglycans that modulate the body's immune response, are modestly elevated in patients with mild-to-moderate disease severity [104]. In those with severe disease, an uncontrolled release of pro-inflammatory cytokines--a cytokine storm--can occur. Cytokine storms originate from an imbalance in T-cell activation with dysregulated release of IL-6, IL-17, and other cytokines. Programmed cell death (apoptosis), ARDS, DIC, and multi-organ system failure can all result from a cytokine storm and increase the risk of mortality.

By comparison, Soviet researchers found in the 1970s that radiofrequency radiation can damage the immune system of animals. Shandala [105] exposed rats to 0.5 mW/cm$^2$ microwaves for 1 month, 7 h/day, and found impaired immune competence and induction of autoimmune disease. Rats irradiated with 2.45 GHz at 0.5 mW/cm$^2$ for 7 h daily for 30 days produced autoimmune reactions, and 0.1 – 0.5 mW/cm$^2$ produced persistent pathological

immune reactions [106]. Exposure to microwave radiation, even at low levels (0.1 – 0.5 mW/cm$^2$), can impair immune function, causing physical alterations in the essential cells of the immune system and a degradation of immunologic responses [107]. Szabo *et al.* [108] examined the effects of 61.2 GHz exposure on epidermal keratinocytes and found an increase in IL-1b, a pro-inflammatory cytokine. Makar *et al.* [109] found that immunosuppressed mice irradiated 30 min/day for 3 days by 42.2 GHz showed increased levels of TNF-α, a cytokine produced by macrophages.

In short, COVID-19 can lead to immune dysregulation as well as cytokine storms. By comparison, exposure to low-level WCR as observed in animal studies can also compromise the immune system, with chronic daily exposure producing immunosuppression or immune dysregulation including hyperactivation.

### 3.4. Increased intracellular calcium

In 1992, Walleczek first suggested that ELF electromagnetic fields (<3000 Hz) may be affecting membrane-mediated Ca$^{2+}$ signaling and lead to increased intracellular Ca$^{2+}$ [110]. The mechanism of irregular gating of voltage-gated ion channels in cell membranes by polarized and coherent, oscillating electric or magnetic fields was first presented in 2000 and 2002 [40,111]. Pall [112] in his review of WCR-induced bioeffects combined with use of calcium channel blockers (CCB) noted that voltage-gated calcium channels play a major role in WCR bioeffects. Increased intracellular Ca$^{+2}$ results from the activation of voltage-gated calcium channels, and this may be one of the primary mechanisms of action of WCR on organisms.

Intracellular Ca$^{2+}$ is essential for virus entry, replication, and release. It has been reported that some viruses can manipulate voltage-gated calcium channels to increase intracellular Ca$^{2+}$ thereby facilitating viral entry and replication [113]. Research has shown that the interaction between a virus and voltage-gated calcium channels promote virus entry at the virus-host cell fusion step [113]. Thus, after the virus binds to its receptor on a host cell and enters the cell through endocytosis, the virus takes over the host cell to manufacture its components. Certain viral proteins then manipulate calcium channels, thereby increasing intracellular Ca$^{2+}$, which facilitates further viral replication.

Even though direct evidence has not been reported, there is indirect evidence that increased intracellular Ca$^{2+}$ may be involved in COVID-19. In a recent study, elderly hospitalized COVID-19 patients treated with CCBs, amlodipine or nifedipine, were more likely to survive and less likely to require intubation or mechanical ventilation than controls [114]. Furthermore, CCBs strongly limit SARS-CoV-2 entry and infection in cultured epithelial lung cells [115]. CCBs also block the increase of intracellular Ca$^{2+}$ caused by WCR exposure as well as exposure to other electromagnetic fields [112].

Intracellular Ca$^{2+}$ is a ubiquitous second messenger relaying signals received by cell surface receptors to effector proteins involved in numerous biochemical processes. Increased intracellular Ca$^{2+}$ is a significant factor in upregulation of transcription nuclear factor KB (NF-κB) [116], an important

DOI: http://dx.doi.org/10.18053/jctres.07.202105.007

regulator of pro-inflammatory cytokine production as well as coagulation and thrombotic cascades. NF-κB is hypothesized to be a key factor underlying severe clinical manifestations of COVID-19 [117].

In short, WCR exposure, therefore, may enhance the infectivity of the virus by increasing intracellular $Ca^{2+}$ that may also indirectly contribute to inflammatory processes and thrombosis.

### 3.5. Cardiac effects

Cardiac arrhythmias are more commonly encountered in critically ill patients with COVID-19 [118]. The cause for arrhythmia in COVID-19 patients is multifactorial and includes cardiac and extra-cardiac processes [119]. Direct infection of the heart muscle by SARS-CoV-19 causing myocarditis, myocardial ischemia caused by a variety of etiologies, and heart strain secondary to pulmonary or systemic hypertension can result in cardiac arrhythmia. Hypoxemia caused by diffuse pneumonia, ARDS, or extensive pulmonary emboli represent extra-cardiac causes of arrhythmia. Electrolyte imbalances, intravascular fluid imbalance, and side effects from pharmacologic regimens can also result in arrhythmias in COVID-19 patients. Patients admitted to ICUs have been shown to have a higher increase in cardiac arrhythmias, 16.5% in one study [120]. Although no correlation between EMFs and arrhythmia in COVID-19 patients has been described in the literature, many ICUs are equipped with wireless patient monitoring equipment and communication devices producing a wide range of EMF pollution [121].

COVID-19 patients commonly show increased levels of cardiac troponin, indicating damage to the heart muscle [122]. Cardiac damage has been associated with arrhythmias and increased mortality. Cardiac injury is thought to be more often secondary to pulmonary emboli and viral sepsis, but direct infection of the heart, that is, myocarditis, can occur through direct viral binding to ACE2 receptors on cardiac pericytes, affecting local, and regional cardiac blood flow [60].

Immune system activation along with alterations in the immune system may result in atherosclerotic plaque instability and vulnerability, that is, presenting an increased risk for thrombus formation, and contributing to development of acute coronary events and cardiovascular disease in COVID-19.

Regarding WCR exposure bioeffects, in 1969 Christopher Dodge of the Biosciences Division, U.S. Naval Observatory in Washington DC, reviewed 54 papers and reported that radiofrequency radiation can adversely affect all major systems of the body, including impeding blood circulation; altering blood pressure and heart rate; affecting electrocardiograph readings; and causing chest pain and heart palpitations [123]. In the 1970s Glaser reviewed more than 2000 publications on radiofrequency radiation exposure bioeffects and concluded that microwave radiation can alter the electrocardiogram, cause chest pain, hypercoagulation, thrombosis, and hypertension in addition to myocardial infarction [27,28]. Seizures, convulsions, and alteration of the autonomic nervous system response (increased sympathetic stress response) have also been observed.

Since then, many other researchers have concluded that WCR exposure can affect the cardiovascular system. Although the nature of the primary response to millimeter waves and consequent events are poorly understood, a possible role for receptor structures and neural pathways in the development of continuous millimeter wave-induced arrhythmia has been proposed [47]. In 1997, a review reported that some investigators discovered cardiovascular changes including arrhythmias in humans from long-term low-level exposure to WCR including microwaves [124]. However, the literature also shows some unconfirmed findings as well as some contradictory findings [125]. Havas *et al.* [126] reported that human subjects in a controlled, double-blinded study were hyper-reactive when exposed to 2.45 GHz, digitally pulsed (100 Hz) microwave radiation, developing either an arrhythmia or tachycardia and upregulation of the sympathetic nervous system, which is associated with the stress response. Saili *et al.* [127] found that exposure to Wi-Fi (2.45 GHz pulsed at 10 Hz) affects heart rhythm, blood pressure, and the efficacy of catecholamines on the cardiovascular system, indicating that WCR can act directly and/or indirectly on the cardiovascular system. Most recently, Bandara and Weller [91] present evidence that people who live near radar installations (millimeter waves: 5G frequencies) have a greater risk of developing cancer and experiencing heart attacks. Similarly, those occupationally exposed have a greater risk of coronary heart disease. Microwave radiation affects the heart, and some people are more vulnerable if they have an underlying heart abnormality [128]. More recent research suggests that millimeter waves may act directly on the pacemaker cells of the sinoatrial node of the heart to change the beat frequency, which may underlie arrhythmias and other cardiac issues [47].

In short, both COVID-19 and WCR exposure can affect the heart and cardiovascular system, directly and/or indirectly.

## 4. Discussion

Epidemiologists, including those at the CDC, consider multiple causal factors when evaluating the virulence of an agent and understanding its ability to spread and cause disease. Most importantly, these variables include environmental cofactors and the health status of the host. Evidence from the literature summarized here suggests a possible connection between several adverse health effects of WCR exposure and the clinical course of COVID-19 in that WCR may have worsened the COVID-19 pandemic by weakening the host and exacerbating COVID-19 disease. However, none of the observations discussed here prove this linkage. Specifically, the evidence does not confirm causation. Clearly COVID-19 occurs in regions with little wireless communication. Furthermore, the relative morbidity caused by WCR exposure in COVID-19 is unknown.

We recognize that many factors have influenced the pandemic's course. Before restrictions were imposed, travel patterns facilitated the seeding of the virus, causing early rapid global spread. Population density, higher mean population age, and socioeconomic factors certainly influenced early viral spread. Air pollution, especially particulate matter $PM_{2.5}$ (2.5

micro-particulates), likely increased symptoms in patients with COVID-19 lung disease [129].

We postulate that WCR possibly contributed to the early spread and severity of COVID-19. Once an agent becomes established in a community, its virulence increases [130]. This premise can be applied to the COVID-19 pandemic. We surmise that "hot spots" of the disease that initially spread around the world were perhaps seeded by air travel, which in some areas were associated with 5G implementation. However, once the disease became established in those communities, it was able to spread more easily to neighboring regions where populations were less exposed to WCR. Second and third waves of the pandemic disseminated widely throughout communities with and without WCR, as might be expected.

The COVID-19 pandemic has offered us an opportunity to delve further into the potential adverse effects of WCR exposure on human health. Human exposure to ambient WCR significantly increased in 2020 as a "side effect" to the pandemic. Stay-at-home measures designed to reduce the spread of COVID-19 inadvertently resulted in greater public exposure to WCR, as people conducted more business and school related activities through wireless communications. Telemedicine created another source of WCR exposure. Even hospital inpatients, particular ICU patients, experienced increased WCR exposure as new monitoring devices utilized wireless communication systems that may exacerbate health disorders. It would potentially provide valuable information to measure ambient WCR power densities in home and work environments when comparing disease severity in patient populations with similar risk factors.

The question of causation could be investigated in future studies. For example, a clinical study could be conducted in COVID-19 patient populations with similar risk factors, to measure the WCR daily dose in COVID-19 patients and look for a correlation with disease severity and progression over time. As wireless device carrier frequencies and modulations may differ, and the power densities of WCR fluctuate constantly at a given location, this study would require patients to wear personal microwave dosimeters (monitoring badges). In addition, controlled laboratory studies could be conducted on animals, for example, humanized mice infected with SARS-CoV-2, in which groups of animals exposed to minimal WCR (control group) as well as medium and high power densities of WCR could be compared for disease severity and progression.

A major strength of this paper is that the evidence rests on a large body of scientific literature reported by many scientists worldwide and over several decades--experimental evidence of adverse bioeffects of WCR exposure at nonthermal levels on humans, animals, and cells. The Bioinitiative Report [42], updated in 2020, summarizes hundreds of peer-reviewed scientific papers documenting evidence of nonthermal effects from exposures $\leq 1$ mW/cm$^2$. Even so, some laboratory studies on the adverse health effects of WCR have sometimes utilized power densities exceeding 1mW/cm$^2$. In this paper, almost all of the studies that we reviewed included experimental data at power densities $\leq 1$ mW/cm$^2$.

A potential criticism of this paper is that adverse bioeffects from nonthermal exposures are not yet universally accepted in science. Moreover, they are not yet considered in establishing public health policy in many nations. Decades ago, Russians and Eastern Europeans compiled considerable data on nonthermal bioeffects, and subsequently set guidelines at lower radiofrequency radiation exposure limits than the US and Canada, that is, below levels where nonthermal effects are observed. However, the Federal Communications Commission (FCC, a US government entity) and ICNIRP guidelines operate on thermal limits based on outdated data from decades ago, allowing the public to be exposed to considerably higher radiofrequency radiation power densities. Regarding 5G, the telecommunication industry claims that it is safe because it complies with current radiofrequency radiation exposure guidelines of the FCC and ICNIRP. These guidelines were established in 1996 [131], are antiquated, and are not safety standards. Thus, there are no universally accepted safety standards for wireless communication radiation exposure. Recently international bodies, such as the EMF Working Group of the European Academy of Environmental Medicine, have proposed much lower guidelines, taking into account nonthermal bioeffects from WCR exposure in multiple sources [132].

Another weakness of this paper is that some of the bioeffects from WCR exposure are inconsistently reported in the literature. Replicated studies are often not true replications. Small differences in method, including unreported details, such as prior history of exposure of the organisms, non-uniform body exposure, and other variables can lead to inadvertent inconsistency. Moreover, not surprisingly, industry-sponsored studies tend to show less adverse bioeffects than studies conducted by independent researchers, suggesting industry bias [133]. Some experimental studies that are not industry-sponsored have also shown no evidence of harmful effects of WCR exposure. It is noteworthy, however, that studies employing real-life WCR exposures from commercially available devices have shown high consistency in revealing adverse effects [134].

WCR bioeffects depend on specific values of wave parameters including frequency, power density, polarization, exposure duration, modulation characteristics, as well as the cumulative history of exposure and background levels of electromagnetic, electric and magnetic fields. In laboratory studies, bioeffects observed also depend on genetic parameters and physiological parameters such as oxygen concentration [135]. The reproducibility of bioeffects of WCR exposure has sometimes been difficult due to failure to report and/or control all of these parameters. Similar to ionizing radiation, the bioeffects of WCR exposure can be subdivided into deterministic, that is, dose-dependent effects and stochastic effects that are seemingly random. Importantly, WCR bioeffects can also involve "response windows" of specific parameters whereby extremely low-level fields can have disproportionally detrimental effects [136]. This nonlinearity of WCR bioeffects can result in biphasic responses such as immune suppression from one range of parameters, and immune hyperactivation from another range of parameters, leading to variations that may appear inconsistent.

In gathering reports and examining existing data for this paper, we looked for outcomes providing evidence to support a proposed connection between the bioeffects of WCR exposure and

DOI: http://dx.doi.org/10.18053/jctres.07.202105.007

COVID-19. We did not make an attempt to weigh the evidence. The radiofrequency radiation exposure literature is extensive and currently contains over 30,000 research reports dating back several decades. Inconsistencies in nomenclature, reporting of details, and cataloging of keywords make it difficult to navigate this enormous literature.

Another shortcoming of this paper is that we do not have access to experimental data on 5G exposures. In fact, little is known about population exposure from real-world WCR, which includes exposure to WCR infrastructure and the plethora of WCR emitting devices. In relation to this, it is difficult to accurately quantify the average power density at a given location, which varies greatly, depending on the time, specific location, time-averaging interval, frequency, and modulation scheme. For a specific municipality it depends on the antenna density, which network protocols are used, as, for example, 2G, 3G, 4G, 5G, Wi-Fi, WiMAX (Worldwide Interoperability for Microwave Access), DECT (Digitally Enhanced Cordless Telecommunications), and RADAR (Radio Detection and Ranging). There is also WCR from ubiquitous radio wave transmitters, including antennas, base stations, smart meters, mobile phones, routers, satellites, and other wireless devices currently in use. All of these signals superimpose to yield the total average power density at a given location that typically fluctuates greatly over time. No experimental studies on adverse health effects or safety issues of 5G have been reported, and none are currently planned by the industry, although this is sorely needed.

Finally, there is an inherent complexity to WCR that makes it very difficult to fully characterize wireless signals in the real world that may be associated with adverse bioeffects. Real world digital communication signals, even from single wireless devices, have highly variable signals: variable power density, frequency, modulation, phase, and other parameters changing constantly and unpredictably each moment, as associated with the short, rapid pulsations used in digital wireless communication [137]. For example, in using a mobile phone during a typical phone conversation, the intensity of emitted radiation varies significantly each moment depending on signal reception, number of subscribers sharing the frequency band, location within the wireless infrastructure, presence of objects and metallic surfaces, and "speaking" versus "non-speaking" mode, among others. Such variations may reach 100% of the average signal intensity. The carrier radiofrequency constantly changes between different values within the available frequency band. The greater the amount of information (text, speech, internet, video, etc.), the more complex the communication signals become. Therefore, we cannot estimate accurately the values of these signal parameters including ELF components or predict their variability over time. Thus, studies on the bioeffects of WCR in the laboratory can only be representative of real-world exposures [137].

This paper points to the need for further research on nonthermal WCR exposure and its potential role in COVID-19. Moreover, some of the WCR exposure bioeffects that we discuss here — oxidative stress, inflammation, and immune system disruption — are common to many chronic diseases, including autoimmune disease

and diabetes. Thus, we hypothesize that WCR exposure may also be a potential contributing factor in many chronic diseases.

When a course of action raises threats of harm to human health, precautionary measures should be taken, even if clear causal relationships are not yet fully established. Therefore, we must apply the Precautionary Principle [138] regarding wireless 5G. The authors urge policymakers to execute an immediate worldwide moratorium on wireless 5G infrastructure until its safety can be assured.

Several unresolved safety issues should be addressed before wireless 5G is further implemented. Questions have been raised about 60 GHz, a key 5G frequency planned for extensive use, which is a resonant frequency of the oxygen molecule [139]. It is possible that adverse bioeffects might ensue from oxygen absorption of 60 GHz. In addition, water shows broad absorption in the GHz spectral region along with resonance peaks, for example, strong absorption at 2.45 GHz that is used in 4G Wi-Fi routers. This raises safety issues about GHz exposure of the biosphere, since organisms are comprised of mostly water, and changes in the structure of water due to GHz absorption have been reported that affect organisms [140]. Bioeffects from prolonged WCR exposure of the whole body need to be investigated in animal and human studies, and long-term exposure guidelines need to be considered. Independent scientists in particular should conduct concerted research to determine the biological effects of real-world exposure to WCR frequencies with digital modulation from the multiplicity of wireless communication devices. Testing could also include real-life exposures to multiple toxins (chemical and biological) [141], because multiple toxins may lead to synergistic effects. Environmental impact assessments are also needed. Once the long-term biological effects of wireless 5G are understood, we can set clear safety standards of public exposure limits and design an appropriate strategy for safe deployment.

## 5. Conclusion

There is a substantial overlap in pathobiology between COVID-19 and WCR exposure. The evidence presented here indicates that mechanisms involved in the clinical progression of COVID-19 could also be generated, according to experimental data, by WCR exposure. Therefore, we propose a link between adverse bioeffects of WCR exposure from wireless devices and COVID-19.

Specifically, evidence presented here supports a premise that WCR and, in particular, 5G, which involves densification of 4G, may have exacerbated the COVID-19 pandemic by weakening host immunity and increasing SARS-CoV-2 virulence by (1) causing morphologic changes in erythrocytes including echinocyte and rouleaux formation that may be contributing to hypercoagulation; (2) impairing microcirculation and reducing erythrocyte and hemoglobin levels exacerbating hypoxia; (3) amplifying immune dysfunction, including immunosuppression, autoimmunity, and hyperinflammation; (4) increasing cellular oxidative stress and the production of free radicals exacerbating vascular injury and organ damage; (5) increasing intracellular $Ca^{2+}$ essential for viral entry, replication, and release, in addition to promoting pro-

DOI: http://dx.doi.org/10.18053/jctres.07.202105.007

inflammatory pathways; and (6) worsening heart arrhythmias and cardiac disorders.

WCR exposure is a widespread, yet often neglected, environmental stressor that can produce a wide range of adverse bioeffects. For decades, independent research scientists worldwide have emphasized the health risks and cumulative damage caused by WCR [42,45]. The evidence presented here is consistent with a large body of established research. Healthcare workers and policymakers should consider WCR a potentially toxic environmental stressor. Methods for reducing WCR exposure should be provided to all patients and the general population.

## Acknowledgments

The authors acknowledge small contributions to early versions of this paper by Magda Havas and Lyn Patrick. We are grateful to Susan Clarke for helpful discussions and suggested edits of early drafts of the manuscript.

## Conflict of Interest

The authors declare that they have no conflicts of interest in preparing and publishing this manuscript. No competing financial interests exist.

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
