## [Reviewer Report · Evidence for a connection between COVID-19 and exposure to radiofrequency radiation from wireless communications including 5G]

Beverly Rubik, Robert R. Brown

Corresponding author: Beverly Rubik *College of Integrative Medicine & Health Sciences, Saybrook University, Pasadena CA; Institute for Frontier Science, Oakland, CA, USA*

Handeling editor:

Michal Heger

*Department of Pharmaceutics, Utrecht University, the Netherlands Department of Pharmaceutics, Jiaxing University Medical College, Zhejiang, China*

Review timeline:

Received: 10 March, 2021

Editorial decision: 12 May, 2021

Revision received: 11 June, 2021

Editorial decision: 28 June, 2021

Revision received: 28 July, 2021

Editorial decision: 03 August, 2021

Revision received: 25 August, 2021

Editorial decision: 25 August, 2021

Published online: 29 September, 2021

1st Editorial decision:

11-June-2021

Ref.: Ms. No. JCTRes-D-21-00034

Evidence for a Connection between COVID-19 and Exposure to Radiofrequency Radiation from Wireless Telecommunications Including Microwaves and Millimeter WavesOriginal Research Paper Journal of Clinical and Translational Research

Dear Dr. Rubik,

Reviewers have now commented on your paper. You will see that they are advising that you revise your manuscript. If you are prepared to undertake the work required, I would be pleased to reconsider my decision.

For your guidance, reviewers’ comments are appended below.

If you decide to revise the work, please submit a list of changes or a rebuttal against each point which is being raised when you submit the revised manuscript. Also, please ensure that the track changes function is switched on when implementing the revisions. This enables the reviewers to rapidly verify all changes made.

Your revision is due by Jun 11, 2021.

To submit a revision, go to https://www.editorialmanager.com/jctres/ and log in as an Author. You will see a menu item call Submission Needing Revision. You will find your submission record there.

Yours sincerely

Michal Heger

Editor-in-Chief

Journal of Clinical and Translational Research

Reviewers’ comments:

Reviewer #1: General comments

The author has considered very important and actual issue putting forward the question whether 5G mobile communication may contribute to the Covid-19 pandemia. To answer this question the author considers several effects in common between Covid patients and RF effects. These effects are summarized in the Table and described in the text referring to over 250 papers retrieved by the author from MEDLINE. However, this consideration has some limitations, which make the conclusions of the author rather immature.

First, the author compares the data on Covid patiens with the data on RF effects, which were obtained in vitro, animal, and human studies. As far as author has considered only about 250 papers from more than 30000 published papers on RF effects, it seems to be reasonable to retrieve for comparison mostly human RF studies.

Second, there is significant number of RF studies where no effects mentioned in the Table were observed. The author considers the RF effects in general, regardless dependences on frequencies, intensities and other key parameters, which were shown to be critical for the RF effects and have been most comprehensively reviewed in the IARC monograph (IARC 2013). Indeed, the author acknowledges these complicated dependences in the Discussion. Nevertheless, statement is made “that RFR and, in particular, 5G, which involves densification of 4G, has exacerbated the COVID-19 pandemic by weakening host immunity and increasing SARS-CoV-2 virulence”. However, this statement would demand consideration of RF effects at signals (i.e. frequency, modulation,…) and intensities as users of 5G are exposed to. As far as 5G is in focus, technical description of 5G signals is needed and studies of effects of RF signals with the same or similar characteristics should be retrieved and reviewed.

In conclusion, while the effects of RF and changes observed in Covid patients seem to overlap at some specific conditions of exposure, and given that the RF effects depend on number of physical and biological variables, more stringent retrieval of RF studies is needed to discuss possible connection of 5G exposure and Covid pandemia. Otherwise, this connection should be significantly down played and the text should be revised accordingly.

Specific comment

The author referred to the ICNIRP 2009 guidelines, which are outdated as far as ICNIRP has recently updated them. It should also be stated that in contrary to the ICNIRP thermally based guidelines, other international bodies such as EMF Working Group of the European Academy of Environmental Medicine has suggested much lower guidelines taking into account no-thermal RF effects reported in multiple studies (Belyaev, Dean et al. 2016).

Belyaev, I., A. Dean, H. Eger, G. Hubmann, R. Jandrisovits, M. Kern, M. Kundi, H. Moshammer, P. Lercher, K. Muller, G. Oberfeld, P. Ohnsorge, P. Pelzmann, C. Scheingraber and R. Thill (2016). “EUROPAEM EMF Guideline 2016 for the prevention, diagnosis and treatment of EMF-related health problems and illnesses.” Rev Environ Health 31(3): 363-397.

IARC (2013). IARC Monographs on the Evaluation of Carcinogenic Risks to Humans. Non-ionizing Radiation, Part 2: Radiofrequency Electromagnetic Fields Lyon, France, IARC Press.

Reviewer #3: This is a well-researched paper, with valuable insights into adverse effects from non-ionizing athermal radiation. However, the authors need to sharpen their language to clarify what has been demonstrated, and have the conclusions be fully reflective of what the data have shown. For example, on p.9 they state: “This evidence presented here does not claim causation.” Yet, on p.10 they state: “We conclude that RFR and, in particular, 5G, which involves densification of 4G, has exacerbated the COVID-19 pandemic by…..”. Later in the same paragraph, they state: “In short, wireless communication radiation is a ubiquitous environmental stressor, and evidence presented here suggests that it is a contributing factor in the COVID-19 pandemic.”

It appears to me they have shown the following type of relationships, as stated on p.7: “In summary, oxidative stress is a major component in the pathophysiology of COVID-19 as well as in cellular damage caused by RFR exposure. Similar effects are observed in both that are caused by increased free radical formation and glutathione deficiency.” Also, as stated on p.8: “In short, COVID-19 can lead to immune dysregulation as well as cytokine storm. By comparison, exposure to low-level RFR as observed in animal studies can also compromise the immune system, with chronic daily exposure producing immunosuppression or immune dysregulation including hyperactivation.”

They have shown quite convincingly that RFR produces a number of damaging bioeffects, and many of these damaging bioeffects are seen in COVID-19 patients. That constitutes a potential indirect linkage between RFR and COVID-19, and laboratory tests will be required to show whether a direct linkage exists. It seems to me that’s how their results need to be presented. The concepts behind the discipline of Literature-Related Discovery would strengthen the arguments for these types of linkages. Minor wording changes are all that are required in order to eliminate any confusion on what has been demonstrated.

Additionally, the authors need to sharpen their usage of the term COVID-19. It is a disease, and is not causing anything, as the authors imply. It is associated with a number of abnormal biomarkers, and this distinction needs to be delineated.

Reviewer #4: This manuscript is totally lacking in scientific value. The statement “This is the first scientific paper documenting a link between RFR emitted by wireless communication devices and COVID-19” is simply false. There is simply no evidence presented in this manuscript to support this conclusion. The report that there are common factors involved in COVID infection and RFR radiation does not in any way indicate a connection between the two diseases. Oxidative stress and immune system disfunction are characteristic of many diseases. The morphologic changes in red blood cells with RFR is not well documented, reported only in a meeting proceeding and a non-peer reviewed publication. Calcium is involved in every aspect of normal physiology and disease. Commonality of factors does not prove anything. You correctly state “This evidence presented here does not claim causation.” In that case why waste the effort to write the manuscript?

Reviewer #5: General Comment

The paper has a realistic basis but needs major revision

Specific Comments

Page 2, left column, last paragraph: Provide references for the technical information regarding 5GRight column, 2nd line, explain “gone live”.The term “wireless radiation” is not correct. Radiation is always wireless… Change to Wireless Communications radiation, or RF radiation or microwave radiation.The document by Payeras 2020 is not official. I understand the difficulty of formally publishing this, but should be referred to with reservation. Moreover, it is one author not two I think. The link in the reference does not work or it is inactivated. Provide another link to this document. Similar reservation for the Tsiang and Havas manuscript which is also not published yet. Any other references connecting Covid with 5G?Right column last paragraph, explain “areas of consolidation”, and CT scans (explain the initials).Page 3, left, lines 33-34: “cell phone antennas, base stations, Wi-Fi, and cell phones” correct to “mobile telephony base antennas, Wi-Fi, and mobile phones”, explain Wi-Fi.Right column, line 6, Should it be “100 times below” or “more than 1000 times below” 1 mW/cm2 ? Even western EMF-bioeffects literature shows adverse effects below 1 μW/cm2. For example cite: Magras and Xenos 1997 [RF Radiation-Induced Changes in the Prenatal Development of Mice. Bioelectromagnetics 18:455-461]. Provide references in this paragraph, and for the first two sentences of next paragraph (Russian research)Lines 46-48: “at non-thermal power densities (< 5 mW/cm2) and with particular emphasis on low power densities (<1mW/cm2)”. Above 1 mW/cm2 there can be thermal effects for frequencies 1-2 GHz. This is a very high power density. Change to: “at non-thermal power densities (< 1 mW/cm2)”.Use identical category tittles in the text (pages 5-9) and in Table 1.Explain ALL names with initials throughout the manuscript first time met (SARS, COVID, Wi-Fi, LTE, ROS, ACE-2, ARDS, ICU, etc.)Page 5, left, first paragraph. Rubik 2014 does not look like a peer reviewed paper. Please refer only to peer review publications, specifically for scientific findings.Page 7, right, lines 3-5. Delete the last sentence “Similar effects are observed in both that are caused by increased free radical formation and glutathione deficiency”.Line 11. Explain “spike protein”.In the effects on immune system cite and discuss the reviews by Szmigielski (2013) [Reaction of the immune system to low-level RF/MW exposures. Science of the Total Environment 454-455 (2013) 393-400], and Johansson 2009 [Disturbance of the immune system by electromagnetic fields-A potentially underlying cause for cellular damage and tissue repair reduction which could lead to disease and impairment. Pathophysioloy. 16(2-3):157-77].Page 8, left, lines 45-49. Pall (2013) made an observation that calcium channels play a major role in EMF bioeffects. A very similar observation was made long before by Walleczek (1992) [Electromagnetic field effects on cells of the immune system: The role of calcium Signaling. FASEB J, 6, 3177-85]. These were both review studies, not mechanisms. The mechanism for ion channel gating by EMFs is published by Panagopoulos et al (2002) [Mechanism of action of electromagnetic fields on cells. Biochemical and Biophysical Research Communications, 298(1), 95-102], and refers not only to calcium but to all cation channels. Please cite and discuss these studies as well.Same paragraph “viruses hijack calcium channels and increase intracellular Ca2+ (Chen et al., 2019)”. Ion channels are ion-specific by means of ion radius. Thus calcium channels would not allow larger molecules to pass through them such as viruses. Therefore the claim by Chen et al 2019 is likely impossible. This should be thoroughly searched and discussed.Right column, line 9. Explain “second messenger”Page 9, left, first paragraph. Explain “plaque instability”.Lines 26-27, “Potekhina et al. (1992) found that certain frequencies (55 GHz; 73 GHz) caused pronounced arrhythmia”. There are no physiological GHz frequencies in any living organism. All living functions are connected with ELF frequencies. It is thus unlikely that the GHz frequencies caused these effects. Instead the effects were most likely induced by the ELF pulsations. Similarly in Havas et al (2010). RF studies should report whether the field is pulsed/modulated or continuous wave. Most RF exposures contain ELF pulsing and/or modulation. This includes 2G-3G-4G and 5G as well. Please search the issue and revise accordingly.Right column, lines 29-31, “The bioeffects of RFR exposure are typically nonlinear rather than exhibiting the familiar linear dose-response effects from biochemical.” This is not true generally. The sporadic existence of “windows” does not render all effects non-linear. Effects depending on intensity or exposure time most usually are dose-dependent, and even close to linear. This should be revised.Page 10, left, lines 16-17, “However, these guidelines were established in 1996”. Provide reference.

Reviewer #6: Xu et al reported that in Feb. 2020, the cases fatality rate was much lower in Zhejiang province and other provinces were much lower than it was in Wuhan. (Xiao-Wei Xu, physician1, Xiao-Xin Wu, physician1, Xian-Gao Jiang, physician2, Kai-Jin Xu, physician1, Ling-Jun Ying, physician3, Chun-Lian Ma, physician4, Shi-Bo Li, physician5, Hua-Ying Wang, physician6, Sheng Zhang, physician7, Hai-Nv Gao, professor8, Ji-Fang Sheng, professor1, Hong-Liu Cai, physician1, Yun-Qing Qiu, professor1, Lan-Juan Li, professor1. Clinical findings in a group of patients infected with the 2019 novel coronavirus (SARS-Cov-2) outside of Wuhan, China: retrospective case series. BMJ 2020; 368 doi: https://doi.org/10.1136/bmj.m606).

The following citation should be used along with the Pakhomov et al citation with regard to millimeter wave effects: Betskii OV, Lebedeva NN. 2004 Low-intensity millimeter waves in biology and medicine. In:, Clinical Application of Bioelectromagnetic Medicine, Marcel Decker, New York, 2004, pp. 30-61. https://gabrielecripezzi.com/wp-content/uploads/2019/06/d75d92b7fb8f4d13ae5461e26afa62e87e60.pdf

On p.5, there is place which states (5G RFR) - this is confusing since 5G is not radiofrequency.

The author should state in several places in the paper, that the findings are suggestive of a link between EMF exposure and the severity of COVID-19 infections but none of these findings are considered to be proof of such a linkage.

The Sen et al, citation has an error in it. It should read NF-kB. and that is a Greek letter kappa. There may also be an error in the text.

The other thing I would suggest is that the author should make a suggestion or two about how the remaining uncertainty here might be resolved. I would make two suggestions that might help:

There could be one or more studies to determine whether some COVID-19 patients admitted to the hospital could be shielded in either Faraday cage or a shielded canopy could be put over the bed. These could lower exposures and hospitals are high EMF environments such with high powered Wi-Fi systems, many wireless communication devices and thousands of electronic devices, producing large amounts of dirty electricity. I know that hospitals have high levels of dirty electricity, having measured levels myself. The question being, whether such shielding would lower mortality rates and/or shorten times to patient release.

Another approach would be to measure home and working environments for EMF levels comparing patients with similar risk factor exposure but different severity of illness.

Reviewer #8: This paper delves into a facet of CoVid-19 and the evolving use of the widespread release of 5G into the environment, and the relationship between the two. This relationship has rarely been studied in the literature. That alone seems to make this paper relatively significant. It also provides evidence that the Precautionary Principle would call for more study before continuing the widespread deployment of 5G towers across the world, as well as the release of satellites that are envisioned to surround the planet. While bringing Internet access to the world’s population seems to be a good thing (in providing information to the societies that lack it now), we need to consider the Law of Unintended Consequences. The plans are for 5G to blanket the world, leaving no place without this new exposure to this Microwave and Milliwave radiation, and potentially effecting all life on earth, with the exception of beings that live in lead-lined shelters, or Faraday cages. Based on this paper alone, it indicates the need for further studies, before this is widely deployed. Especially when we are still learning about the effects of the CoVid-19 pandemic itself. The emerging understanding of how CoVid-19 has been linked to coagulation disorders, and to the hypoxic effect on the lung function of oxygenation absorption and release, via the red cells, adds additional urgency to the situation. As does the information on free radical creation causing havoc to Biological systems, the problems that either CoVid Or 5 G-type RFR can have on the heart, and especially how either can effect the immune system, and the paper points out that that could make a lot of illnesses deadlier. And all of these have never been fully tested together, but usually apart, by impartial researchers.

As this paper has reviewed the literature on the effects of RFR RadioFrequency Radiation, and considering how most studies in the past, looked at the heating effects of such radiation, and then said that if the RFR did not effect heating of biological tissue, it was labeled as safe. But as this paper shows, there is multiple evidence that there are adverse effects on tissue systems and their physiology, that have nothing to do with the heating effects of RFR. These low power high frequency effects have been collated, reviewed, and well documented in this paper, in a clearly collected and tabular form, making it easy to compare the RFR research, and the emerging data from the effects of the CoVid-19 virus. And by documenting the similarity of the recognized RFR effects on biological systems, then when these are clearly shown to be similar effects that are being discovered with the ever-changing documented effects of the CoVid-19 pandemic, the paper makes a strong case for a halt that is needed at this point in the pandemic, to step back and really investigate the nature and gravity of these effects, when combined, before further RFR 5 G potentially blankets the world.

As the paper points out in the discussion, while the connections and the evidence that both RFR and CoVid-19 attack similar biological systems and physiologies, the paper has not proven causation, but they clearly have proven the point that further, independent research is needed, and soon. Given that CoVid-19 has shown the ability to mutate, and future pandemics are predicted, this might be one “bright side” to this pandemic, in that it forces us to do this research, before it is too late. If indeed, all of the systems effected by CoVid-19 are also potentially weakened or effected by RFR like the upcoming wave of 5G (or 6 G, etc.) that is planned for the future, we have a chance to do the research that is needed to make our future safer, but to not do this research, and hide our heads in the sand, and ignore this paper, and its implications, then future generations may not look back kindly on our “rush to get faster internet to everyone.” Instead, we may do much more damage than when people thought that Radium watches that glow in the dark were “cool,” or that it was “cool” to use the Xray machines in shoe stores to see if your shoes fit the feet, by a live Xray that showed your foot skeleton in the shoe, while the Xray device was near the gonads. One wonders how many people died of those “cool” technologies, from cancer or other illnesses, before their dangers were finally recognized and removed from the market. How does the saying go: “Those who refuse to learn from the mistakes of the past, are condemned to repeat them,” only this time it may be on a world-wide scale? So I highly recommend that this paper be accepted for publication, so that it can stimulate much more research that must be done in this area. Reviewer #9: Dear Editor,

I completed review process of the manuscript numbered as “JCTRes-D-21-00034_reviewer”. Although the ideas put forward by the authors are not negligible, they are open issues to criticism. Because there is no scientific study that clearly reveals the relationship between RFRs, especially 5G and SARS-CoV-2. The authors have tried to make a good review, but the ideas they put forward show that only RFRs and SARS-CoV-2 have similar effects. Unfortunately, there is no scientific data about whether these similarities create a synergistic effect or not. So I suggest the authors change the title such as “Similarities in the effects of RFRs and SARS-CoV-2: Could there be a synergistic effect?” The decision is yours, dear editor. In summary, the article can be printed, but the title is very ambitious! On the other hand, I suggest to the authors to read the articles given below to find some hints for the topic of the manuscript.

Sincerely

Recommendation for Table 1 in page 4; Table 1 is not sufficient as the authors stated. Fort this reason, the authors have to present charasteristics of RFRs and name of the references. Therefore the table should be more informative for the readers for evaluation of the situation.

Barlas SB, Adalier N, Dasdag O, Dasdag S, Evaluation of SARS-CoV-2 with a biophysical perspective. Biotechnology & Biotechnological Equipment. 35:1, 392-406, 2021. DOI: 10.1080/13102818.2021.1885997Dasdag S, Akdag MZ, Celik MS (2008), Bioelectrical parameters of people exposed to radiofrequency in workplace and houses provided to workers. Biotechnology & Biotechnological Equipment. 22: 3: 859-863.Alkis ME, Akdag MZ, Dasdag S, E¡ects of Low-Intensity Microwave Radiation on Oxidant-Antioxidant Parameters and DNA Damage in the Liver of Rats. 2020 Bioelectromagnetics. 42:76—85, 2021. DOI:10.1002/bem.22315Dasdag S, Balci K, Celik MS, Batun S, Kaplan A, Bolaman Z, Tekes S, Akdag Z (1992), Neurologic and biochemical findings and CD4 / CD8 ratio in people occupationally exposed to RF and microwave. Biotechnol. & Biotechnol. Eq. 6 / 4, 37 -39.Yilmaz F, Dasdag S, Akdag MZ, Kilinc N (2008).Whole body exposure of radiation emitted from 900 MHz mobile phones does not seem to affect the levels of anti-apoptotic BCL-2 protein. Electromagnetic Biology and Medicine. 27: 1; 65-72.Dasdag S, Akdag MZ, Ulukaya E (2009), Effects of Mobile Phone Exposure on Apoptotic Glial Cells and Status of Oxidative Stress in Rat Brain. Electromagnetic Biology and Medicine. 28: 4; 342-354.Dasdag S, Bilgin HM, Akdag MZ, et al. (2008), Effect of Long Term Mobile Phone Exposure on Oxidative and Antioxidative Process and Nitric Oxide in Rats. Biotechnology & Biotechnological Equipment. 22: 4; 992-997Alkis ME, Bilgin HM, Akpolat V, Dasdag S, Yegin K, Yavas MC, Akdag MZ, Effect of 900-, 1800-, and 2100-MHz radiofrequency radiation on DNA and oxidative stress in brain. Electromagn Biol Med. 38(1): 32-47, 2019.Akdag M, Dasdag S, Canturk F, Akdag MZ, Exposure to non-ionizing electromagnetic fields emitted from mobile phones induced DNA damage in human ear canal hair follicle cells. Electromagn Biol Med. 2018, 37 (2): 66-75. https://doi.org/10.1080/15368378.2018.1463246Bektas H, Dasdag S, Effect of Radiofrequencies Emitted from Mobile Phones and Wi-Fİ on Pregnancy. Journal of International Dental and Medical Research. 10(3): 1084-1095, 2017Bektas H, Dasdag S, Bektas S, Comparison of effects of 2.4 GHz Wi-Fi and mobile exposure on human placenta and cord Blood. Biotechnology & Biotechnological Equipment. 2020, VOL. 34 (1): 154-162, 2020, https://doi.org/10.1080/13102818.2020.1725639

Reviewer #10: - Background section. Page 1, lines 49-52. “…that will dramatically increase the population’s wireless radiation exposure both inside structures and outdoors”.

Please cite evidence (referenced studies) supporting this hypothesis by specific models and/or real-time measurements.

- Background section. Page 1. “During the first wave in the United States, COVID-19 attributed cases and deaths were higher in states with 5G infrastructure compared with states that did not yet have this technology (Tsiang and Havas, manuscript submitted)”. Unpublished/not available data should not be considered as citation.- Overview on covid-19 (page 2). In consideration of the main aim of the review, this paragraph can be significantly shortened.- Authors should better describe the main technical characteristics of the 5G infrastructures (i.e, small cells, MIMO, multiple frequencies etc.), briefly listing the main technical differences with the previous radiofrequency networks.- Table 1. Authors should indicate in the table the most relevant and specific reference(s) for each listed point.- Table 1. it is not clear if the cited effects (“RFR exposure bioeffects”) have been generically linked with high frequency electromagnetic fields or, specifically, with 5G frequencies.Authors should include in the “RFR exposure bioeffects” listed in this table the frequency, the level (i.e., power density) and the period of exposure linked with each of the cited effects. Authors should also specify the type of study (i.e., in vitro, animal or human study).- Authors should report the average level of RFR exposure measured in at least some geographical areas implementing 5G infrastructure. A comparison of the “real-life” level of exposure with the RFR exposure levels generating the majority of bioeffects described in the paper is needed.- Several bioeffects described by authors are generated by levels of exposure significantly higher than those generally recorded in urban areas. Authors should include in the paper a new table listing the bioeffects potentially linked with covid-19 and observed in the presence of levels of environmental exposure comparable with those recorded in the most exposed urban areas.- Authors should report and comment previous studies, if available, linking RFR exposure with viral diseases different from Covid-19.- Authors discuss evidence deriving from exposure to cell phones to support possible effects of environmental exposure to 5G. However, exposure to cell phones or to 5G infrastructure (i.e., base stations, MIMO antennas, devices etc.) may significantly differ in terms of SAR and are not fully comparable.- The majority of the bio-effects described by Authors could also be, at least theoretically, attributed to pre-existing radiofrequency exposure, in particular in highly exposed geographical areas. Furthermore, in the short-medium term, in exposed areas the level of RFR exposure can be assumed to be constant. On the other hand, covid-19 incidence, morbidity and mortality significantly varied during the last year. The lack of a parallel trend should limit the hypothesis of a direct link between 5G exposure and covid-19 clinical and epidemiological aspects.- The majority of the bio-effects described by Authors could also be attributed to other sources of environmental pollution and, in particular, to air pollution. The effect of this and other relevant confounders in urban areas characterized by high population density is not discussed by Authors.- According to some evidence, children can be particularly vulnerable to RFR effects. However, the pediatric age seems to be the less involved, at least in terms of clinical manifestation, by the covid-19 pandemic. How Authors could explain this different outcome in different age classes equally exposed to RFR?- Page 9, discussion section. “The evidence indicates that RFR may weaken the host, exacerbate COVID-19 disease, and thereby worsen the pandemic”. In the opinion of this reviewer, the reported evidence only indicate that mechanisms possibly involved in the clinical progression of SARS-CoV-2 could be also generated, according to experimental data, by RFR exposure. It is still under debate, however, if these biological effects can be present in the case of frequencies and levels of RFR exposure commonly found in the urban areas where 5G networks have been implemented.- Point of strength and limitations of the review performed by authors should be clearly stated.

Reviewer #11: In this work authors summarize current state of knowledge about the harmful effects of Radiofrequency radiation (RFR), with special focus on those that could possibly enhance the possibility of being infected by COVID-19.

This review is a nice mix of very recent publications (last 5 years) and some classic papers mainly from Soviet Union and United States, showing, that knowledge about harmful effects of RFR have been extensively studied already few decades ago, what is very important mainly because of the number of conspiracy theories available of the Internet about 5G RFR. Call for the scientific evaluation of this exposure type is more than proper. Otherwise we will be wittnesses of very big population study, that will reveal the truth in the future times.

I do not have any big questions rather some comments:

1) In the Introduction part authors cite quite few review articles. I would suggest citation of the few of the best experimental articles considering this type of deleterious effect (e.g. oxidative stress, reproductive damage), since the number of the reviews considering RFR is relatively high, but actual experimental studies that strongly supports the conclusion of reviews are sometimes hard to find or other times not so conclusive.2) Authors stated that oxidative stress induced by RFR may exacerbate the seriousness of COVID -19 disease. I agree that induction of oxidative stress is the most common harmful effect observed after exposure to RFR, targeting mostly the cells with the high level of metabolism, like sperm cells. But in a lot of studies induction of ROS after exposure is not higher than 50% of control values and some studies have even seen adaptation to the radiation with increased exposure time on the cellular level.3) Authors suggest that 5G introduction in the cities that have been hit with the COVID-19 very hard in the first wave could lead to the increased mortality and number of cases. Since there are some connections, this could be also explaine by the fact that Northern Italy is the region with the highest percentage of elderly people that often have other comorbidities like diabetes and hypertension that significntly increase the probability of serious condition, next the precautions in the Italy have not been sufficient what is more likely the cause of such strong hit by COVID than 5G. To the New York, that is one of the crowdest cities in the world and social distancing have not been established soon enough.4) Other argument is that in the second wave midlle europe (Czech, Hungary, Poland, Slovakia) have been hit very hard with COVID and 5G is still not introduced in this countries (maybe only capital cities, but certainly not smaller cities and villages, that had been hit even harder). So I think overally, mobility of people, meeting of families during holidays and inproper precautions have much larger effect on the pandemia than RFR exposure. But on the other hand I agree that RFR could add some stress to individuals already weakened by the COVID.5) Authors should also focus on the fact that lot of experimental studies did not provide any evidence of harmful effect of RFR (and not all of them are industry financed and ordered). Other problem with experimental evidence of harmful effects is the reproducibility of the observed effects and replicability of the studies, that are often performed with questionable devices, under not precisely characterized exposure conditions.

Despite this comments bit of article provide nice review of the RFR effects on the human beings supported by the number of peer-reviewed studies and also overview of COVID- 19 disease, that is very valuable and worth publishing after applying some of the comments to the manuscript.

Reviewer #12 (editor-in-chief): SO THESE SUGGESTIONS ARE MANDATORY TO ADDRESS

1) Please contextualize the narrative to centers/regions of outbreak where 5G is not prevalent, such as rural India, beyond the premise that correlation is not causation. Cite regions that had a 5G rollout but were not hit by the pandemic, and please provide explanations for such exemptions.2) What is the average power density (mW/cm2) of 5G RFR in Wuhan, and how does this compare to the cities that have harbored 5G but with low manifestation of COVID-19?3) Your paper is hypothetical, so please remain in this hypothetical framework throughout the manuscript. Phrases such as “This is the first scientific paper documenting a link between RFR emitted by wireless communication devices and COVID-19” are unwarranted. Although your paper provides argumentation in favor of this hypothesis, it does not establish a link (cause-effect) between 5G and the incidence of COVID-19. Please nuance this statement and other similar statements in the text.4) Please unify all units of power density throughout the manuscript to conform to the standard unit used in the US (mW/cm2). The text is inconsistent with the nomenclature, where sometimes the unit is abbreviated while in other instances the unit is written out. It is advisable to consistently abbreviate to mW/cm2. That makes it easier for the readers to contextualize research results with the upheld norm for RFR exposure.5) Please include a paragraph where you attempt to introduce gaps/flaws in your hypotheses. One of the main ingredients of such a paragraph would be to point out to readers that in many studies the power densities used to study biological effects exceeded the maximum level of 1 mW/cm2. Note all other aspects of the setup and execution of cited experimental studies that deviate from the manner in which 5G RFR is reduced to practice in Wuhan and elsewhere. Such a paragraph helps put the narrative into complete perspective.

Author’s response

Reviewers’ comments:

Reviewer #1: General comments

The author has considered very important and actual issue putting forward the question whether 5G mobile communication may contribute to the Covid-19 pandemia. To answer this question the author considers several effects in common between Covid patients and RF effects. These effects are summarized in the Table and described in the text referring to over 250 papers retrieved by the author from MEDLINE. However, this consideration has some limitations, which make the conclusions of the author rather immature.

First, the author compares the data on Covid patiens with the data on RF effects, which were obtained in vitro, animal, and human studies. As far as author has considered only about 250 papers from more than 30000 published papers on RF effects, it seems to be reasonable to retrieve for comparison mostly human RF studies.

Second, there is significant number of RF studies where no effects mentioned in the Table were observed. The author considers the RF effects in general, regardless dependences on frequencies, intensities and other key parameters, which were shown to be critical for the RF effects and have been most comprehensively reviewed in the IARC monograph (IARC 2013). Indeed, the author acknowledges these complicated dependences in the Discussion. Nevertheless, statement is made “that RFR and, in particular, 5G, which involves densification of 4G, has exacerbated the COVID-19 pandemic by weakening host immunity and increasing SARS-CoV-2 virulence”. However, this statement would demand consideration of RF effects at signals (i.e. frequency, modulation,…) and intensities as users of 5G are exposed to. As far as 5G is in focus, technical description of 5G signals is needed and studies of effects of RF signals with the same or similar characteristics should be retrieved and reviewed.

In conclusion, while the effects of RF and changes observed in Covid patients seem to overlap at some specific conditions of exposure, and given that the RF effects depend on number of physical and biological variables, more stringent retrieval of RF studies is needed to discuss possible connection of 5G exposure and Covid pandemia. Otherwise, this connection should be significantly down played and the text should be revised accordingly.

***Thank you for your observations. Regarding your first point, it is true that there are perhaps 30,000 or even more scientific papers documenting the bioeffects of RFR on living systems. In choosing studies to review and reference for our paper, we discovered that controlled RFR exposure studies on human subjects alone were insufficient for this review, as most of them, in English, were conducted as short-term studies. In this paper, we are concerned mainly on long-term health effects from chronic RFR exposure. Because there are very few long-term studies on humans, aside from occupational studies, which we did include in our literature review, it was essential that we expand our literature search to include controlled, mostly long-term animal and cell studies*.**

***Regarding your second point, there are indeed a significant number of RFR exposure studies where no observed effects listed in our Table were found. We pointed out in the paper that the published literature not only contains some reports with contradictory results, but a bias is clearly evident in that studies conducted or sponsored by the industry generally tend to conclude negative results, while studies conducted by independent scientists, in general, tend to uncover adverse bioeffects. This bias was reported in a systematic review [Huss, A., M. Egger, K. Hug, K. Huwiler-Muntener, M. Roosli. 2007. Source of funding and results of studies of health effects of mobile phone use: systematic review of experimental studies. Environmental Health Perspectives, 115 (1): 14. DOI: 10.1289/ehp.9149]. We had already cited and discussed this paper and its ramifications in our Discussion Section. It is because of this documented bias that we did not make a systematic effort to include negative studies in our paper. Instead, we sought research papers that supported our hypothesis*.**

**
*Next, reviewer #1 wrote, “The author considers the RF effects in general, regardless of dependences on frequencies, intensities and other key parameters, which were shown to be critical for the RF effects and have been most comprehensively reviewed in the IARC monograph (IARC 2013).”*
**

**
*In the draft of our paper that you reviewed, we stated the following in our Methods Section regarding our selection of papers for review: “This included the world literature in English and Russian reports translated to English, on RFR from 600 MHz – 90 GHz, the spectrum of wireless communication radiation (2G – 5G inclusive), with particular emphasis on non-thermal, low power densities (< 1 mW/cm^2^) and long term exposures.”*
**

**
*Reviewer #1 wrote, “As far as 5G is in focus, technical description of 5G signals is needed and studies of effects of RF signals with the same or similar characteristics should be retrieved and reviewed.”*
**

***As we mentioned above, we selected papers testing exposure to frequencies from 600 MHz – 90 GHz, which comprise the spectrum of wireless communication radiation from 2G to 5G, inclusive. We provided a more technical description of 5G on page 2, as follows, and cited the official technical document on 5G*.**

***“5G is a protocol that will use high frequency bands of the electromagnetic spectrum in the vast radiofrequency range from 600 MHz to nearly 100 GHz, which includes millimeter waves (>20 GHz), in addition to the currently used 3G (third generation) and 4G (fourth generation) long term evolution (LTE) microwave bands. 5G frequency spectrum allocations differ from country to country. Focused pulsed beams of radiation will emit from new base stations and phased array antennas placed close to buildings whenever persons access the 5G network. Because these high frequencies are strongly absorbed by the atmosphere and especially during rain, a transmitter’s range is limited to 300 meters. Therefore 5G involves base stations and antennas much more closely spaced than previous generations, plus satellites in orbit that will emit 5G bands globally to create a wireless worldwide web. The system requires significant densification of 4G as well as new 5G antennas that may dramatically increase the population’s wireless communications radiation exposure both inside structures and outdoors. In addition, up to 100,000 emitting satellites are planned to be launched into orbit. This infrastructure will significantly alter the world’s electromagnetic environment to unprecedented levels and may cause unknown consequences to the entire biosphere, including humans. The new infrastructure will service the new 5G devices, including 5G mobile phones, routers, computers, tablets, self-driving vehicles, machine-to-machine communications, and the Internet of Things (IoT)*.**

**
*The global industry standard for 5G is set by the 3rd Generation Partnership Project (3GPP), which is an umbrella term for several organizations developing standard protocols for mobile telecommunications. The 5G standard specifies all key aspects of the technology, including frequency spectrum allocation, beam-forming, beam steering, multiplexing MIMO (multiple in, multiple out) schemes to nearly simultaneously serve a large number of devices within a cell as well as modulation schemes among many others. The latest finalized 5G standard, Release 16, is codified in the 3GPP published Technical Report TR 21.916 and may be downloaded from the 3GPP server at https://www.3gpp.org/ftp/Specs/archive/21_series/21.916/ (3GGP, 2020).”*
**

***We had already pointed out in the Discussion Section that no health or safety studies have been published specifically on 5G signals in the manner in which people will experience them in the real world: 5 to 10 frequency bands of 5G, along with 5 to 10 frequency bands of 4G, which are necessary for 5G to work. We included a statement about the lack of controlled studies on health effects from real-world wireless communication exposures in our Discussion Section, and we also strongly recommended that such studies need to be done*.**

**
*We “downplayed” our statements on the connection between RFR exposure bioeffects and COVID-19, as you request, throughout the manuscript. We re-wrote our conclusion to state, “There is a substantial overlap in pathobiology between COVID-19 and RFR exposure. The evidence presented here indicates that mechanisms involved in the clinical progression of COVID-19 could also be generated, according to experimental data, by RFR exposure. We propose a link between adverse bioeffects of RFR exposure from wireless communication devices and COVID-19.”*
**

Specific comment

The author referred to the ICNIRP 2009 guidelines, which are outdated as far as ICNIRP has recently updated them. It should also be stated that in contrary to the ICNIRP thermally based guidelines, other international bodies such as EMF Working Group of the European Academy of Environmental Medicine has suggested much lower guidelines taking into account no-thermal RF effects reported in multiple studies (Belyaev, Dean et al. 2016).

Belyaev, I., A. Dean, H. Eger, G. Hubmann, R. Jandrisovits, M. Kern, M. Kundi, H. Moshammer, P. Lercher, K. Muller, G. Oberfeld, P. Ohnsorge, P. Pelzmann, C. Scheingraber and R. Thill (2016). “EUROPAEM EMF Guideline 2016 for the prevention, diagnosis and treatment of EMF-related health problems and illnesses.” Rev Environ Health 31(3): 363-397.

IARC (2013). IARC Monographs on the Evaluation of Carcinogenic Risks to Humans. Non-ionizing Radiation, Part 2: Radiofrequency Electromagnetic Fields Lyon, France, IARC Press.

**
*Thank you for this helpful comment. We appreciate learning from you that ICNIRP updated the RFR exposure guidelines in 2020. We removed our old citation and reference and updated our citation and reference as follows:*
**

***International Commission on Non-Ionizing Radiation Protection (ICNIRP), 2020. Guidelines for Limiting Exposure to Electromagnetic Fields (100 kHz to 300 GHz), Health Physics: May 2020 – 118(5): 483-524*.**

**
*doi: 10.1097/HP.0000000000001210*
**

***We are also grateful to learn from you that there are other international bodies working to reduce the exposure guidelines. We have added the following sentence to the Discussion Section of our paper and utilized the reference (Belyaev, 2016) that you provided; thank you*.**

**
*“Recently other international bodies such as the EMF Working Group of the European Academy of Environmental Medicine, have proposed much lower guidelines, taking into account non-thermal bioeffects from RFR exposure reported in multiple sources (Belyaev et al., 2016).”*
**

Reviewer #3: This is a well-researched paper, with valuable insights into adverse effects from non-ionizing athermal radiation. However, the authors need to sharpen their language to clarify what has been demonstrated, and have the conclusions be fully reflective of what the data have shown. For example, on p.9 they state: “This evidence presented here does not claim causation.” Yet, on p.10 they state: “We conclude that RFR and, in particular, 5G, which involves densification of 4G, has exacerbated the COVID-19 pandemic by…..”. Later in the same paragraph, they state: “In short, wireless communication radiation is a ubiquitous environmental stressor, and evidence presented here suggests that it is a contributing factor in the COVID-19 pandemic.”

**
*We have modified the text on page 9 as follows:*
**

**
*“This evidence presented here suggests a link between EMF exposure and the severity of COVID-19 infection, but none of these observations are considered to be proof of such a linkage. Specifically, the evidence does not confirm causation.”*
**

**
*Additionally, our Conclusion Section has been changed as follows:*
**

**
*“There is a substantial overlap in pathobiology between COVID-19 and RFR exposure. The evidence presented here indicates that mechanisms involved in the clinical progression of COVID-19 could also be generated, according to experimental data, by RFR exposure. We propose a link between adverse bioeffects of RFR exposure from wireless communication devices and COVID-19.”*
**

It appears to me they have shown the following type of relationships, as stated on p.7: “In summary, oxidative stress is a major component in the pathophysiology of COVID-19 as well as in cellular damage caused by RFR exposure. Similar effects are observed in both that are caused by increased free radical formation and glutathione deficiency.” Also, as stated on p.8: “In short, COVID-19 can lead to immune dysregulation as well as cytokine storm. By comparison, exposure to low-level RFR as observed in animal studies can also compromise the immune system, with chronic daily exposure producing immunosuppression or immune dysregulation including hyperactivation.”

They have shown quite convincingly that RFR produces a number of damaging bioeffects, and many of these damaging bioeffects are seen in COVID-19 patients. That constitutes a potential indirect linkage between RFR and COVID-19, and laboratory tests will be required to show whether a direct linkage exists. It seems to me that’s how their results need to be presented. The concepts behind the discipline of Literature-Related Discovery would strengthen the arguments for these types of linkages. Minor wording changes are all that are required in order to eliminate any confusion on what has been demonstrated.

**
*Thank you for pointing out that the discipline of Literature-Related Discovery is relevant to our approach. We added the following to the Methods Section:*
**

**
*“Our approach is akin to Literature-Related Discovery, in which two concepts that have heretofore not been linked are explored in literature searches to look for linkage(s) in order to produce novel, interesting, plausible, and intelligible knowledge, i.e., potential discovery (Kostoff et al., 2007).”*
**

**
*Reference: Kostoff RN, Block JA, Solka JL, Briggs MB, Rushenberg RL, Stump JA, Johnson D, Lyons TJ, Wyatt JR. 2007. Literature-Related Discovery: A Review. Report to the Office of Naval Research, 2007, pp. 1-58. https://ia801006.us.archive.org/4/items/DTIC_ADA473643/DTIC_ADA473643.pdf*
**

Additionally, the authors need to sharpen their usage of the term COVID-19. It is a disease, and is not causing anything, as the authors imply. It is associated with a number of abnormal biomarkers, and this distinction needs to be delineated.

Reviewer #4: This manuscript is totally lacking in scientific value. The statement “This is the first scientific paper documenting a link between RFR emitted by wireless communication devices and COVID-19” is simply false. There is simply no evidence presented in this manuscript to support this conclusion. The report that there are common factors involved in COVID infection and RFR radiation does not in any way indicate a connection between the two diseases. Oxidative stress and immune system disfunction are characteristic of many diseases. The morphologic changes in red blood cells with RFR is not well documented, reported only in a meeting proceeding and a non-peer reviewed publication. Calcium is involved in every aspect of normal physiology and disease. Commonality of factors does not prove anything. You correctly state “This evidence presented here does not claim causation.” In that case why waste the effort to write the manuscript?

***The claim that this paper is “totally lacking in scientific value” is contrary to nine other reviews of this paper, that indicate that the paper is dealing with a “very important” issue; “has a realistic basis;” and is “studying a relationship that has rarely been studied in the literature…which makes the paper significant.” These are a sample of the positive remarks made by the other reviewers, who also provided constructive criticism that we implemented to improve the paper*.**

**
*We removed the sentence from the paper, “This is the first scientific paper documenting a link between RFR emitted by wireless communication devices and COVID-19.”*
**

***While we appreciate that you would prefer all references to be peer-reviewed, we would like to cite certain important papers that have not been peer-reviewed. At this critical time during the pandemic, many manuscripts that are not peer-reviewed are being cited in professional journal papers on COVID-19 to help experts bring forth knowledge as quickly as possible to facilitate an end to human suffering and death. In this particular case, we maintain that it is appropriate to cite the work of morphological changes in red blood cells that relate to blood clotting, especially since SARS-CoV-2 and its spike protein have been shown to be thrombogenic and can directly bind to ACE2 receptors on platelets (Zhang et al., 2020). Even when isolated, the spike protein has been shown to cause endothelial injury (Lei et al., 2021)*.**

**
*We modified the paragraph on the blood changes associated with SARS-CoV-2 infection to read as follows:*
**

***“Endothelial damage may occur from spike protein engagement with ACE2 receptors lining the blood vessels, even when isolated and removed from its vital RNA (Lei et al., 2021). Rouleaux formation, particularly in the setting of underlying endothelial damage, can clog the microcirculation, impeding oxygen transport, contributing to hypoxia, and increasing the risk of thrombosis (Wagner et al., 2013)*.**

**
*Thrombogenesis associated with SARS-CoV-2 infection may also be caused by direct binding of the virus to ACE2 receptors on platelets (Zhang et al., 2020).”*
**

***Furthermore, this particular research on morphological changes in red blood cells due to cell phone radiation exposure was conducted by one of us (Rubik) who has 25+ years of experience in live blood microscopy. So, although the morphological changes in red blood cells are not as well supported in this paper as we would like, we hope to plant seeds for future research to explore this phenomenon*.**

**
*Therefore, we modified the paragraph as follows:*
**

**
*“Although not peer reviewed, one of us (Rubik) investigated the effect of 4G LTE (fourth generation long-term evolution) mobile phone radiation on the peripheral blood of ten human subjects each of whom had been exposed to cell phone radiation for two consecutive 45 minute intervals (Rubik, 2014). Two types of effects were observed: initially increased stickiness of peripheral red blood cells with rouleaux formation and subsequently formation of echinocytes (spiky red blood cells.) Red blood cell clumping and aggregation are known to be actively involved in blood clotting (Wager et al., 2013). The prevalence of such blood changes upon exposure to RFR in the human population has not yet been determined. Larger controlled studies should be performed to further investigate this phenomenon.”*
**

***As we wrote in the manuscript, according to the CDC, the epidemiological triad – the agent (virus in this case), the health of the host, and the environment, is a useful model to explain how an environmental cofactor may contribute to any disease. We point out that environmental toxins in general, and RFR in particular, may have exacerbated the pandemic and have not been sufficiently explored or addressed. From our extensive knowledge of the literature on the adverse health effects of wireless communication radiation, we saw similarities and possible connections between the adverse health effects from RFR and COVID-19 manifestations that we discuss in this manuscript. We certainly agree with you that “immune system dysfunction and oxidative stress are nonspecific conditions and characteristic of many diseases.” However, we do not believe this diminishes our argument, that because these disease states are both encountered with COVID-19 as well as with chronic wireless communication radiation exposure, it is possible that the bioeffects related to chronic exposure to wireless communication radiation may have exacerbated the disease COVID-19. We hope that our manuscript will facilitate further research, and that it might also encourage consideration of environmental factors and public health measures to help mitigate the pandemic and protect human health*.**

Reviewer #5: General Comment

The paper has a realistic basis but needs major revision

Specific Comments

Page 2, left column, last paragraph: Provide references for the technical information regarding 5G

**
*We provided more technical detail on 5G, including the official technical document specifying 5G as our reference (3GPP, 2020). We re-wrote and expanded upon 5G on page 2 as follows:*
**

***“5G is a protocol that will use high frequency bands of the electromagnetic spectrum in the vast radiofrequency range from 600 MHz to nearly 100 GHz, which includes millimeter waves (>20 GHz), in addition to the currently used 3G (third generation) and 4G (fourth generation) long term evolution (LTE) microwave bands. 5G frequency spectrum allocations differ from country to country. Focused pulsed beams of radiation will emit from new base stations and phased array antennas placed close to buildings whenever persons access the 5G network. Because these high frequencies are strongly absorbed by the atmosphere and especially during rain, a transmitter’s range is limited to 300 meters. Therefore 5G involves base stations and antennas much more closely spaced than previous generations, plus satellites in orbit that will emit 5G bands globally to create a wireless worldwide web. The system requires significant densification of 4G as well as new 5G antennas that may dramatically increase the population’s wireless communications radiation exposure both inside structures and outdoors. In addition, up to 100,000 emitting satellites are planned to be launched into orbit. This infrastructure will significantly alter the world’s electromagnetic environment to unprecedented levels and may cause unknown consequences to the entire biosphere, including humans. The new infrastructure will service the new 5G devices, including 5G mobile phones, routers, computers, tablets, self-driving vehicles, machine-to-machine communications, and the Internet of Things (IoT)*.**

**
*The global industry standard for 5G is set by the 3rd Generation Partnership Project (3GPP), which is an umbrella term for several organizations developing standard protocols for mobile telecommunications. The 5G standard specifies all key aspects of the technology, including frequency spectrum allocation, beam-forming, beam steering, multiplexing MIMO (multiple in, multiple out) schemes to nearly simultaneously serve a large number of devices within a cell as well as modulation schemes among many others. The latest finalized 5G standard, Release 16, is codified in the 3GPP published Technical Report TR 21.916 and may be downloaded from the 3GPP server at https://www.3gpp.org/ftp/Specs/archive/21_series/21.916/ (3GGP, 2020).”*
**

**Reference: 3GPP (Third Generation Partnership Project, 2020. Technical Report TR 21.916, V1.0.0. (2020-12), pages 1-149.**
***https://www.3gpp.org/ftp/Specs/archive/21_series/21.916/***

2. Right column, 2nd line, explain “gone live”.

***The term “gone live” has been clarified as follows, thank you*.**

**
*“COVID-19 began in Wuhan, China in December 2019, shortly after city-wide 5G had “gone live”, i.e. become a fully operational system, on October 31, 2019.”*
**

3. The term “wireless radiation” is not correct. Radiation is always wireless… Change to Wireless Communications radiation, or RF radiation or microwave radiation.

***We changed it throughout the document and title of the manuscript to “wireless communication radiation,” thank you*.**

4. The document by Payeras 2020 is not official. I understand the difficulty of formally publishing this, but should be referred to with reservation. Moreover, it is one author not two I think. The link in the reference does not work or it is inactivated. Provide another link to this document. Similar reservation for the Tsiang and Havas manuscript which is also not published yet. Any other references connecting Covid with 5G?

**
*We removed the old text and reference from Payeras (2020) whose link no longer works. We found a new link to his updated and expanded paper posted online here: Bartomeu Payeras i Cifre is indeed a single author as you indicate. We now refer to his updated and greatly expanded paper, which also has a revised title. http://www.untumbes.edu.pe/vcs/biblioteca/document/varioslibros/0567.%20Estudio%20sobre%20la%20asim%C3%A9trica%20distribuci%C3%B3n%20de%20casos%20de%20COVID-19%20y%20su%20relaci%C3%B3n%20con%20la%20tecnolog%C3%ADa%205G.pdf*
**

***This is a more extensive paper with 81 pages compared to the previous version that was 21 pages. Although it is not “official” or peer-reviewed, we prefer to include it, too, because it is relevant to our thesis. Moreover, we are dealing with a pandemic, and many COVID-19 researchers are citing non-peer-reviewed papers to present evidence as soon as it is available that might help us understand more and/or mitigate the pandemic. We have also stated in our revised manuscript that this paper is not peer-reviewed. In addition, we added a peer-reviewed publication from Mordachev (2020). The reference of Tsiang and Havas has now been published in a peer-reviewed journal. This paper analyzes and compares the incidence of COVID-19 as well as the mortality rates in the United States as well as in US cities with and without 5G. The updated citation and reference is included*.**

**
*The text was modified as follows: “During the first pandemic wave in the United States, COVID-19 attributed cases and deaths were statistically higher in states and major cities with 5G infrastructure as compared with states and cities that did not yet have this technology (Tsiang and Havas, 2021).”*
**

5. Right column last paragraph, explain “areas of consolidation”, and CT scans (explain the initials).

***The word “consolidation” was removed and replaced with “airspace opacification” and the initials “CT” (computed tomography) have been explained accordingly, thank you*.**

**
*“Massive oxidative damage to the lungs has been observed in areas of airspace opacification documented on chest radiographs and computed tomography (CT) scans in patients with COVID-19 pneumonia (Cecchini and Cecchini, 2020).”*
**

6. Page 3, left, lines 33-34: “cell phone antennas, base stations, Wi-Fi, and cell phones” correct to “mobile telephony base antennas, Wi-Fi, and mobile phones”, explain Wi-Fi.

***The terms cell phone antennas, base stations, Wi-Fi, and cell phones have been changed accordingly and the term Wi-Fi explained accordingly, thank you. Wi-Fi is a trademarked name, and contrary to popular thought, it does not mean “Wireless Fidelity.” Instead, it refers to “IEEE 802.11b Direct Sequence,” (IEEE is Institute of Electrical and Electronic Engineers) type of local area network (LAN). Here*** is ***a website that explains it in detail: https://www.tanaza.com/tanazaclassic/blog/wi-fi-not-mean-wireless-fidelity/***

**
*Here is how we modified the text:*
**

***“Organisms are electrochemical beings, and low-level RFR from wireless communication devices, including mobile telephony base antennas, wireless network protocols utilized for the local networking of devices and internet access, trademarked as Wi-Fi (officially IEEE 802.11b Direct Sequence, where IEEE is Institute of Electrical and Electronic Engineers)*
*by the Wi-Fi alliance, and mobile phones, among others, may disrupt regulation of numerous physiological functions.”***

7. Right column, line 6, Should it be “100 times below” or “more than 1000 times below” 1 mW/cm2 ?

***We changed it to 1000 times below 1 mW/cm^2^*.**

**
*“The Soviet and Eastern European literature from 1960-70s demonstrates significant biological effects, even at exposure levels more than 1000 times below 1 mW/cm^2^, the current guideline for maximum public exposure in the US.”*
**

Even western EMF-bioeffects literature shows adverse effects below 1 μW/cm2. For example cite: Magras and Xenos 1997 [RF Radiation-Induced Changes in the

Prenatal Development of Mice. Bioelectromagnetics 18:455-461]. Provide references in this paragraph, and for the first two sentences of next paragraph (Russian research)

***We added the following to the manuscript in the section on the overview of health effects of RFR, which included the reference you provided (Magras and Xenos, 1997) as well as citations and references to reports by Adendano et al., 2012; Bucher and Eger, 2012; Navarro et al., 2003; and Hutter et al., 2006*.**

**
*“Adverse bioeffects from EMF exposure levels below 0.001mW/cm^2^ have also been documented in the Western literature. Damage to human sperm viability including DNA fragmentation by internet-connected laptop computers at power densities from 0.0005 – 0.001 mW/cm^2^ has been reported (Avendano et al., 2012). Chronic human exposure to 0.000006 to 0.00001 mW/cm2 produced significant changes in human stress hormones following a mobile phone base station installation (Bucher and Eger, 2012). Human exposures to cell phone radiation at 0.00001 – 0.00005 mW/cm^2^ resulted in complaints of headache, neurological problems, sleep problems, and concentration problems, corresponding to “microwave sickness” (Navarro et al., 2003; Hutter et al., 2006). The effects of RFR on prenatal development in mice placed near a mobile phone “antenna park” exposed to power densities from 0.000168-0.001053 mW/cm2 showed a progressive decrease in the number of newborns and ended in irreversible infertility (Magras and Xenos 1997). However, most US research has been performed over short durations of weeks or less. There have been few long-term studies on animals or humans.”*
**

**
*We also added the following to the manuscript on the paragraph on Russian research:*
**

***“A wide variety of bioeffects from exposure to nonthermal levels of RWR were reported by Soviet research groups since the 1960s…*.**

***Several notable Russian studies are as follows. Studies on E. coli bacteria cultures show power density windows for microwave resonance effects for 51.755 GHz stimulation of bacterial growth, observed at extremely low power densities of 10E-13mW/cm^2^ (Belyaev et al., 1996)*. *Recent Russian studies confirm earlier results of Soviet research groups on the effects of 2.45 GHz at 0.5mW/cm^2^ on rats (30 days exposure for 7 hrs/day), with the formation of antibodies to the brain (autoimmune response) and stress reactions (Grigoriev et al., 2010). In a long-term (1 to 4 year) study on children who use mobile phones compared to a control group, functional changes, including greater fatigue, decreased voluntary attention, and weakening of semantic memory, among other adverse psychophysiological changes, were reported (Grigoriev, 2012). The key Russian research reports that underlie the scientific basis for Soviet and Russian RFR exposure guidelines to protect the public, which are much lower than in the US, have been summarized (Repacholi et al., 2012).”***

**
*These references on the Russian/Soviet papers have been added to the manuscript:*
**

***Y. Belyaev, V. S. Shcheglov, Y. D. Alipov, and V. A. Polunin, “Resonance effect of millimeter waves in the power range from 10(-19) to 3 x 10(-3) W/cm2 on Escherichia coli cells at different concentrations,” Bioelectromagnetics, vol. 17, pp. 312-321, 1996*.**

***Grigoriev, Y.G., Grigoriev, O.A., Ivanov, A.A., Lyaginskaya, A.M., Merkulov, A.V., Shagina, N.B., Maltsev, V.N., Lévêque, P., Ulanova, A.M., Osipov, V.A. and Shafirkin, A.V., 2010. Confirmation studies of Soviet research on immunological effects of microwaves: Russian immunology results. Bioelectromagnetics, 31(8), pp.589-602*.**

***Grigoriev, Y., 2012. Mobile communications and health of population: the risk assessment, social and ethical problems. The Environmentalist, 32(2), pp.193-200*.**

***Repacholi, M., Grigoriev, Y., Buschmann, J. and Pioli, C., 2012. Scientific basis for the Soviet and Russian radiofrequency standards for the general public. Bioelectromagnetics, 33(8), pp.623-633*.**

8. Lines 46-48: “at non-thermal power densities (< 5 mW/cm2) and with particular emphasis on low power densities (<1mW/cm2)”. Above 1 mW/cm2 there can be thermal effects for frequencies 1-2 GHz. This is a very high power density. Change to: “at non-thermal power densities (< 1 mW/cm2)”.

**
*We changed it to the following sentence:*
**

**
*“This included the world literature in English and Russian reports translated to English, on RFR from 600 MHz – 90 GHz, the spectrum of wireless communication radiation (2G – 5G inclusive), with particular emphasis on non-thermal, low power densities (< 1 mW/cm^2^) and long term exposures.”*
**

9. Use identical category tittles in the text (pages 5-9) and in Table 1.

**
*As per your request, we revised the text titles as follows:*
**

**
* “Effects on the Blood” to “Blood Changes”*
**

**
* “The Immune Response” to “Immune System Disruption and Activation”*
**

**
* “Intracellular Calcium Levels” to “Increased Intracellular Calcium”*
**

**
* “Heart Disease and Arrhythmias” to “Cardiac Effects”*
**

10. Explain ALL names with initials throughout the manuscript first time met (SARS, COVID, Wi-Fi, LTE, ROS, ACE-2, ARDS, ICU, etc.)

***Full names have been provided for all acronyms (abbreviations) when they first appear I the manuscript, as requested, thank you*.**

11. Page 5, left, first paragraph. Rubik 2014 does not look like a peer reviewed paper. Please refer only to peer review publications, specifically for scientific findings.

***While we appreciate that you would prefer all references to be peer-reviewed, we would like to cite certain important papers that have not been peer-reviewed. At this critical time of the pandemic, many manuscripts that are not peer-reviewed are being cited in professional journal papers on COVID-19 to help experts bring forth knowledge as quickly as possible to facilitate an end to human suffering and death. In this particular case, we maintain that it is important to cite this paper on morphological changes in red blood cells from wireless communication radiation that relate to blood clotting, especially since SARS-CoV-2 and its spike protein have been shown to be thrombogenic [Grobbelaar, L.M., Venter, C., Vlok, M., Ngoepe, M., Laubscher, G.J., Lourens, P.J., Steenkamp, J., Kell, D.B. and Pretorius, E., 2021. SARS-CoV-2 spike protein S1 induces fibrin (ogen) resistant to fibrinolysis: Implications for microclot formation in COVID-19. medRxiv.]. Furthermore, this particular research on morphological changes in red blood cells from cell phone radiation exposure was conducted by one of us (Rubik) who has 25+ years of experience in live blood microscopy. So, although the morphological changes in red blood cells are not as well supported in this paper as we would like, we hope to plant seeds for future research to explore this phenomenon*.**

**
*Therefore, we modified the paragraph as follows:*
**

**
*“RFR exposure can cause morphologic changes in blood readily seen via microscopic examination of live peripheral blood samples. In 2013, Havas observed erythrocyte aggregation including rouleaux (rolls of stacked red blood cells) in live peripheral blood samples following 10 minute human exposure to a 2.4 GHz cordless phone. Although not peer reviewed, one of us (Rubik) investigated the effect of 4G LTE (fourth generation, long-term evolution) mobile phone radiation on the peripheral blood of ten human subjects, each of whom had been exposed to cell phone radiation for two consecutive 45-minute intervals (Rubik, 2014). Two types of effects were observed: increased stickiness and clumping of red blood cells with rouleaux formation, and subsequent formation of echinocytes (spiky red blood cells). Red blood cell clumping and aggregation are known to be actively involved in blood clotting (Wagner et al., 2013). The prevalence of such blood changes upon exposure to RFR in the human population has not yet been determined. Larger controlled studies should be performed to further investigate this phenomenon.”*
**

12. Page 7, right, lines 3-5. Delete the last sentence “Similar effects are observed in both that are caused by increased free radical formation and glutathione deficiency”.

***Sentence deleted as requested, thank you*.**

13. Line 11. Explain “spike protein”.

**
*Explanation for the term spike protein provided accordingly as requested as shown here:*
**

**
*“Once the virus gains access to a host cell via one of its spike proteins, which are the multiple protuberances projecting from the viral envelope that bind to ACE-2 receptors, it converts the cell into a virus self replicating machine.”*
**

14. In the effects on immune system cite and discuss the reviews by Szmigielski (2013) [Reaction of the immune system to low-level RF/MW exposures. Science of the Total Environment 454-455 (2013) 393-400], and Johansson 2009 [Disturbance of the immune system by electromagnetic fields-A potentially underlying cause for cellular damage and tissue repair reduction which could lead to disease and impairment. Pathophysioloy. 16(2-3):157-77].

***We appreciate these references. We added the following paragraph to the section on Immune System Disruption and Activation, and we added both references to the Reference Section*.**

**
*“In 2009, Johansson conducted a literature review, which included the 2007 Bioinitiative Report. He concluded that EMF exposure, including RFR, can disturb the immune system and cause allergic and inflammatory responses at exposure levels significantly less than current national and international safety limits and raise the risk for systemic disease. A review conducted by Szmigielski in 2013 concluded that weak RF/microwave fields, such as those emitted by mobile phones, can affect various immune functions both in vitro and in vivo. Although the bioeffects have been somewhat inconsistent, most research studies document alterations in the number and activity of immune cells from RF exposure. In general, short term exposures to weak microwave radiation may temporarily stimulate an innate or adaptive immune response, but prolonged irradiation inhibits those same functions.”*
**

15. Page 8, left, lines 45-49. Pall (2013) made an observation that calcium channels play a major role in EMF bioeffects. A very similar observation was made long before by Walleczek (1992) [Electromagnetic field effects on cells of the immune system: The role of calcium Signaling. FASEB J, 6, 3177-85]. These were both review studies, not mechanisms. The mechanism for ion channel gating by EMFs is published by Panagopoulos et al (2002) [Mechanism of action of electromagnetic fields on cells. Biochemical and Biophysical Research Communications, 298(1), 95-102], and refers not only to calcium but to all cation channels. Please cite and discuss these studies as well.

**
*Thank you for providing these references. The following text was added:*
**

***“In 1992, Walleczek first suggested that ELF (extremely low frequency) electromagnetic fields (<300 Hz) may be affecting membrane-mediated Ca^2^+ signaling and lead to increased intracellular Ca^2^+. The irregular gating of electrosensitive cell membrane ion channels by coherent, pulsed, oscillating electromagnetic fields was first presented by Panagopoulos, et al., in 2002. Pall combined these two observations to propose that low frequency RFR may be causing increased intracellular Ca^2+^ via the activation of voltage-gated calcium channels (Pall, 2013)*.**

16. Same paragraph “viruses hijack calcium channels and increase intracellular Ca^2^+ (Chen et al., 2019)”. Ion channels are ion-specific by means of ion radius. Thus calcium channels would not allow larger molecules to pass through them such as viruses. Therefore the claim by Chen et al 2019 is likely impossible. This should be thoroughly searched and discussed.

***Thank you for pointing this out so that we can clarify the concept. As you surmised, viruses do not pass through calcium channels. In the case of most viruses, including SARS-CoV-2, the process of a virus entering a host cell is called “viral endocytosis,” which involves the initial binding of the SARS-CoV-2 spike protein to the ACE2 receptor of host cells. The next steps involve a complex process leading to the viral penetration through the host cell plasma membrane into the cytosol [Mercer, J., Schelhaas, M. and Helenius, A. 2010. Virus entry by endocytosis. Annual review of biochemistry 79:803-833. https://doi.org/10.1146/annurev-biochem-060208-104626]*.**

***However, contrary to what you wrote, Chen’s claim is upheld, because following endocytosis and viral takeover of the host cell, certain viral proteins, which are then manufactured in the host cell, manipulate calcium channels and increase intracellular Ca^2+^*.**

**
*The text has been modified as follows:*
**

**
*“It has been reported that some viruses can manipulate voltage-gated calcium channels to increase intracellular Ca^2+^ thereby facilitating viral entry and replication (Chen et al. 2019). Research has shown that the interaction between a virus and voltage-gated calcium channels promote virus entry at the virus-host cell fusion step. Then, after the virus binds to its receptor on a host cell and enters the cell via endocytosis, the virus takes over the host cell to manufacture its components. Certain viral proteins then manipulate calcium channels, thereby increasing intracellular Ca^2+^, which facilitates further viral replication.”*
**

17. Right column, line 9. Explain “second messenger”

**
*The term, “second messenger,” was explained as requested:*
**

**
*“Intracellular Ca^2+^ is a ubiquitous second messenger relaying signals received by cell surface receptors to effector proteins involved in numerous biochemical processes.”*
**

18. Page 9, left, first paragraph. Explain “plaque instability”.

***The term plaque instability was explained as requested*.**

**
*“Immune system activation along with alterations in the immune system may result in atherosclerotic plaque instability and vulnerability, i.e., presenting an increased risk for thrombus formation, and contributing to the development of acute coronary events and cardiovascular disease in COVID-19.”*
**

19. Lines 26-27, “Potekhina et al. (1992) found that certain frequencies (55 GHz; 73 GHz) caused pronounced arrhythmia”. There are no physiological GHz frequencies in any living organism. All living functions are connected with ELF frequencies. It is thus unlikely that the GHz frequencies caused these effects. Instead the effects were most likely induced by the ELF pulsations. Similarly in Havas et al (2010). RF studies should report whether the field is pulsed/modulated or continuous wave. Most RF exposures contain ELF pulsing and/or modulation. This includes 2G-3G-4G and 5G as well. Please search the issue and revise accordingly.

***Water absorbs broadly in the GHz spectral region and also displays GHz resonant frequencies. Since living organisms consist of mostly water, organisms absorb GHz, too. Consider the fact that water absorbs 2.45 GHz, which is widely used in wireless communication routers and also in microwave ovens. Irradiation at water resonant frequencies, of which there are several in the GHz spectral region, may elicit bioeffects due to structural changes in the aqueous matrix of living cells. A paper reported that low-intensity electromagnetic radiation of 70.6 and 73 GHz affects E. coli bacterial growth and changes the properties of water. [Torgomyan H, Kalantaryan V, Trchounian A. Low intensity electromagnetic irradiation with 70.6 and 73 GHz frequencies affects Escherichia coli growth and changes water properties. 2011. Cell biochemistry and biophysics. 60(3):275-81. https://doi.org/10.1007/s12013-010-9150-8 ] It is hypothesized that water affected by absorption of GHz radiation affects the hydration of protein molecules in organisms that may alter rates of biochemical reactions (Betskii and Lebedeva, 2004). Thus, continuous wave GHz radiation, by altering intracellular water structure and protein hydration, could subsequently change the biochemistry and physiology*.**

**
*In addition, we have been careful to report pulse modulation and other wave parameters as reported in the literature. In the section where we describe the Havas (2010) study, we admit that we initially missed the 100 Hz pulse modulation, but we have now added it.*
**

**
*Please also see these two review papers, which we have cited and referenced in our manuscript, which summarize a substantial number of bioeffects of continuous wave as well as various types of modulated GHz radiation:*
**

**
*Pakhomov, A.G., Y. Akyel, O.N. Pakhomova, B.E. Stuck and M.R. Murphy. 1998. Review article: current state and implications of research on the biological effects of millimeter waves. Bioelectromagnetics, 19: 393-413. DOI:10.1002/(SICI)1521-186X(1998)19:7<393::AID-BEM1>3.0.CO;2-X*
**

**
*Betskii O.V. and Lebedeva, N.N. 2004. Low-intensity millimeter waves in biology and medicine. In: Clinical Application of Bioelectromagnetic Medicine, Marcel Decker, New York, pp. 30-61. https://gabrielecripezzicom/wp-content/uploads/2019/06/d75d92b7fb8f4d13ae5461e26afa62e87e60.pdf*
**

**
*Thank you for drawing our attention to these references: (Potekhina et al., 1992; Havas et al., 2010). The text has been modified as follows:*
**

**
*“Potekhina et al. (1992) found that certain frequencies (55 GHz; 73 GHz) caused pronounced arrhythmia. Although the nature of the primary response to millimeter waves and consequent events are poorly understood, a possible role for receptor structures and neural pathways in the development of continuous millimeter wave-induced arrhythmia has been proposed (Pakhomov et al., 1998).”*
**

**
*“Havas et al. (2010) reported that human subjects in a controlled, double-blinded study were hyper-reactive when exposed to 2.45 GHz, digitally pulsed (100 Hz) microwave radiation, developing either an arrhythmia or tachycardia and up-regulation of the sympathetic nervous system, which is associated with the stress response.”*
**

20. Right column, lines 29-31, “The bioeffects of RFR exposure are typically nonlinear rather than exhibiting the familiar linear dose-response effects from biochemical.” This is not true generally. The sporadic existence of “windows” does not render all effects non-linear. Effects depending on intensity or exposure time most usually are dose-dependent, and even close to linear. This should be revised.

**
*Thank you for drawing our attention to this matter. We have revised the text as follows:*
**

**“*RFR bioeffects depend upon specific values of wave parameters including frequency, power density, exposure time and modulation characteristics, as well as the cumulative history of exposure. Similar to ionizing radiation, the bioeffects of RFR exposure should be subdivided into deterministic, i.e. dose dependent effects, and stochastic effects that are seemingly random. Importantly, RFR bioeffects can also involve “response windows” of specific parameters whereby extremely low level fields can have disproportionally detrimental effects (Blackman, et al., 1989).”***

21. Page 10, left, lines 16-17, “However, these guidelines were established in 1996”. Provide reference.

**
*We provided a citation and reference to the original FCC document as follows:*
**

**
*Federal Communications Commission (FCC), 1996. Guidelines for evaluating the environmental effects of radiofrequency radiation. FCC96-326; ET Docket No. 93-62. https://transition.fcc.gov/Bureaus/Engineering_Technology/Orders/1996/fcc96326.pdf*
**

Reviewer #6: Xu et al reported that in Feb. 2020, the cases fatality rate was much lower in Zhejiang province and other provinces were much lower than it was in Wuhan. (Xiao-Wei Xu, physician1, Xiao-Xin Wu, physician1, Xian-Gao Jiang, physician2, Kai-Jin Xu, physician1, Ling-Jun Ying, physician3, Chun-Lian Ma, physician4, Shi-Bo Li, physician5, Hua-Ying Wang, physician6, Sheng Zhang, physician7, Hai-Nv Gao, professor8, Ji-Fang Sheng, professor1, Hong-Liu Cai, physician1, Yun-Qing Qiu, professor1, Lan-Juan Li, professor1. Clinical findings in a group of patients infected with the 2019 novel coronavirus (SARS-Cov-2) outside of Wuhan, China: retrospective case series. BMJ 2020; 368 doi: https://doi.org/10.1136/bmj.m606).

***We read this paper (Xu, 2020), and then we searched online to determine whether 5G was implemented in Zhejiang province by late 2019. We found that Zhejiang province had partial installation of 5G in 2019, at least in major cities such as Hangzhou, Wenzhou, and Ningbo. Moreover, only a small number of cases were used in this retrospective study whose reference you provided, which may not accurately reflect the actual number of cases and fatality rates in these provinces. Due to this, we maintain that it would be inappropriate for us to utilize these findings or this reference in our manuscript. Thus, we did not make any modifications to our paper based on this information*.**

The following citation should be used along with the Pakhomov et al citation with regard to millimeter wave effects: Betskii OV, Lebedeva NN. 2004 Low-intensity millimeter waves in biology and medicine. In: Clinical Application of Bioelectromagnetic Medicine, Marcel Decker, New York, 2004, pp. 30-61. https://gabrielecripezzi.com/wp-content/uploads/2019/06/d75d92b7fb8f4d13ae5461e26afa62e87e60.pdf

**
*Thank you for providing this additional citation and reference. It has been added accordingly as shown here in the manuscript text, in the last sentence of the section, Overview on Bioeffects of Radiofrequency Radiation (RFR) Exposure:*
**

**
*“Two comprehensive reviews on the bioeffects of millimeter waves report that even short-term exposures produce marked bioeffects (Pakhomov et al., 1998; Betskii & Lebedeva, 2004).”*
**

On p.5, there is place which states (5G RFR) - this is confusing since 5G is not radiofrequency.

***Thank you for making this observation. (5G RFR) has been removed from the sentence on page 5*.**

The author should state in several places in the paper, that the findings are suggestive of a link between EMF exposure and the severity of COVID-19 infections but none of these findings are considered to be proof of such a linkage.

**
*Thank you for this recommendation. We included this text in the Discussion Section:*
**

**“*This evidence suggests that RFR may have worsened the COVID-19 pandemic by weakening the host and exacerbating COVID-19 disease. However, none of the observations discussed here have proven this linkage. Specifically, the evidence does not confirm causation.”***

The Sen et al, citation has an error in it. It should read NF-kB. and that is a Greek letter kappa. There may also be an error in the text.

***These corrections have been made accordingly in the manuscript text, table, and reference*.**

The other thing I would suggest is that the author should make a suggestion or two about how the remaining uncertainty here might be resolved. I would make two suggestions that might help:

There could be one or more studies to determine whether some COVID-19 patients admitted to the hospital could be shielded in either Faraday cage or a shielded canopy could be put over the bed. These could lower exposures and hospitals are high EMF environments such with high powered Wi-Fi systems, many wireless communication devices and thousands of electronic devices, producing large amounts of dirty electricity. I know that hospitals have high levels of dirty electricity, having measured levels myself. The question being, whether such shielding would lower mortality rates and/or shorten times to patient release.

***In consideration of your proposed study utilizing RFR shielding of COVID-19 patients, we think it is impractical and potentially dangerous to patients, and therefore, unlikely to be carried out in hospitals. Please know that wireless monitoring of patients in hospitals is now routine, such that shielding patients would not permit this critical real-time online patient monitoring. It is therefore unlikely that an Institutional Review Board would approve of placing shielding around a patient or placing a patient in a Faraday cage where they could not be easily monitored for any potentially dangerous physiological changes. Thus, we did not add this proposed study to our Discussion Section of the manuscript*.**

Another approach would be to measure home and working environments for EMF levels comparing patients with similar risk factor exposure but different severity of illness.

**
*We have now proposed two future studies in the Discussion Section:*
**

**
*“The question of causation could be investigated in future studies. For example, a clinical study could be conducted in COVID-19 patient populations with similar risk factors, to measure the RFR daily dose in COVID-19 patients and look for a correlation with disease severity and progression over time. As wireless device frequencies may differ, and the power densities of RFR fluctuate constantly at a given location, this study would require patients to wear personal microwave dosimeters (monitoring badges). In addition, controlled laboratory studies could be conducted on animals, e.g., humanized mice infected with SARS-CoV-2, in which groups of animals exposed to minimal RFR (control group) as well as medium and high power densities of RFR could be compared for COVID-19 disease severity and progression.”*
**

Reviewer #8: This paper delves into a facet of CoVid-19 and the evolving use of the widespread release of 5G into the environment, and the relationship between the two. This relationship has rarely been

studied in the literature. That alone seems to make this paper relatively significant. It also provides evidence that the Precautionary Principle would call for more study before continuing the

widespread deployment of 5G towers across the world, as well as the release of satellites that are envisioned to surround the planet. While bringing Internet access to the world’s population seems

to be a good thing (in providing information to the societies that lack it now), we need to consider the Law of Unintended Consequences. The plans are for 5G to blanket the world, leaving no place

without this new exposure to this Microwave and Milliwave radiation, and potentially effecting all life on earth, with the exception of beings that live in lead-lined shelters, or Faraday cages. Based

on this paper alone, it indicates the need for further studies, before this is widely deployed. Especially when we are still learning about the effects of the CoVid-19 pandemic itself. The emerging

understanding of how CoVid-19 has been linked to coagulation disorders, and to the hypoxic effect on the lung function of oxygenation absorption and release, via the red cells, adds additional

urgency to the situation. As does the information on free radical creation causing havoc to Biological systems, the problems that either CoVid Or 5 G-type RFR can have on the heart, and especially

how either can effect the immune system, and the paper points out that that could make a lot of illnesses deadlier. And all of these have never been fully tested together, but usually apart, by

impartial researchers.

As this paper has reviewed the literature on the effects of RFR RadioFrequency Radiation, and considering how most studies in the past, looked at the heating effects of such radiation, and

then said that if the RFR did not effect heating of biological tissue, it was labeled as safe. But as this paper shows, there is multiple evidence that there are adverse effects on tissue systems and

their physiology, that have nothing to do with the heating effects of RFR. These low power high frequency effects have been collated, reviewed, and well documented in this paper, in a clearly

collected and tabular form, making it easy to compare the RFR research, and the emerging data from the effects of the CoVid-19 virus. And by documenting the similarity of the recognized RFR

effects on biological systems, then when these are clearly shown to be similar effects that are being discovered with the ever-changing documented effects of the CoVid-19 pandemic, the paper

makes a strong case for a halt that is needed at this point in the pandemic, to step back and really investigate the nature and gravity of these effects, when combined, before further RFR 5 G

potentially blankets the world.

As the paper points out in the discussion, while the connections and the evidence that both RFR and CoVid-19 attack similar biological systems and physiologies, the paper has not proven

causation, but they clearly have proven the point that further, independent research is needed, and soon. Given that CoVid-19 has shown the ability to mutate, and future pandemics are predicted,

this might be one “bright side” to this pandemic, in that it forces us to do this research, before it is too late. If indeed, all of the systems effected by CoVid-19 are also potentially weakened or

effected by RFR like the upcoming wave of 5G (or 6 G, etc.) that is planned for the future, we have a chance to do the research that is needed to make our future safer, but to not do this research,

and hide our heads in the sand, and ignore this paper, and its implications, then future generations may not look back kindly on our “rush to get faster internet to everyone.” Instead, we may do

much more damage than when people thought that Radium watches that glow in the dark were “cool,” or that it was “cool” to use the Xray machines in shoe stores to see if your shoes fit the feet,

by a live Xray that showed your foot skeleton in the shoe, while the Xray device was near the gonads. One wonders how many people died of those “cool” technologies, from cancer or other

illnesses, before their dangers were finally recognized and removed from the market.

How does the saying go: “Those who refuse to learn from the mistakes of the past, are condemned to repeat them,” only this time it may be on a world-wide scale? So I highly recommend that

this paper be accepted for publication, so that it can stimulate much more research that must be done in this area.

***We thank you for your review, support, and encouragement*.**

Reviewer #9: Dear Editor,

I completed review process of the manuscript numbered as “JCTRes-D-21-00034_reviewer”. Although the ideas put forward by the authors are not negligible, they are open issues to criticism. Because there is no scientific study that clearly reveals the relationship between RFRs, especially 5G and SARS-CoV-2. The authors have tried to make a good review, but the ideas they put forward show that only RFRs and SARS-CoV-2 have similar effects. Unfortunately, there is no scientific data about whether these similarities create a synergistic effect or not. So I suggest the authors change the title such as “Similarities in the effects of RFRs and SARS-CoV-2: Could there be a synergistic effect?”

***We prefer not to change the title to your suggested title, because your use of the word, “synergistic” implies a specific type of relationship that goes beyond a possible connection. “Synergy” is defined as the interaction of two or more agents or forces so that their combined effect is greater than the sum of their individual effects. We are not addressing the question of possible synergy in this paper. Instead, we are merely investigating the intersection of bioeffects of radiofrequency radiation exposure and COVID-19 manifestations*.**

**
*However, we have changed our paper title, eliminating the word “telecommunications” and substituting the word, “communications,” which is more general and inclusive. Our new title is, “Evidence for a Connection between COVID-19 and Exposure to Radiofrequency Radiation from Wireless Communications Including Microwaves and Millimeter Waves.” We also have an alternative title, if the Editor prefers, “A Proposed Connection between COVID-19 and Exposure to Radiofrequency Radiation from Wireless Communications Including Microwaves and Millimeter Waves.”*
**

The decision is yours, dear editor. In summary, the article can be printed, but the title is very ambitious! On the other hand, I suggest to the authors to read the articles given below to find some hints for the topic of the manuscript.

Sincerely

Recommendation for Table 1 in page 4; Table 1 is not sufficient as the authors stated. Fort this reason, the authors have to present characteristics of RFRs and name of the references. Therefore, the table should be more informative for the readers for evaluation of the situation.

***Thank you for this impressive reference list, which we have investigated*.**

**
*Our table was meant to be only a visual summary for the reader, not a comprehensive list with details and references. However, we changed the subheadings on the bioeffects in the text to reiterate the table subheadings. In this way, the reader is referred to particular sections of the text to obtain details regarding RFR exposure parameters and literature citations. We have also added a sentence to the table legend indicating how the reader can find this supportive information as follows:*
**

**
*“Supportive evidence including study details and citations are provided in the paper under each subject heading, i.e., Blood Changes, Oxidative Stress, etc.”*
**

Barlas SB, Adalier N, Dasdag O, Dasdag S, Evaluation of SARS-CoV-2 with a biophysical perspective. Biotechnology & Biotechnological Equipment. 35:1, 392-406, 2021. DOI: 10.1080/13102818.2021.1885997Dasdag S, Akdag MZ, Celik MS (2008), Bioelectrical parameters of people exposed to radiofrequency in workplace and houses provided to workers. Biotechnology & Biotechnological Equipment. 22: 3: 859-863.Alkis ME, Akdag MZ, Dasdag S, E¡ects of Low-Intensity Microwave Radiation on Oxidant-Antioxidant Parameters and DNA Damage in the Liver of Rats. 2020 Bioelectromagnetics. 42:76—85, 2021. DOI:10.1002/bem.22315Dasdag S, Balci K, Celik MS, Batun S, Kaplan A, Bolaman Z, Tekes S, Akdag Z (1992), Neurologic and biochemical findings and CD4 / CD8 ratio in people occupationally exposed to RF and microwave. Biotechnol. & Biotechnol. Eq. 6 / 4, 37 -39.Yilmaz F, Dasdag S, Akdag MZ, Kilinc N (2008).Whole body exposure of radiation emitted from 900 MHz mobile phones does not seem to affect the levels of anti-apoptotic BCL-2 protein. Electromagnetic Biology and Medicine. 27: 1; 65-72.Dasdag S, Akdag MZ, Ulukaya E (2009), Effects of Mobile Phone Exposure on Apoptotic Glial Cells and Status of Oxidative Stress in Rat Brain. Electromagnetic Biology and Medicine. 28: 4; 342-354.Dasdag S, Bilgin HM, Akdag MZ, et al. (2008), Effect of Long Term Mobile Phone Exposure on Oxidative and Antioxidative Process and Nitric Oxide in Rats. Biotechnology & Biotechnological Equipment. 22: 4; 992-997Alkis ME, Bilgin HM, Akpolat V, Dasdag S, Yegin K, Yavas MC, Akdag MZ, Effect of 900-, 1800-, and 2100-MHz radiofrequency radiation on DNA and oxidative stress in brain. Electromagn Biol Med. 38(1): 32-47, 2019.Akdag M, Dasdag S, Canturk F, Akdag MZ, Exposure to non-ionizing electromagnetic fields emitted from mobile phones induced DNA damage in human ear canal hair follicle cells. Electromagn Biol Med. 2018, 37 (2): 66-75. https://doi.org/10.1080/15368378.2018.1463246Bektas H, Dasdag S, Effect of Radiofrequencies Emitted from Mobile Phones and Wi-Fİ on Pregnancy. Journal of International Dental and Medical Research. 10(3): 1084-1095, 2017Bektas H, Dasdag S, Bektas S, Comparison of effects of 2.4 GHz Wi-Fi and mobile exposure on human placenta and cord Blood. Biotechnology & Biotechnological Equipment. 2020, VOL. 34 (1): 154-162, 2020, https://doi.org/10.1080/13102818.2020.1725639

**
*We have modified our manuscript to include summaries of the following papers from your list which are relevant for our paper:*
**

**(1) *Alkis, M.E., Akdag, M.Z. and Dasdag, S., 2021. Effects of low-intensity microwave radiation on oxidant-antioxidant parameters and DNA damage in the liver of rats. Bioelectromagnetics, 42(1), pp.76-85*. DOI:10.1002/bem.22315;****(2) *Dasdag, S., Bilgin, H.M., Akdag, M.Z., Celik, H. and Aksen, F., 2008. Effect of long term mobile phone exposure on oxidative-antioxidative processes and nitric oxide in rats. Biotechnology & Biotechnological Equipment, 22(4), pp.992-997. https://doi.org/10.1080/13102818.2008.10817595***

**
*We added short summaries of these two papers to the section on Oxidative Stress:*
**

**
*“In a long-term controlled study on rats exposed to 900 MHz (mobile phone frequency) at 0.0782 mW/cm2 for 2 hrs/day for 10 months, there was a significant increase in malondialdehyde (MDA) and total oxidant status (TOS) over controls (Dasdag et al., 2008). In another long-term controlled study on rats exposed to two mobile phone frequencies, 1800 MHz and 2100 MHz, at power densities 0.04 -0.127 mW/cm^2^ for 2 hours/day over 7 months, significant alterations in oxidant-antioxidant parameters, DNA strand breaks, and oxidative DNA damage were found (Alkis et al., 2021).”*
**

Reviewer #10: - Background section. Page 1, lines 49-52. “…that will dramatically increase the population’s wireless radiation exposure both inside structures and outdoors”.

Please cite evidence (referenced studies) supporting this hypothesis by specific models and/or real-time measurements.

**
*We could not find specific models, nor are real-time measurements available, so we decided instead to modify this sentence such that it remains hypothetical. However, with densification of 4G infrastructure along with the placement of new 5G antennas approximately every 300 meters, and 42,000 5G emitting satellites, it is logical to hypothesize that the population will experience increased wireless communication radiation exposure. Here is our modified sentence:*
**

**
*“The system requires significant densification of 4G as well as new 5G antennas that may dramatically increase the population’s wireless communications radiation exposure both inside structures and outdoors.”*
**

- Background section. Page 1. “During the first wave in the United States, COVID-19 attributed cases and deaths were higher in states with 5G infrastructure compared with states that did not yet have this technology (Tsiang and Havas, manuscript submitted)”. Unpublished/not available data should not be considered as citation.

**
*We agree, and we included this paper because we expected it to be published during the review process and it has, in fact, been published in a peer reviewed journal since our submission. The full reference, now added to our reference list in the manuscript, is:*
**

**
*“Tsiang, A. and Havas, M. 2021. COVID-19 Attributed Cases and Deaths are Statistically Higher in States and Counties with 5th Generation Millimeter Wave Wireless Telecommunications in the United States. Medical Research Archives 9(4): 1-32. DOI: 10.18103/mra.v9i4.2371”*
**

- Overview on covid-19 (page 2). In consideration of the main aim of the review, this paragraph can be significantly shortened.

***Our overview on COVID-19 is quite short at approximately ½ page. We think it provides a good background for readers in need, so we prefer not to shorten it*.**

- Authors should better describe the main technical characteristics of the 5G infrastructures (i.e, small cells, MIMO, multiple frequencies etc.), briefly listing the main technical differences with the previous radiofrequency networks.

**
*We provided a reference to the official document specifying 5G (3GPP, 2020). In addition, we re-wrote and expanded upon 5G on page 2 as follows:*
**

***“5G is a protocol that will use high frequency bands of the electromagnetic spectrum in the vast radiofrequency range from 600 MHz to nearly 100 GHz, which includes millimeter waves (>20 GHz), in addition to the currently used 3G (third generation) and 4G (fourth generation) long term evolution (LTE) microwave bands. 5G frequency spectrum allocations differ from country to country. Focused pulsed beams of radiation will emit from new base stations and phased array antennas placed close to buildings whenever persons access the 5G network. Because these high frequencies are strongly absorbed by the atmosphere and especially during rain, a transmitter’s range is limited to 300 meters. Therefore 5G involves base stations and antennas much more closely spaced than previous generations, plus satellites in orbit that will emit 5G bands globally to create a wireless worldwide web. The system requires significant densification of 4G as well as new 5G antennas that may dramatically increase the population’s wireless communications radiation exposure both inside structures and outdoors. Approximately 100,000 emitting satellites are planned to be launched into orbit. This infrastructure will significantly alter the world’s electromagnetic environment to unprecedented levels and may cause unknown consequences to the entire biosphere, including humans. The new infrastructure will service the new 5G devices, including 5G mobile phones, routers, computers, tablets, self-driving vehicles, machine-to-machine communications, and the Internet of Things (IoT)*.**

**
*The global industry standard for 5G is set by the 3rd Generation Partnership Project (3GPP), which is an umbrella term for several organizations developing standard protocols for mobile telecommunications. The 5G standard specifies all key aspects of the technology, including frequency spectrum allocation, beam-forming, beam steering, multiplexing MIMO (multiple in, multiple out) schemes to nearly simultaneously serve a large number of devices within a cell as well as modulation schemes among many others. The latest finalized 5G standard, Release 16, is codified in the 3GPP published Technical Report TR 21.916 and may be downloaded from the 3GPP server at https://www.3gpp.org/ftp/Specs/archive/21_series/21.916/ (3GGP, 2020).”*
**

**
*Reference: 3GPP (Third Generation Partnership Project, 2020. Technical Report TR 21.916, V1.0.0. (2020-12), pages 1-149. https://www.3gpp.org/ftp/Specs/archive/21_series/21.916/*
**

- Table 1. Authors should indicate in the table the most relevant and specific reference(s) for each listed point.

***Rather than modify the Table, which is meant to be only a summary to assist the reader, we changed the text subheadings on bioeffects to be identical to the Table subheadings. We added this sentence in the Table legend to direct the reader to those text sections for the evidence and citations:*
*“Supportive evidence including study details and citations are provided in the paper under each subject heading, i.e., Blood Changes, Oxidative Stress, etc.”***

- Table 1. it is not clear if the cited effects (“RFR exposure bioeffects”) have been generically linked with high frequency electromagnetic fields or, specifically, with 5G frequencies.

Authors should include in the “RFR exposure bioeffects” listed in this table the frequency, the level (i.e., power density) and the period of exposure linked with each of the cited effects. Authors should also specify the type of study (i.e., in vitro, animal or human study).

***As previously stated in our Methods Section, all studies reviewed in this paper involve exposures to RFR in the range from 600 MHz – 90 GHz, the spectrum of wireless communication radiation, from 2G – 5G inclusive. The detailed information on exposure parameters and types of studies is given in each section of the text with the same headings as in the Table, i.e., Blood Changes, Oxidative Stress, etc*.**

- Authors should report the average level of RFR exposure measured in at least some geographical areas implementing 5G infrastructure. A comparison of the “real-life” level of exposure with the RFR exposure levels generating the majority of bioeffects described in the paper is needed.

**
*We inserted the following text in the manuscript in the section on the Overview on Bioeffects of RFR Exposure:*
**

**
*“By comparison to the exposure levels employed in these studies, we measured the ambient level of RFR from 100 MHz – 8 GHz in downtown San Francisco, California in December, 2020, and found an average power density of 0.0002 mW/cm^2^. This is approximately 2x10E10 times above the natural background.”*
**

- Several bioeffects described by authors are generated by levels of exposure significantly higher than those generally recorded in urban areas. Authors should include in the paper a new table listing the bioeffects potentially linked with covid-19 and observed in the presence of levels of environmental exposure comparable with those recorded in the most exposed urban areas.

***Such data on the average power density of 5G (or 4G) in various geographical locations are unavailable, neither shown in the scientific literature, nor published by cities or other governments. Little is known about population exposure from real-world radiofrequency radiation sources as we previously wrote in our Discussion Section. It is also very difficult to accurately quantify the average power density at a given location. Moreover, the average power density varies greatly, depending upon the specific location, time, averaging interval, frequency, and modulation scheme. For a specific municipality it depends on the antenna density, what network protocols are used, as, for example, 2G, 3G, 4G, 5G, Wi-Fi (IEEE 802.11b Direct Sequence), WiMAX (Worldwide Interoperability for Microwave Access), DECT (Digital European Cordless Telecommunications), RADAR (Radio Detection and Ranging), and what the legal limits are for public exposure in the particular jurisdiction. RFR (radiofrequency radiation) from ubiquitous radiowave transmitters, including antennas, base stations, smart meters, mobile phones, routers, satellites, and other wireless devices currently in use superimposes and yields an additive average power density at a given location that typically fluctuates greatly over time. Using a consumer grade radiofrequency power meter to measure ambient levels from 100 MHz – 8 GHz in downtown San Francisco, California, recently we found an average power density of 0.0002 mW/cm^2^, which is approximately 1 billion times greater than the natural background. However, our RF meter was insensitive to 5G frequencies above 8 GHz*.**

**
*We added the following paragraph to the Discussion Section:*
**

**
*“Another shortcoming of this study is that we do not have access to experimental data on 5G exposures. In fact, little is known about population exposure from real-world RFR, which includes exposure to RFR infrastructure and the plethora of RFR emitting devices. In relation to this, it is difficult to accurately quantify the average power density at a given location, which varies greatly, depending upon the time, specific location, averaging interval, frequency, and modulation scheme. For a specific municipality it depends on the antenna density, what network protocols are used, as, for example, 2G, 3G, 4G, 5G, Wi-Fi, WiMAX (Worldwide Interoperability for Microwave Access), DECT (Digital European Cordless Telecommunications), and RADAR (Radio Detection and Ranging). RFR from ubiquitous radiowave transmitters, including antennas, base stations, smart meters, mobile phones, routers, satellites, and other wireless devices currently in use, superimposes and yields an additive average power density at a given location that typically fluctuates greatly over time. No experimental studies on adverse health effects or safety issues of 5G have been reported, and none are currently planned by the industry, although this is sorely needed.”*
**

- Authors should report and comment previous studies, if available, linking RFR exposure with viral diseases different from Covid-19.

***We searched the scientific and medical literature, but we did not find any studies on RFR exposure linked to other viral diseases*.**

- Authors discuss evidence deriving from exposure to cell phones to support possible effects of environmental exposure to 5G. However, exposure to cell phones or to 5G infrastructure (i.e., base stations, MIMO antennas, devices etc.) may significantly differ in terms of SAR and are not fully comparable.

***We discussed evidence not only from exposure to cell phone radiation, but also from exposure to Wi-Fi in this paper. We agree that exposure to 5G infrastructure (i.e., base stations, MIMO antennas, etc.) may significantly differ in terms of SAR and are not fully comparable. We had previously pointed out in our Discussion Section that data on bioeffects from real world 5G emissions was seriously lacking*.**

**
*We added additional material discussing evidence from exposure to cell or mobile phones, which appears on page 4:*
**

**
*“Chronic human exposure to 0.000006 to 0.00001 mW/cm^2^ produced significant changes in human stress hormones following a mobile phone base station installation (Bucher and Eger, 2012). Human exposures to cell phone radiation at 0.00001 – 0.00005 mW/cm^2^ resulted in complaints of headache, neurological problems, sleep problems, and concentration problems, corresponding to ‘microwave sickness’ (Navarro et al., 2003; Hutter et al., 2006).”*
**

**
*And in another paragraph on page 4 we added this:*
**

**
*“In a long-term (1 - 4 year) study on children who use mobile phones compared to a control group, functional changes, including greater fatigue, decreased voluntary attention, and weakening of semantic memory, among other adverse psychophysiological changes, were reported (Grigoriev, 2012).”*
**

**
*On page 9, we added this:*
**

**
*“A review conducted by Szmigielski in 2013 concluded that weak RF/microwave fields, including those emitted by mobile phones, can affect various immune functions both in vitro and in vivo.”*
**

- The majority of the bio-effects described by Authors could also be, at least theoretically, attributed to pre-existing radiofrequency exposure, in particular in highly exposed geographical areas. Furthermore, in the short-medium term, in exposed areas the level of RFR exposure can be assumed to be constant. On the other hand, covid-19 incidence, morbidity and mortality significantly varied during the last year. The lack of a parallel trend should limit the hypothesis of a direct link between 5G exposure and covid-19 clinical and epidemiological aspects.

***The total radiofrequency radiation exposure is due to radiation from a combination of wireless infrastructure (4G and 5G antennas, base stations, smart meters) as well as wireless communication products in the homes, schools, and workplaces. We disagree with your assumption that in exposed regions the level of RFR exposure can be assumed to be constant last year, especially since the installation of 5G was being implemented in many locations throughout the world in 2020. Thus, in 2020, we expected the level of RFR exposure to increase in these locations*.**

- The majority of the bio-effects described by Authors could also be attributed to other sources of environmental pollution and, in particular, to air pollution. The effect of this and other relevant confounders in urban areas characterized by high population density is not discussed by Authors.

**
*It is possible that air pollution is another potential contributing environmental factor in the pandemic, although it is not a subject relevant to our thesis, such that we do not address it in depth in our paper. Nonetheless, we added the following sentence to the Discussion Section:*
**

**
*“Air pollution, particularly PM 2.5 microparticulates, likely increased symptoms in patients with COVID-19 lung disease (Fiasca et al., 2020).”*
**

**
*Reference: Fiasca F., Minelli M., Maio D., Minelli M., Vergallo I., Necozione S., Mattei A. 2020. Associations between COVID-19 Incidence Rates and the Exposure to PM2.5 and NO_2_: A Nationwide Observational Study in Italy. Int J Environ Res Public Health. 17(24):9318. doi: 10.3390/ijerph17249318*
**

- According to some evidence, children can be particularly vulnerable to RFR effects. However, the pediatric age seems to be the less involved, at least in terms of clinical manifestation, by the covid-19 pandemic. How Authors could explain this different outcome in different age classes equally exposed to RFR?

***Children are less vulnerable than adults to the SARS-CoV-2 virus because they have fewer ACE2 receptors. Elderly adults have the most ACE2 receptors, i.e., more “targets” for the virus to enter their cells, and are thus more vulnerable to the virus. Both the very young and the very old populations are the most vulnerable to adverse effects from RFR exposure. Even so, the question of age-related exposure to wireless communication radiation in relation to the pandemic goes beyond the scope of our paper*.**

- Page 9, discussion section. “The evidence indicates that RFR may weaken the host, exacerbate COVID-19 disease, and thereby worsen the pandemic”. In the opinion of this reviewer, the reported evidence only indicate that mechanisms possibly involved in the clinical progression of SARS-CoV-2 could be also generated, according to experimental data, by RFR exposure. It is still under debate, however, if these biological effects can be present in the case of frequencies and levels of RFR exposure commonly found in the urban areas where 5G networks have been implemented.- Point of strength and limitations of the review performed by authors should be clearly stated.

**
*We added this sentence to the Conclusion Section:*
**

**
*“The evidence presented here indicates that mechanisms involved in the clinical progression of COVID-19 could also be generated, according to experimental data, by RFR exposure.”*
**

**
*We have rewritten our Discussion Section and point out more clearly the strengths and limitations of our review, as follows:*
**

***“A major strength of this study is that the evidence rests on a large body of scientific literature reported by many scientists worldwide and over several decades--experimental evidence of adverse bioeffects of RFR exposure at nonthermal levels on humans, animals, and cells. The Bioinitiative Report (Sage and Carpenter, 2012) and updated in 2020, summarizes hundreds of peer-reviewed scientific papers documenting evidence of nonthermal effects from exposures less than or equal to 1mW/cm^2^. Even so, some laboratory studies on the adverse health effects of RFR have sometimes utilized power densities exceeding 1mW/cm^2^. In this paper, almost all of the studies that we reviewed included experimental data at power densities less than or equal to 1mW/cm^2^*.**

***A potential criticism of this study is that adverse bioeffects from nonthermal exposures are not yet universally accepted in science and are not considered when establishing public health policy in many nations. Decades ago, Russians and Eastern Europeans compiled considerable data on nonthermal bioeffects, and subsequently set guidelines at lower RFR exposure limits than the US and Canada, i.e., below levels where nonthermal effects are observed. However, the Federal Communications Commission (FCC, a US government entity) and ICNIRP (International Commission for Non-Ionizing Radiation Protection) guidelines operate on thermal limits based on outdated data from decades ago, allowing the public to be exposed to considerably higher RFR power densities. Regarding 5G, the telecommunication industry claims that it is safe because it complies with current RFR exposure guidelines of the FCC and ICNIRP. These guidelines were established in 1996 (Federal Communications Commission, 1996), are antiquated, and are not safety standards. Thus, there are no universally accepted safety standards for wireless communication radiation exposure. Recently international bodies such as the EMF Working Group of the European Academy of Environmental Medicine, have proposed much lower guidelines, taking into account nonthermal bioeffects from RFR exposure in multiple sources (Belyaev et al., 2016)*.**

***Another weakness of this study is that some of the bioeffects from RFR exposure are inconsistently reported in the literature. Replicated studies are often not true replications. Small differences in method, including unreported details such as prior history of exposure by the organisms, non-uniform body exposure, and other variables can lead to inadvertent inconsistency. Moreover, not surprisingly, industry-sponsored studies tend to show less adverse bioeffects than studies conducted by independent researchers, suggesting industry bias (Huss et al., 2007). Some experimental studies that are not industry-sponsored have also shown no evidence of harmful effects of RFR exposure. It is noteworthy, however, that studies employing real-life RFR exposures from commercially available devices have shown high consistency in revealing adverse effects (Panagopoulos, 2019)*.**

***RFR bioeffects depend upon specific values of wave parameters including frequency, power density, exposure time, and modulation characteristics, as well as the cumulative history of exposure. Similar to ionizing radiation, the bioeffects of RFR exposure can be subdivided into deterministic, i.e., dose dependent effects and stochastic effects that are seemingly random. Importantly, RFR bioeffects can also involve “response windows” of specific parameters whereby extremely low level fields can have disproportionally detrimental effects (Blackman et al., 1989). This nonlinearity of RFR bioeffects can result in biphasic responses such as immune suppression from one range of parameters, and immune hyperactivation from another range of parameters, leading to variations that may appear inconsistent*.**

***In gathering papers and examining existing data for this study, we looked for outcomes providing evidence to support a proposed connection between the bioeffects of RFR exposure and COVID-19. We did not make an attempt to weigh the evidence. The RFR exposure literature is extensive and currently contains over 30,000 research reports dating back several decades. Inconsistencies in nomenclature, reporting of details, and cataloging of keywords make it difficult to navigate the literature*.**

**
*Another shortcoming of this study is that we do not have access to experimental data on 5G exposures. In fact, little is known about population exposure from real-world RFR, which includes exposure to RFR infrastructure and the plethora of RFR emitting devices. In relation to this, it is difficult to accurately quantify the average power density at a given location, which varies greatly, depending upon the time, specific location, averaging interval, frequency, and modulation scheme. For a specific municipality it depends on the antenna density, what network protocols are used, as, for example, 2G, 3G, 4G, 5G, Wi-Fi, WiMAX (Worldwide Interoperability for Microwave Access), DECT (Digital European Cordless Telecommunications), and RADAR (Radio Detection and Ranging). RFR from ubiquitous radiowave transmitters, including antennas, base stations, smart meters, mobile phones, routers, satellites, and other wireless devices currently in use, superimposes and yields an additive average power density at a given location that typically fluctuates greatly over time. No experimental studies on adverse health effects or safety issues of 5G have been reported, and none are currently planned by the industry, although this is sorely needed.”*
**

Reviewer #11: In this work authors summarize current state of knowledge about the harmful effects of Radiofrequency radiation (RFR), with special focus on those that could possibly enhance the possibility of being infected by COVID-19.

This review is a nice mix of very recent publications (last 5 years) and some classic papers mainly from Soviet Union and United States, showing, that knowledge about harmful effects of RFR have been extensively studied already few decades ago, what is very important mainly because of the number of conspiracy theories available of the Internet about 5G RFR. Call for the scientific evaluation of this exposure type is more than proper. Otherwise we will be wittnesses of very big population study, that will reveal the truth in the future times.

I do not have any big questions rather some comments:

1) In the Introduction part authors cite quite few review articles. I would suggest citation of the few of the best experimental articles considering this type of deleterious effect (e.g. oxidative stress, reproductive damage), since the number of the reviews considering RFR is relatively high, but actual experimental studies that strongly supports the conclusion of reviews are sometimes hard to find or other times not so conclusive.

***We already selected the best experimental articles, including review articles, to support our purported thesis that adverse effects to RFR exposure intersect with COVID-19 manifestations*.**

2) Authors stated that oxidative stress induced by RFR may exacerbate the seriousness of COVID -19 disease. I agree that induction of oxidative stress is the most common harmful effect observed after exposure to RFR, targeting mostly the cells with the high level of metabolism, like sperm cells. But in a lot of studies induction of ROS after exposure is not higher than 50% of control values and some studies have even seen adaptation to the radiation with increased exposure time on the cellular level.

***It is certainly true that not all studies designed to test for oxidative stress following exposure to RFR show positive results. However, a very large number of studies do show positive results*. *As we already stated in our earlier paper draft in the section, Oxidative Stress, “Among 100 currently available peer-reviewed studies investigating oxidative effects of low-intensity RFR, 93 studies confirmed that RFR induces oxidative effects in biological systems (Yakymenko et al., 2015).” Moreover, we have added summaries of two more studies on long-term exposure (7 – 10 months) of rats to mobile phone radiation frequencies that show statistically significantly greater oxidative stress levels over controls:***

**
*“In a controlled study on rats exposed to 900 MHz (mobile cell phone frequency) at 0.0782 mW/cm2 for 2 hrs/day for 10 months, there was a significant increase in malondialdehyde (MDA) and total oxidant status (TOS) over controls (Dasdag et al., 2008). In another controlled study on rats exposed to two mobile phone frequencies, 1800 MHz and 2100 MHz, at power densities 0.04 -0.127 mW/cm^2^ for 2 hours/day over 7 months, significant alterations in oxidant-antioxidant parameters, DNA strand breaks, and oxidative DNA damage were found (Alkis et al., 2021).”*
**

3) Authors suggest that 5G introduction in the cities that have been hit with the COVID-19 very hard in the first wave could lead to the increased mortality and number of cases. Since there are some connections, this could be also explaine by the fact that Northern Italy is the region with the highest percentage of elderly people that often have other comorbidities like diabetes and hypertension that significntly increase the probability of serious condition, next the precautions in the Italy have not been sufficient what is more likely the cause of such strong hit by COVID than 5G. To the New York, that is one of the crowdest cities in the world and social distancing have not been established soon enough.

**
*We agree with you and have added the following text to in our Discussion Section to address these points; thank you:*
**

***“We recognize that many factors have influenced the pandemic’s course. Before restrictions were imposed, travel patterns facilitated the seeding of the virus, causing early rapid global spread. Population density, higher mean population age, and socioeconomic factors certainly influenced early viral spread. Air pollution, particularly PM 2.5 microparticulates, likely increased symptoms in patients with COVID-19 lung disease (Fiasca, et al. 2020). In this paper, we postulate that RFR, by potentially weakening the host immune systems of large populations, among other bioeffects that we discussed, has possibly contributed to the early spread and severity of COVID-19.”***

***In addition, we would also like to respond to you with this paragraph, although we did not include it in our paper. Community public health response to the pandemic dramatically influenced the spread and intensity of COVID-19 once it became established within a community. Individual risk factors such as old age, hypertension, diabetes, and obesity put patient populations at greater risk for severe disease. Detailing the complex pathophysiology of each condition with COVID-19 is beyond the scope of this paper. Obesity, for example, a significant recognized risk factor, may be so in part perhaps because adipose cells contain a high level of ACE-2 receptors (Al-Benna, 2020). In addition, patients with morbid obesity may have restricted lung tidal volumes, exacerbating the clinical effect of lung disease caused by Sars-CoV-2*.**

4) Other argument is that in the second wave midlle europe (Czech, Hungary, Poland, Slovakia) have been hit very hard with COVID and 5G is still not introduced in this countries (maybe only capital cities, but certainly not smaller cities and villages, that had been hit even harder). So I think overally, mobility of people, meeting of families during holidays and inproper precautions have much larger effect on the pandemia than RFR exposure. But on the other hand I agree that RFR

could add some stress to individuals already weakened by the COVID.

***We agree with you and have added the following text to the Discussion Section:***

**
*“Once an agent becomes established in a community, its virulence increases (Hoyt et al., 2020). This premise can be applied to the COVID-19 pandemic. We surmise that “hot spots” of the disease that initially spread around the world were perhaps seeded by air travel, which in some areas was associated with 5G implementation. However, once the disease became established in those communities, it was able to spread more easily to neighboring regions where populations were less exposed to RFR. Second and third waves of the pandemic disseminated widely throughout communities with and without RFR, as would be expected.”*
**

5) Authors should also focus on the fact that lot of experimental studies did not provide any evidence of harmful effect of RFR (and not all of them are industry financed and ordered). Other problem with experimental evidence of harmful effects is the reproducibility of the observed effects and replicability of the studies, that are often performed with questionable devices, under not precisely characterized exposure conditions.

***Thank you for bringing this to our attention*.**

**
*We have now included this paragraph in the Discussion Section:*
**

**
*“Another weakness of this study is that some of the bioeffects from RFR exposure are inconsistently reported in the literature. Replicated studies are often not true replications. Small differences in method, including unreported details such as prior history of exposure by the organisms, non-uniform body exposure, and other variables can lead to inadvertent inconsistency. Moreover, not surprisingly, industry-sponsored studies tend to show less adverse bioeffects than studies conducted by independent researchers, suggesting industry bias (Huss et al., 2007). Some experimental studies that are not industry-sponsored have also shown no evidence of harmful effects of RFR exposure.”*
**

Despite this comments bit of article provide nice review of the RFR effects on the human beings supported by the number of peer-reviewed studies and also overview of COVID- 19 disease, that is very valuable and worth publishing after applying some of the comments to the manuscript.

Reviewer #12 (editor-in-chief): SO THESE SUGGESTIONS ARE MANDATORY TO ADDRESS

1) Please contextualize the narrative to centers/regions of outbreak where 5G is not prevalent, such as rural India, beyond the premise that correlation is not causation. Cite regions that had a 5G rollout but were not hit by the pandemic, and please provide explanations for such exemptions.

*
**Worldwide maps illustrating the similarity of COVID-19 and RFR distribution during the early phase of the pandemic are stunning, suggesting a relationship as shown in these two maps from WIGLE.net and the Johns Hopkins Coronavirus Research Center from December 2019 and April 7, 2020, respectively.***

***However, when considering only 5G networks, the correlation is less striking (Tsiang & Havas, 2021). This would be expected as 5G represents a relatively small subset of the global RFR emission in late 2019 and early 2020. Early on in the pandemic, there were active 5G networks in Thailand and Indonesia, where the reported number of COVID-19 cases in the early phase of the pandemic was minimal. In addition, the proliferation of 5G networks in Finland, which began in June 2019, was not associated with an increased incidence of COVID-19 infections. Assuming accurate data reporting by these countries, other factors, such as overall better general health of the population compared to other regions, and environmental cofactors may have provided protection to these populations. For example, less international air travel in these regions compared to other regions with greater incidence of the disease could be a factor that provided greater protection to such regions with 5G*.**

*** However, once an agent becomes established in a communal reservoir, its virulence increases (Hoyt, et al., 2020). This premise can be applied to the COVID-19 pandemic. We surmise that “hot spots” of the disease that initially spread around the world were perhaps seeded by air travel, but were then more easily spread in regions of increased RFR exposure, which in some areas was associated with 5G implementation. However, once the disease became well established in those communities, it was able to more easily spread to neighboring regions where populations were less exposed to RFR environmental toxicity. This may explain why the disease incidence in India was initially localized to Delhi, but then dispersed throughout the country over time. In addition, further waves of the pandemic disseminated more virulent variants widely throughout communities around the world with and without RF radiation, as would be expected*.**

2)What is the average power density (mW/cm2) of 5G RFR in Wuhan, and how does this compare to the cities that have harbored 5G but with low manifestation of COVID-19?

***Such data on 5G (or 4G) are unavailable to us, neither shown in the scientific literature, nor published by cities or other governments. Little is known about population exposure from real world radiofrequency radiation sources. It is also very difficult to accurately quantify the average power density at a given location. Moreover, the average power density varies greatly, depending upon the specific location, time, averaging interval, frequency, and modulation scheme. For a specific municipality it depends on the antenna density, what network protocols are used, as, for example, 2G, 3G, 4G, 5G, Wi-Fi (IEEE 802.11b Direct Sequence), WiMAX (Worldwide Interoperability for Microwave Access), DECT (Digital European Cordless Telecommunications), RADAR (Radio Detection and Ranging), and what the legal limits are for public exposure in the particular jurisdiction. RFR (radiofrequency radiation) from ubiquitous radiowave transmitters, including antennas, base stations, smart meters, mobile phones, routers, satellites, and other wireless devices currently in use superimposes and yields an additive average power density at a given location that typically fluctuates greatly over time. Using a radiofrequency power meter to measure ambient levels from 100 MHz – 8 GHz in downtown San Francisco, California, recently we found an average power density of 0.0002 mW/cm^2^, which is approximately 2x10E10 times greater than the natural background*.**

***The increasing radiofrequency power density has spawned a new application: harvesting this ambient wireless communication energy for practical use (Hassani et al., 2019). [Hassani, S.E. et al., 2019. Overview on 5G radio frequency energy harvesting. Advances in Science, Technology, and Engineering Systems 4(4): 328-346.] The burgeoning industry of radio frequency energy harvesting from such ambient levels to power the Internet of Things (IoT) and body-worn devices attests to today’s high level of electromagnetic pollution*.**

***5G is the most complex network protocol to date as it spans a vast spectral range from 600MHz to over 90GHz in half a dozen bands. The specific band frequencies and bandwidth allocations vary from country to country, as does the active usage of the various bands. Moreover, different network providers operate on different bands and on different frequencies within a band, depending on their spectrum purchase from the domestic regulatory body. So, the parameters of wavelength and bandwidth from location to location are different*.**

***Wuhan is unique in that it was among the first cities in the world to offer citywide 5G service starting on October 31, 2019, with purportedly 10,000 antennas reaching approximately 8 million citizens. The average distance between antennas was approximately 1,000 feet, meaning that every citizen was no farther than 500 feet from the nearest 5G antenna in the Wuhan metropolitan region. In other urban centers around the world where 5G had been partially installed by late 2019, the area of 5G coverage was typically limited to only certain neighborhoods. However, the aggressive 5G rollout during 2020 has most certainly increased 5G coverage dramatically*.**

***Regarding studies that would provide some data underlying your question, both of us (Rubik and Brown) are independently engaged in an international research project to measure the average and maximum power densities of wireless communication radiation from 100 MHz to 8 GHz, thus covering all above-mentioned networks, including the 5G low- and mid-bands, but not the high 5G bands of 24 GHz and above. Measurement of 24 GHz and above requires highly specialized research-grade equipment that costs several tens of thousands of USD, and has consequently been unaffordable for surveyance to most researchers and cities*.**

***Finally, in relation to the RFR power density and COVID mortality, the Mordachev study that we discuss in the manuscript does show a relationship, but it is not specific to 5G*.**

3) Your paper is hypothetical, so please remain in this hypothetical framework throughout the manuscript. Phrases such as “This is the first scientific paper documenting a link between RFR emitted by wireless communication devices and COVID-19” are unwarranted. Although your paper provides argumentation in favor of this hypothesis, it does not establish a link (cause-effect) between 5G and the incidence of COVID-19. Please nuance this statement and other similar statements in the text.

***We deleted this sentence, “This is the first scientific paper documenting a link between RFR emitted by wireless communication devices and COVID-19,” from the manuscript*.**

***We have also rephrased other statements less definitively throughout the document as you request, in the Abstract, Discussion Section, and Conclusion Section. Moreover, our Discussion and Conclusion Sections have been vastly rewritten to reflect a hypothetical framework for the paper*.**

4) Please unify all units of power density throughout the manuscript to conform to the standard unit used in the US (mW/cm2). The text is inconsistent with the nomenclature, where sometimes the unit is abbreviated while in other instances the unit is written out. It is advisable to consistently abbreviate to mW/cm2. That makes it easier for the readers to contextualize research results with the upheld norm for RFR exposure.

***Thank you for pointing out these inconsistencies. All standard units referring to power density have been converted to mW/cm^2^, as requested*.**

5) Please include a paragraph where you attempt to introduce gaps/flaws in your hypotheses. One of the main ingredients of such a paragraph would be to point out to readers that in many studies the power densities used to study biological effects exceeded the maximum level of 1 mW/cm2. Note all other aspects of the setup and execution of cited experimental studies that deviate from the manner in which 5G RFR is reduced to practice in Wuhan and elsewhere. Such a paragraph helps put the narrative into complete perspective.

***Proving causation of a pandemic’s severity or spread to a fluctuating environmental agent such as RFR is unrealistic, perhaps impossible. However, we knew this from the onset, and we only attempted to look for correlations in bioeffects that suggest RFR exposure may be a contributing factor in the pandemic*.**

***Laboratory studies on animals and cell cultures, designed to prove RFR bioeffects, have sometimes utilized power densities exceeding real world exposure, above 1 mW/cm^2^; for example, up to 15 mW/cm^2^ (Huang AT, Mold NG. 1980. Immunologic and hematopoietic alterations by 2,450-MHz electromagnetic radiation. Bioelectromagnetics 1:77–87. However, it must be said that the literature on non-thermal radiofrequency radiation effects from exposures at or less than 1 mW/cm^2^ is extensive. The Bioinitiative Report (https://bioinitiative.org/research-summaries) written in 2012 by 14 scientists, public health, and policy experts, and updated in 2020, summarizes hundreds of peer-reviewed scientific papers documenting evidence of non-thermal effects from exposures less than or equal to 1mW/cm^2^*.**

***In order to limit the number of variables, controlled scientific studies typically look for bioeffects from a constant source of RFR and thus do not account for the superposition of fields from multiple emitters with different modulations and variable power densities that constitute the ever-changing fields in the 5G real world, from numerous RFR emitting devices: antennas, base stations, smart meters, wireless routers, 5G satellites, mobile phones, cordless phones and their bases, computers, tablets, Bluetooth devices, and other wireless devices. Controlled scientific studies are typically short-term and often involve animals or cell cultures as targets rather than humans. It is unclear whether such results can be extrapolated to humans in the 5G real world and over the long term*.**

***We acknowledge in reviewing the literature since the initial phase of the pandemic, the declaration that a country, state or city has 5G doesn’t translate into 5G exposure to the entire population of that locality. The Wuhan city-wide 5G service that began October 31, 2019 may be a notable exception. During 2019-2020 in most instances, only small sections of each city equipped with some 5G actually had 5G antennas or base stations installed, and an unknown number of people had 5G wireless devices. Therefore, only those inhabitants that travelled into 5G regions and those who worked with 5G devices were exposed to the more intense 5G network. Even within a single household, exposure to RFR can vary dramatically depending upon a person’s relative distance to wireless routers, tablets, smart meters, mobile phones, Bluetooth devices, and other wireless products. In most communities, there are no concrete measurements available that would predict harmful effects from ambient RFR. Exposure to different frequency bands, power densities, and modulations of RFR varies from one person to another and from day to day. This variability in RFR parameters and the variability of the host’s health status on a daily basis may affect the host’s susceptibility to disease. The inability to control all of the variables in scientific studies to demonstrate a bioeffect (or its reproducibility) doesn’t translate into, ‘there is no effect here’*.**

**
*We have now included several paragraphs in the Discussion Section indicating the strengths and weaknesses (potential criticisms) of the paper, as follows:*
**

***“A major strength of this paper is that the evidence rests on a large body of scientific literature reported by many scientists worldwide and over several decades--experimental evidence of adverse bioeffects of RFR exposure at nonthermal levels on humans, animals, and cells. The Bioinitiative Report (Sage and Carpenter, 2012) and updated in 2020, summarizes hundreds of peer-reviewed scientific papers documenting evidence of nonthermal effects from exposures less than or equal to 1mW/cm^2^. Even so, some laboratory studies on the adverse health effects of RFR have sometimes utilized power densities exceeding 1mW/cm^2^. In this paper, almost all of the studies that we reviewed included experimental data at power densities less than or equal to 1mW/cm^2^*.**

***A potential criticism of this paper is that adverse bioeffects from nonthermal exposures are not yet universally accepted in science and are not considered when establishing public health policy in many nations. Decades ago, Russians and Eastern Europeans compiled considerable data on nonthermal bioeffects, and subsequently set guidelines at lower RFR exposure limits than the US and Canada, i.e., below levels where nonthermal effects are observed. However, the Federal Communications Commission (FCC, a US government entity) and ICNIRP (International Commission for Non-Ionizing Radiation Protection) guidelines operate on thermal limits based on outdated data from decades ago, allowing the public to be exposed to considerably higher RFR power densities. Regarding 5G, the telecommunication industry claims that it is safe because it complies with current RFR exposure guidelines of the FCC and ICNIRP. These guidelines were established in 1996 (Federal Communications Commission, 1996), are antiquated, and are not safety standards. Thus, there are no universally accepted safety standards for wireless communication radiation exposure. Recently international bodies such as the EMF Working Group of the European Academy of Environmental Medicine, have proposed much lower guidelines, taking into account nonthermal bioeffects from RFR exposure in multiple sources (Belyaev et al., 2016)*.**

***Another weakness of this paper is that some of the bioeffects from RFR exposure are inconsistently reported in the literature. Replicated studies are often not true replications. Small differences in method, including unreported details such as prior history of exposure by the organisms, non-uniform body exposure, and other variables can lead to inadvertent inconsistency. Moreover, not surprisingly, industry-sponsored studies tend to show less adverse bioeffects than studies conducted by independent researchers, suggesting industry bias (Huss et al., 2007). Some experimental studies that are not industry-sponsored have also shown no evidence of harmful effects of RFR exposure. It is noteworthy, however, that studies employing real-life RFR exposures from commercially available devices have shown high consistency in revealing adverse effects (Panagopoulos, 2019)*.**

***RFR bioeffects depend upon specific values of wave parameters including frequency, power density, exposure time, and modulation characteristics, as well as the cumulative history of exposure. Similar to ionizing radiation, the bioeffects of RFR exposure can be subdivided into deterministic, i.e., dose dependent effects and stochastic effects that are seemingly random. Importantly, RFR bioeffects can also involve “response windows” of specific parameters whereby extremely low level fields can have disproportionally detrimental effects (Blackman et al., 1989). This nonlinearity of RFR bioeffects can result in biphasic responses such as immune suppression from one range of parameters, and immune hyperactivation from another range of parameters, leading to variations that may appear inconsistent*.**

***In gathering reports and examining existing data for this paper, we looked for outcomes providing evidence to support a proposed connection between the bioeffects of RFR exposure and COVID-19. We did not make an attempt to weigh the evidence. The RFR exposure literature is extensive and currently contains over 30,000 research reports dating back several decades. Inconsistencies in nomenclature, reporting of details, and cataloging of keywords make it difficult to navigate the literature*.**

**
*Another shortcoming of this paper is that we do not have access to experimental data on 5G exposures. In fact, little is known about population exposure from real-world RFR, which includes exposure to RFR infrastructure and the plethora of RFR emitting devices. In relation to this, it is difficult to accurately quantify the average power density at a given location, which varies greatly, depending upon the time, specific location, time-averaging interval, frequency, and modulation scheme. For a specific municipality it depends on the antenna density, which network protocols are used, as, for example, 2G, 3G, 4G, 5G, Wi-Fi, WiMAX (Worldwide Interoperability for Microwave Access), DECT (Digital European Cordless Telecommunications), and RADAR (Radio Detection and Ranging). There is also RFR from ubiquitous radiowave transmitters, including antennas, base stations, smart meters, mobile phones, routers, satellites, and other wireless devices currently in use. All of these signals superimpose to yield the total average power density at a given location that typically fluctuates greatly over time. No experimental studies on adverse health effects or safety issues of 5G have been reported, and none are currently planned by the industry, although this is sorely needed.”*
**

2^nd^ editorial decision

28-Jul-2021

Ref.: Ms. No. JCTRes-D-21-00034R1

Evidence for a Connection between COVID-19 and Exposure to Radiofrequency Radiation from Wireless Communications Including Microwaves and Millimeter Waves

Journal of Clinical and Translational Research

Dear Dr. Rubik,

Reviewers have now commented on your paper. You will see that they are advising that you revise your manuscript. If you are prepared to undertake the work required, I would be pleased to reconsider my decision.

For your guidance, reviewers’ comments are appended below.

If you decide to revise the work, please submit a list of changes or a rebuttal against each point which is being raised when you submit the revised manuscript. Also, please ensure that the track changes function is switched on when implementing the revisions. This enables the reviewers to rapidly verify all changes made.

Your revision is due by Jul 28, 2021.

To submit a revision, go to https://www.editorialmanager.com/jctres/ and log in as an Author. You will see a menu item call Submission Needing Revision. You will find your submission record there.

Yours sincerely

Michal Heger

Editor-in-Chief

Journal of Clinical and Translational Research

Reviewers’ comments:

Reviewer #1: The authors have sufficiently addressed my comments in their response and revised manuscript. I would recommend paper for publishing after minor corrections.

It came to my attention that the authors have included references to the publications, which were not peer - reviewed. These references should be omitted to comply with generally accepted standards of scientific publications. Otherwise this paper will not be considered as meeting the accepted standards for scientific publication.

I have also noticed the evident incorrectness in the title. As far as millimeter waves is a part of microwaves, which are in turn a part of radiofrequency band, the title should be changed to “Evidence for a Connection between COVID-19 and Exposure to Radiofrequency Radiation from Wireless

Communications Including Millimeter Waves”

Finally this is one more comprehensive review of non-thermal effects of millimeter waves, which show their dependence on variety of physical and biological variables, to be cited in this paper [^1^]. These dependencies are of critical issue as they account for why some studies on millimeter waves biological effects were not replicated.

[^1^] I.Y. Belyaev, V.S. Shcheglov, E.D. Alipov, V.D. Ushalov, Nonthermal effects of extremely high-frequency microwaves on chromatin conformation in cells in vitro - Dependence on physical, physiological, and genetic factors, IEEE Transactions on Microwave Theory and Techniques, 48 (2000) 2172-2179.

Reviewer #3: The authors have responded adequately to my comments, and I recommend publication without reservations. Making changes as suggested by the Editor-in-Chief would also improve the cpontent and structure of the document.

Reviewer #8: I stand by my prior review, which is on file with you. I have read the author’s response to all the reviewers, and I think that the author has understood the suggestions of these reviewers, and has incorporated their suggested changes, where possible, to make the paper even clearer and better. The author clearly states when a suggested edit was beyond the scope of the paper in a few instances, but she included enough references for people to check out themselves, regarding additional data or studies, or made it clear that such studies have not been done, as far as she knows, but should be done by researchers in the future.

So again, I feel that this paper has made its point, that further research on 5G and the non-thermic effects on the body’s physiology, need to be studied further and more comprehensively, before 5G is implemented worldwide to blanket the earth, and then makes any future research difficult to do, since soon it will already be in widespread use, and will serve to inhibit most controlled studies, that would then require large faraday shielding cages. And since she clearly shows that this micro and milli-radiation and frequencies, may have additive effects with illnesses such as CoVid and other diseases that effect the tissues (such as lung and blood vessels) and their physiology (such as membrane changes that effect oxygen exchange and CO2 release, or effect the coagulation system, among others), that seems to demand that further scientific controlled studies be done before corporations launch a massive change in the environment that effects micro and macrobiotic organisms, plants, animals, and humans, and changes the Biosphere forever. So I highly support that this paper be published as soon as possible, so that its thesis can be studied and be debated by a much wider scientific audience, and further testing and research can be done, to further investigate these findings and projections for our survival in our collective future

Reviewer #10: - Page 2, introduction section, last paragraph. “Here we present the evidence suggesting that RFR has been a contributing factor exacerbating COVID-19.”

This is still an unconfirmed hypothesis. The sentence should be reformulated.

- Page 3. Lines 31-33. “Therefore 5G requires base stations and antennas to be much more closely spaced than previous generations”.

It is also true that 5G base stations usually have a lower power than previous generations. This point should be commented by authors.

- introduction section: “The new system therefore requires significant densification of 4G infrastructure as well as new 5G antennas that may dramatically increase the population’s

wireless communications radiation exposure both inside structures and outdoors.”

Some evidence from Switzerland ( doi.org/10.3390/app11083592 ), Sweden (doi:10.3390/app10155280 ), South Korea (doi.org/10.1002/bem.22345), UK (https://www.ofcom.org.uk/__data/assets/pdf_file/0021/214644/emf-test-summary-010321.pdf ) show that the real impact of 5G base stations in urban areas is very limited. How can authors support the hypothesis of a “dramatic increase” in the radiation exposure?

- Table 1. Authors should include in the “RFR exposure bioeffects” listed in this table the frequency and the level (i.e., power density) of exposure linked with each of the cited effects. Authors should also specify if these exposures are below or above the international limits. This information is only partially reported in the text (results section), and is of critical importance.- As authors confirmed, “both the very young and the very old populations are the most vulnerable to adverse effects from RFR exposure”. However, since RFR exposure in children is the same than in elderly, this evidence is not in line with epidemiologic data showing divergent COVID-19-related risk in children and in aged people, nor with the hypothesis formulated by authors about the role of RFR in COVID-19 pandemic. Authors should discuss this contradiction.

Reviewer #11: Authors have responded to all of my questions and notes. Now I support the manuscript for the publication.

There is additional documentation related to this decision letter. To access the file(s), please click the link below. You may also login to the system and click the ’View Attachments’ link in the Action column.

Author’s response

Reviewers’ comments:

Reviewer #1: The authors have sufficiently addressed my comments in their response and revised manuscript. I would recommend paper for publishing after minor corrections.

It came to my attention that the authors have included references to the publications, which were not peer - reviewed. These references should be omitted to comply with generally accepted standards of scientific publications. Otherwise this paper will not be considered as meeting the accepted standards for scientific publication.

***While we appreciate that you would prefer all references to be peer-reviewed, we would like to cite two important papers that have not been peer-reviewed. Many non-peer-reviewed manuscripts have been cited in professional journal papers during the COVID-19 pandemic to help experts bring forth knowledge as quickly as possible. We believe it is fully appropriate to cite research on morphological changes in red blood cells that relate to blood clotting, since SARS-CoV-2 and its spike protein have been shown to be thrombogenic (cause blood clotting in the body) and can directly bind to ACE2 receptors on platelets (Zhang et al., 2020). Even when isolated, the spike protein has been shown to cause endothelial injury (Lei et al., 2021) that can lead to clotting. In addition, we find it fully appropriate to cite a paper investigating the implementation of 5G infrastructure in relation to the initial spread of COVID-19 worldwide. These are the only 2 instances of non-peer-reviewed papers among over 130 citations and references in our review paper. We maintain that they are essential to our thesis. Moreover, we have pointed out in our manuscript that these two papers have not yet been published in peer-reviewed journals, so readers can exercise critical discernment*.**

I have also noticed the evident incorrectness in the title. As far as millimeter waves is a part of microwaves, which are in turn a part of radiofrequency band, the title should be changed to “Evidence for a Connection between COVID-19 and Exposure to Radiofrequency Radiation from Wireless Communications Including Millimeter Waves”

**
*We have had several requests to change our paper title for various reasons from reviewers. Our last title was, “Evidence for a Connection between COVID-19 and Exposure to Radiofrequency Radiation from Wireless Communications Including Microwaves and Millimeter Waves” We specified “microwaves and millimeter waves” because we believe the medical professionals that read this journal are likely to be unfamiliar with conventional physics or engineering nomenclature of the radiofrequency spectrum. We wanted to be certain that readers understood that both microwaves and millimeter waves would be discussed. Based on a different reviewer’s recommendation, we propose the following title for the paper, “Evidence for a Connection between COVID-19 and Exposure to Radiation from Wireless Communications Including 5G”*
**

Finally this is one more comprehensive review of non-thermal effects of millimeter waves, which show their dependence on variety of physical and biological variables, to be cited in this paper [^1^]. These dependencies are of critical issue as they account for why some studies on millimeter waves biological effects were not replicated. [^1^] I.Y. Belyaev, V.S. Shcheglov, E.D. Alipov, V.D. Ushalov, Nonthermal effects of extremely high-frequency microwaves on chromatin conformation in cells in vitro - Dependence on physical, physiological, and genetic factors, IEEE Transactions on Microwave Theory and Techniques, 48 (2000) 2172-2179. ***Thank you for this reference. We have augmented the discussion section of the manuscript as follows, cited this paper, and included the reference:***

***“RFR bioeffects depend upon specific values of wave parameters including frequency; power density; polarization; exposure duration; modulation characteristics; as well as cumulative history of exposure and background levels of electromagnetic, electric and magnetic fields. In laboratory studies, bioeffects observed also depend upon genetic parameters and physiological parameters such as oxygen concentration (Belyaev et al., 2000). The reproducibility of bioeffects of RFR exposure has sometimes been difficult due to failure to report and/or control all of these parameters.”*** Reviewer #3: The authors have responded adequately to my comments, and I recommend publication without reservations. Making changes as suggested by the Editor-in-Chief would also improve the cpontent and structure of the document.

***Thank you. We also made all changes as recommended by the Editor-in-Chief in our revision following the first peer-review*.** Reviewer #8: I stand by my prior review, which is on file with you. I have read the author’s response to all the reviewers, and I think that the author has understood the suggestions of these reviewers, and has incorporated their suggested changes, where possible, to make the paper even clearer and better. The author clearly states when a suggested edit was beyond the scope of the paper in a few instances, but she included enough references for people to check out themselves, regarding additional data or studies, or made it clear that such studies have not been done, as far as she knows, but should be done by researchers in the future. So again, I feel that this paper has made its point, that further research on 5G and the non-thermic effects on the body’s physiology, need to be studied further and more comprehensively, before 5G is implemented worldwide to blanket the earth, and then makes any future research difficult to do, since soon it will already be in widespread use, and will serve to inhibit most controlled studies, that would then require large faraday shielding cages. And since she clearly shows that this micro and milli-radiation and frequencies, may have additive effects with illnesses such as CoVid and other diseases that effect the tissues (such as lung and blood vessels) and their physiology (such as membrane changes that effect oxygen exchange and CO2 release, or effect the coagulation system, among others), that seems to demand that further scientific controlled studies be done before corporations launch a massive change in the environment that effects micro and macrobiotic organisms, plants, animals, and humans, and changes the Biosphere forever. So I highly support that this paper be published as soon as possible, so that its thesis can be studied and be debated by a much wider scientific audience, and further testing and research can be done, to further investigate these findings and projections for our survival in our collective future

***Thank you*.**

Reviewer #10: - Page 2, introduction section, last paragraph. “Here we present the evidence suggesting that RFR has been a contributing factor exacerbating COVID-19.” This is still an unconfirmed hypothesis. The sentence should be reformulated.

***This sentence has been removed. The new sentence inserted is, “We explore the scientific evidence suggesting a possible relationship between COVID-19 and radiofrequency radiation including 5G (fifth generation) of wireless communication technology, henceforth referred to as RFR.”*** - Page 3. Lines 31-33. “Therefore 5G requires base stations and antennas to be much more closely spaced than previous generations”.

It is also true that 5G base stations usually have a lower power than previous generations. This point should be commented by authors.

***The operating power of a base station is only one of several parameters determining the actual radiation exposure at a certain location. The directional concentration of RF energy in sector antennas, typically used in 4G base stations, and the highly collimated pencil beam generated by phased array antennas used in 5G greatly increase the so-called equivalent isotropically radiated power (EIRP) and effective radiated power (ERP). ERIP and ERP are much more relevant for estimates of human exposure and user equipment (UE) receiver performance alike*.**

***An accurate determination of a base station’s radiated RF power at a certain location is only possible through well-defined measurement protocols, such as the FCC Publication No. 412172^1^, with specialized and calibrated equipment. Because the power varies from moment to moment, a meaningful measurement must be made continuously and be capable of detecting fast pulses in the microsecond range to determine peak power density, integrate all registered power to derive an average power density and then calculate the peak-to-average power ratio (PAPR) to determine FCC compliance. Furthermore, because the received power varies from location to location due to obstacles in the propagation path and anisotropy in the antenna’s radiation pattern, these measurements must be performed at all locations of interest. Such measurements are expensive and impractical and are typically only being performed to settle legal compliance issues*.**

***For practical purposes estimates are produced that offer a rough guidance. These estimates must consider a complex array of determining parameters, such as federally regulated transmitter power limitations, which in turn are determined by the authorized frequency and the EIRP and ERP reflecting the actual gain (directionality) of the particular antenna in use. Additional complicating factors are the allotted bandwidth and the PAPR. As a rule of thumb, the power density from one transmitter at a given location is proportional to EIRP, ERP, PAPR, bandwidth, number of polarizations (horizontal, vertical, circular, etc.), data rate (increases with ever more complex modulation schemes, which require higher received power to function reliably), carrier aggregation (a technique to increase data rate, akin to allocate more bandwidth or operate several channels simultaneously) and inversely proportional to the distance from the base station. Power densities from other transmitters, be they neighboring base stations or the myriad of nearby user equipment (UE) transmitters, superimpose linearly and are additive*.**

***A comparison between 4G and 5G networks must at a minimum take into account the allocated spectrum, allotted bandwidth, permitted EIRP/ERP/PAPR, base station densification and service capacity, a measure of the density of UE transmitters (5G is planned to simultaneously serve up to 1,000-times as many UEs as 4G)*.**

**
*4G and 5G Spectrum and Bandwidth in the US*
**

***For spectrum allocation the FCC follows the guidelines of the 3^rd^ Generation Partnership Project^2^ (3GPP), an umbrella term for a number of standards organizations for developing protocols for wireless communication, including 4G and 5G*.**

***For 4G the 3GPP Technical Specification 36.101 version 17.1.0 released 01-09-2021^3^ lists in Table 5.5-1 the Long Term Evolution (LTE) Operating Bands. 4G bands use microwaves in the range of 617-2,369 MHz with an aggregated base station bandwidth of 414 MHz*.**

***5G uses millimeter-waves in addition to microwaves. The 3GPP is publishing separate specifications for each frequency range designated as Frequency Range 1 (FR 1) and Frequency Range 2 (FR 2). For the 410-7,125 MHz FR 1 used for 5G the 3GPP Technical Specification 38.101-1 version 17.1.0 released 04-13-2021^4^ lists in Table 5.2-1 the New Radio (NR) Operating Bands in FR1 with an aggregated base station bandwidth of 1,471 MHz. For the 24,250-52,600 MHz FR 2 used for 5G the 3GPP Technical Specification 38.104 version 17.1.0 released 04-08-2021^5^ lists in Table 5.2-2 the NR Operating Bands in FR2 with an aggregated base station bandwidth of 3,850 MHz. Together, 5G bands are used in the range of 617-40,000 MHz with an aggregated base station bandwidth of 5,321 MHz*.**

***The above referenced bandwidth allocations are confirmed by a statement on the FCC website under the heading “America’s 5G Future” that in the 5G high-band (24 – 47 GHz) “the FCC is releasing almost 5 gigahertz of 5G spectrum into the market—more than all other flexible use bands combined” and in the 5G mid-band “…we will make more than 600 megahertz available for 5G deployments”. The rollout of 5G has resulted in a 10-fold increase in allocated bandwidth for cellular base stations and user equipment alike*.**

***4G and 5G ERIP, ERP and PAPR Levels in the US*.**

***The FCC legally permitted ERP levels for Cellular Radiotelephone Service are listed in Article 47 of the United States Code (U.S.C.), Section ¦22.913 of the Code of Federal Regulations (CFR)^6^.The average maximum ERP is limited to 500W per channel or 400W/MHz per sector where a sector is typically 120º. With many channels and dozens of MHz of bandwidth per band the legal limit of the total radiated power from a base station can reach into tens of kW. However, the FCC website states on its Consumer Guide page entitled “Human Exposure to Radio Frequency Fields: Guidelines for Cellular Antenna Sites” that “…the majority of cellular or PCS cell sites in urban and suburban areas operate at an ERP of 100 watts per channel or less”. Therefore, the telecommunication industry could increase the ERP approximately 5-fold and still operate within legal limits*.**

***It is important to note that Section ¦22.913 regulates only the radiofrequency emission characteristics of a transmitter and does not distinguish between 4G and 5G, which are only different signal transmission protocols*.**

***The 4G and 5G ERIP, ERP and PAPR levels will increase because the touted increase in data throughput for both, 4G and 5G, is in part achieved through ever more complex modulations schemes such as quadrature phase-shift keying (QPSK) and quadrature amplitude modulation (QAM). To achieve higher data throughput QAM can be set to a larger constellation size to increase spectral efficiency. The constellation size is given by the number of constellation points, each of which represents a specific combination of amplitude and phase of the carrier wave. Currently 4G uses QPSK, 16QAM and 64QAM while 5G will additionally use 256QAM and beyond. Electronic noise is the limiting factor to the highest achievable modulation order, which is why the signal strength must be increased to provide a higher signal-to-noise ratio (SNR). However, as the density of UEs increases, so does the man-made noise (to be distinguished from electronic noise), which degrades the SNR due to interference and the ERIP, ERP and PAPR must be further increased. The push to deliver ever more data throughput and serve ever more UEs increases the ERIP, ERP and PAPR levels*.**

**
*Base station densification and service capacity*
**

***A key governing body of the global wireless industry is the International Telecommunication Union (ITU). It is a specialized agency of the United Nations that promotes the shared global use of the radio spectrum. In 2017 the ITU published the report^7^ “Minimum Requirements Related to Technical Performance for IMT-2020 Radio Interface(s)” outlining a 1,000-fold capacity increase. The report stipulates a minimum connection density requirement of 1,000,000 devices per km^2^. If all devices were evenly distributed in a plane, this calculates to an average density of one wireless device per meter^2^. 5G small cells using millimeter-waves will have to be placed at most 300m apart as the signal is heavily absorbed by the atmosphere and humidity. Such a small cell covers approximately 70,000m^2^ and must therefore be able to serve 70,000 devices simultaneously. To achieve such a vast capacity, 5G employs several highly sophisticated schemes, such as massive multiple-input/multiple-output (MIMO), beam forming and beam steering. To implement these schemes, 5G deploys active phased array antennas containing tens, hundreds, even up to 1,000 individual antenna elements, each driven by precisely-controlled transmitter circuitry in very specific power and phase relations. To serve a user, a highly collimated pencil beam is created, dynamically steered toward her device and tracked with it in real time*.**

***We maintain that discussion on this topic that delves into 5G engineering standards, which specify details of the communication protocol, is quite technical and beyond the scope of our manuscript. There are organizations that develop 5G standards containing protocol details including the IEEE (Institute of Electrical and Electronics Engineers) and ICNIRP (International Commission on Non-Ionizing Radiation Protection), and we have previously cited and referred to them in the manuscript*.**

**
*References*
**

***^1^FCC Draft Laboratory Division Publication, Guidelines for Determining the Effective Radiated Power (ERP) and Equivalent Isotropically Radiated Power (EIRP) of a RF Transmitting System. FCC Publication Number 412172, publication date: 08/07/2015*.**

***^2^The 3rd Generation Partnership Project (3GPP) unites seven telecommunications standard development organizations (ARIB, ATIS, CCSA, ETSI, TSDSI, TTA, TTC), known as “Organizational Partners” and provides their members with a stable environment to produce the Reports and Specifications that define 3GPP technologies*.**

***^3^3rd Generation Partnership Project, Evolved Universal Terrestrial Radio Access (E-UTRA); User Equipment (UE) radio transmission and reception. Technical Specification 36.101 v17.1.0, released 04-08-2021*.**

***^4^3rd Generation Partnership Project, NR; User Equipment (UE) radio transmission and reception; Part 1: Range 1 Standalone. Technical Specification 38.101-1 v17.1.0, released 04-13-2021*.**

***^5^3rd Generation Partnership Project, NR; Base Station (BS) radio transmission and reception. Technical Specification 38.104 v17.1.0, released 04-08-2021*.**

**
*^6^Electronic Code of Federal Regulation, Effective Radiated Power Levels. Title 47 U.S.C., Chapter I, Subchapter B, Part 22, Subpart H, ¦22.913. Published at https://www.ecfr.gov/*
**

***^7^International Telecommunication Union, Minimum Requirements Related to Technical Performance for IMT-2020 Radio Interface(s). Report ITU-R M.2410-0, November 2017*.**

**
*^8^Mobile & Wireless Forum website: https://www.mwfai.org/about.cfm*
**

- introduction section: “The new system therefore requires significant densification of 4G infrastructure as well as new 5G antennas that may dramatically increase the population’s wireless communications radiation exposure both inside structures and outdoors.” Some evidence from Switzerland ( doi.org/10.3390/app11083592 ), Sweden (doi:10.3390/app10155280 ), South Korea (doi.org/10.1002/bem.22345), UK (https://www.ofcom.org.uk/__data/assets/pdf_file/0021/214644/emf-test-summary-010321.pdf ) show that the real impact of 5G base stations in urban areas is very limited. How can authors support the hypothesis of a “dramatic increase” in the radiation exposure?

***The hypothesis of “dramatic increase” in radiation exposure is supported by the rationale and sources provided in the above rebuttal statement. The reviewer cited four publications all showing that the real impact of 5G base stations in urban areas is very limited. Here each publication will be addressed individually*.**

**
*Aertz, et al., In Situ Assessment of 5G NR Massive MIMO Base Station*
**

**
*Exposure in a Commercial Network in Bern, Switzerland*
**

***There are several concerns regarding the validity of this publication and its conclusions*.**

***1) The research independence appears questionable. While the authors claim no conflict of interest they acknowledged that “this work was supported by the Mobile & Wireless Forum (MWF)”. Member companies of the MWF include the largest corporations in the telecommunication industry, such as Apple, Cisco, Ericsson, Huawei, Intel, LG, Motorola, Qualcomm, Samsung, SONY and TCT Mobile. According to its website^8^, the Mobile & Wireless Forum was established in 1998 and its “regulatory activities are focused on developing and presenting the views of the mobile industry to regulatory agencies and authorities in a globally coordinated manner.” MWF appears to act as a political lobbying firm for the telecommunications industry*.**

***2) The authors studied only one 5G frequency at 3.6 GHz within the n78 band (3.3 – 3.8 GHz). Since 5G will use nearly a dozen frequency bands when fully deployed, the measurements are not representative for the near future*.**

***3) 3.6 GHz is part of the low 5G frequency range FR1 and is in the microwave range, not the millimeter-wave range. The vast 5G speed increases touted by the industry can only be realized in the approximately 10-times higher frequencies of the FR 2 frequency range between 24 – 43 GHz*.**

***4) The authors chose Switzerland for their measurements. However, Switzerland has one of the most stringent legal limits with regard to public exposure to RF radiation as these limits are based on the precautionary principle. For example, while Switzerland limits the power density to 10 µW/cm^2^, the US and most other countries around the world permit levels of 450 µW/cm^2^ or even higher. Selecting Switzerland as a test country does therefore result in measurements that are too low to be useful for global representation*.**

***5) Instead of using the industry standard 6 minutes and 30 minutes time averaging intervals, the authors opted to use only 30 second intervals for “convenience”. The authors claim they found this much reduced interval sufficiently representative but do not produce supportive data*.**

***6) In the abstract the authors state that the maximum exposure level extrapolated to 200 W antenna input power reaches 4.9 V/m, or 0.6% of the ICNIRP reference level. However, the ICNIRP level is 61 V/m and 4.9 V/m is 8%. Therefore, the authors understate the 5G contributing factor to the overall exposure by more than 10-fold*.**

***7) On page 8 the authors correctly quote the ICNIRP power density reference level to 10W/m^2^, which is equivalent to 61V/m. However, they then state that their extrapolated maximum field levels reached 0.6 V/m. They say that this value amounts to less than 0.01% of the ICNIRP level, when it is 1%. They miscalculated it by 100-fold as 0.6V/m is approximately 1% of 61V/m. The fact that this easily discernible error, of orders of magnitude, was missed by the peer review process is disconcerting*.**

**
*Colombi, et al., Analysis of the Actual Power and EMF Exposure from*
**

**
*Base Stations in a Commercial 5G Network*
**

***There are several concerns regarding the validity of this publication and its conclusions*.**

***1) The research is not independent. While the authors claim no conflict of interest they also acknowledge that “this research received no external funding”, meaning that all funding was provided by Ericsson, one of the largest telecommunication equipment manufacturers in the world. These statements are contradictory*.**

***2) The authors studied only one 5G frequency band, the n78 band spanning 3.3 – 3.8 GHz. Since 5G will use nearly a dozen frequency bands when fully deployed, the measurements provided are not representative for the near future*.**

***3) This study focused on the spatial power distribution from 5G base stations using beamforming techniques. Instead of measuring exposure levels at a certain location, which would measure the actual exposure of a 5G user, the authors employed an Ericsson Network Manager (software) to directly access information on the 5G base station operation. This setup allowed the analysis of the spatial distribution of the base station transmitting power in a three-dimensional space within the scan range of the antenna. It is not clear why direct exposure levels were not measured at representative user locations, which would yield real-world data. Beamforming requires an active user in order to have a signal sent from a 5G base station*.**

**
*4) Measurements were averaged over the entire range of an antenna panel covering essentially 180º of azimuth angle. The authors then anchor their main arguments to a simplistic equation to calculate, not measure, the average exposure level in equation (1): ERIP_act_ = G_ave_ * P_ave_*
**

***where ERIP_act_ is the calculated “actual” ERIP, G_ave_ is the time-averaged gain and P_ave_ is the total, cell-wide, time-averaged transmit power. The ERIP calculated from the product of two time-averaged parameters cannot reflect true temporal exposure patterns and makes it impossible to determine the important peak-to-average power ratio*.**

***5) In the discussion section the authors offer a puzzling and speculative logic to argue against a substantial ERIP increase with the expected increase in 5G users and their vastly increased data demand. While the authors acknowledge that “P_ave_ is directly related to the amount of downlink traffic”, they argue that G_ave_ will drop as “…more users will result in an even larger spread of the energy over the antenna scan range, which will contribute to a reduction of G_ave_…”, largely compensating for the increased P_ave_ in equation (1) and “therefore, ERIP_act_ is likely not to increase substantially when increasing the number of users.” This reasoning is stunningly inaccurate, as an increase in both the number of users and data demand will both contribute to a substantial increase in ERIP and thus increased levels of public exposure*.**

**
*Selmaoui, et al., Exposure of South Korean Population to 5G Mobile Phone Networks (3.4-3.8 GHz)*
**

***This publication has several weaknesses that make an accurate estimation of the true exposure of the South Korean public to the newly rolled out 5G network difficult or impossible*.**

***1) Measurements published in this paper were taken in November 2019, only 6 months after the 5G network was brought online for the public. The authors could not determine the degree of network usage and had to qualify their conclusions by stating that “it is likely that the 5G network was not being used to its maximum and the number of subscribers was relatively low”. But since the number of users and their transmitted data rate are key metrics to determine user exposure levels, the publication does not permit a quantitative estimation of the ERIP contribution from 5G base stations*.**

***2) The authors studied only one 5G frequency band, the n78 band spanning 3.4 – 3.8 GHz. Since 5G will use nearly a dozen frequency bands when fully deployed, the measurements are not representative for the near future*.**

***3) The author’s main conclusion is that the 5G network contributes only 15% to the total telecommunications emission. However, they acknowledge that measurements taken in the vicinity of a base station yielded 12 V/m and 21 V/m for the antenna’s baseline and maximum power, respectively. But this amounts to 20% and 34% of the maximum field strength permitted by ICNIRP. And given that 2G, 3G and 4G networks typically operate well below the maximum permitted ICNIRP levels, these measured levels appear comparable and might even be significantly higher than the existing telecommunication emissions combined. Without more accurate measurements the reader may conclude that the 5G network has the potential to double the public’s exposure levels, contrary to the seemingly low contribution of 15% quoted by the authors*.**

***Ofcom, Electromagnetic Field (EMF) measurements near 5G mobile phone base stations, Summary of results. Technical Report – Version 3, March 2021*.**

***1) The impartiality of the Office of Communication (Ofcom) is questionable. Ofcom is the government-approved regulatory and competition authority for the broadcasting and telecommunications industries of the UK. Recently Ofcom has been marred by the Martin Bashir scandal involving the British Broadcasting Corporation (BBC). During the scandal it was revealed that more than half of its board members had links to the BBC, a corporation Ofcom is supposed to regulate. While not specifically linked to the telecommunications industry, Ofcom’s pro-industry bias must be considered*.**

***2) For several of the measurements the authors had to choose areas of less than typical user density because of COVID-19 restrictions. These measurements skewed the results toward lower than normal exposure levels*.**

***3) The authors studied only one 5G frequency band, the n78 band spanning 3.41 – 3.68 GHz. Since 5G will use nearly a dozen frequency bands when fully deployed, the measurements are not representative for the near future*.**

***4) The biggest shortcoming of this study is the failure of the authors to create realistic scenarios of users actually using the 5G network. 5G base stations mainly emit their signals on demand when UEs request service. Since the authors did not create a UE service demand at the measurement location, their probe largely registered radiation not emitted from the 5G base stations. The very small signals attributed to 5G emission likely reflects only radiation scattered from reflecting objects (buildings, trees, etc.) of side lobes of a formed beam serving a UE at a location separate from the monitored location. Because of this shortcoming, the quoted test results for 5G base stations are meaningless*.**

***5) The authors did not indicate the distance of their sensor from a base station. Since the ERIP changes greatly as a function of distance, the data cannot properly be judged as being a realistic and representative mix of near and far distances, the way typical users would be represented*.**

- Table 1. Authors should include in the “RFR exposure bioeffects” listed in this table the frequency and the level (i.e., power density) of exposure linked with each of the cited effects. Authors should also specify if these exposures are below or above the international limits. This information is only partially reported in the text (results section), and is of critical importance.

***We created Table 1 to be a visual summary for the reader rather than a comprehensive compilation of data with references. In addition, following our first peer review, we changed the subheadings on the bioeffects in the text to match the Table 1 subheadings. The reader is thus guided to particular sections of the manuscript text to obtain details regarding RFR exposure parameters and literature citations. Since the manuscript has already been changed to be hypothetical, we maintain that this is sufficient*.**

- As authors confirmed, “both the very young and the very old populations are the most vulnerable to adverse effects from RFR exposure”. However, since RFR exposure in children is the same than in elderly, this evidence is not in line with epidemiologic data showing divergent COVID-19-related risk in children and in aged people, nor with the hypothesis formulated by authors about the role of RFR in COVID-19 pandemic. Authors should discuss this contradiction.

***Children are less vulnerable than adults to the SARS-CoV-2 virus because they have fewer ACE2 receptors, as we explained in our first rebuttal to another reviewer. Elderly adults have the most ACE2 receptors, i.e., more “targets” for the virus to enter their cells, and are thus more vulnerable to the virus. Both the very young and the very old populations are the most vulnerable to adverse effects from RFR exposure. Even so, the question of age-related exposure to wireless communication radiation in relation to the pandemic goes beyond the scope of our manuscript*.**

Reviewer #11: Authors have responded to all of my questions and notes. Now I support the manuscript for the publication.

***Thank you*.**

{There is additional documentation related to this decision letter. To access the file(s), please click the link below. You may also login to the system and click the ’View Attachments’ link in the Action column.}

Review of the paper titled “**Evidence for a Connection between COVID-19 and Exposure to Radiofrequency Radiation from Wireless Telecommunications Including Microwaves and Millimeter Wave**s”

**General Comment**

The paper is significantly improved. Certain points need to be further refined

**Specific Comments**

1. The title is not accurate. Mm waves are the highest frequency part of microwaves, and microwaves are the highest frequency part of the wider RF band. Thus they are not different to be named separately. Moreover wireless communications do not emit only RF but ELF as well and the effects are mainly due to the ELFs. The title must be changed to: “Evidence for a Connection between COVID-19 and Exposure to Radiation from Wireless Communications” (at the end of the title the authors may add “including 5G”). For the same reasons, I suggest RFR should be replaced in most places throughout the paper by wireless communications radiation.

**
*As you suggest, we changed the title to, “Evidence for a Connection between COVID-19 and Exposure to Radiation from Wireless Communications including 5G.”*
**

***We defined RFR (radiofrequency radiation) explicitly in our manuscript to mean “wireless communications radiation,” and not the entire radiofrequency spectrum. Furthermore, numerous research reports studying the bioeffects of exposure to this part of the electromagnetic spectrum—the wireless communications spectrum--also refer to it as RFR, so this use of nomenclature and abbreviation “RFR” is consistent with other scientific literature on exposure bioeffects*.**

2. The technical description of 5G should include the frequencies of the ELF pulsations which play by far the most important role in the bioeffects. It would be important if the authors were able to collect information regarding this. If they are unable to find such information it must be reported, (that in spite of their search they found no information regarding the ELF pulsations, although this is an important part of this type of radiation).

***ELF pulsations are not part of the 5G protocols per se; rather they may be considered a ‘side effect’. Although ELF frequencies are not deliberately designed into the 5G (or 4G) communication protocol and its modulation scheme, polling communications devices use ELF pulsations. For example, the ubiquitous Wi-Fi and DECT (cordless phones) devices each use polling as the basis of their communication protocol. In electronic communication, ’polling’ is the periodic checking of a device by other devices to see what state they are in, usually to see whether they are still connected or want to communicate*.**

***One of us (Rubik) measured the radiation pattern of a Wi-Fi router, shown in Figure 1, and a DECT base station, shown in Figure 2. The left-hand oscillograms were recorded during idle states and the right-hand ones during user/caller activity. Wi-Fi polls devices within its range at a rate of 10Hz, whereas DECT base stations poll their satellite phones at a rate of 50Hz*.**

**Figure 1 F1:** Wi-Fi radiation pulse pattern. No user activity during “Idle”.

**Figure 2 F2:** DECT radiation pulse pattern. No phone call during “Idle”.

***4G and 5G per se do not use polling. Rather, they are asynchronous, which is more akin to event-driven mode. However, even asynchronous complex signals will generally have a slew of frequency components. Any ELF manifesting from the complex 5G signals, if present, will result from the superposition of waves from numerous phenomena: 5G’s digital modulation, fast-sweeping steered beams, and the rapid multiplexing of a base station’s radiation to quasi-simultaneously serve multiple users. Therefore, only careful measurement of 5G installations and active user devices would reveal any resulting ELF signals*.**

***In relation to this question, there is scientific data indicating that pulsed RFR is much more detrimental to biological organisms than continuous-wave (CW) radiation, which the reviewer clearly understands*.**

***The study of non-thermal effects from pulsed versus CW RFR is difficult because of a number of confounding effects from engineering variables, such as frequency, power density, modulation type, pulse rate, pulse rate variation, exposure time, and background or stray electric, magnetic, and electromagnetic levels. Research is further complicated due to individual organisms’ genetics, physiology, prior history of exposure, and resilience to RFR. Below we discuss several key papers on this topic*.**

**
*Pakhomov and Murphy Paper: Review of Russian Research*
**

**
*Historically the former Soviet Union has conducted much more in-depth research in this field than Western industrialized countries. Pakhomov and Murphy, two researchers with military backgrounds, published a seminal paper comprehensively reviewing some 1,200 research reports^1^. While a great number of specific effects were reported, the brief summary discerns the following trends:*
**

***the studies emphasized RF-induced changes in the nervous system function*.*****Many studies convincingly demonstrated significant bioeffects of pulsed microwaves*.*****Modulation was often the determining factor in substantial differences between pulsed and CW radiation at comparable time-averaged intensities*.*****Many bioeffects from low-intensity pulsed microwaves reported clearly pathogenic effects*.*****The specific mechanisms of interaction are not well understood*.**

**
*Belyaev Paper: Researched CW and Pulsed RFR, Summarized Details on Pulsed RFR*
**

**
*An in-depth study of non-thermal CW and pulsed RFR was presented by I. Belyaev^2^. This paper provides an overview of the complex effects of such radiation on various physical and biological parameters. Besides the well-known dependencies on carrier frequency and modulation, the compiled data suggest also dependencies on polarization, intermittence and coherence time of exposure, static magnetic fields, electromagnetic stray fields, genotype, gender, physiological and individual traits, and cell density during exposure. Belyaev offered more detailed findings in his summary, as follows:*
**

***biological effect dependence on frequency within specific frequency windows of “resonance-type”******narrowing of the frequency windows with decreasing intensity******dependence on modulation and polarization******sigmoid dependence on intensity within specific intensity windows including super-low power density comparable to intensities from base stations******thresholds in intensity and exposure time (coherence time)******dependence on duration of exposure and post-exposure time dependence on cell density that suggests cell-to-cell interaction during exposure******dependence on physiological conditions during exposure, such as stage of cell growth, concentration of oxygen and divalent ions, and activity of radicals******dependence on genotype, cell type, and cell line******gender, age, and individual differences******the presence of electromagnetic stray fields during exposure***

**
*Semin et al. Paper: Power Level Window Effects and Resonance Effects*
**

***Semin^5^ studied the effect of weak RFR on the stability of DNA’s secondary structure in vitro. Samples were exposed to microwave radiation consisting of 25 ms pulses, 1-6 Hz repetition rate and 0.4-0.7 mW/cm^2^ peak power. The experiments established that irradiation at 3 or 4 Hz and 0.6 mW/cm^2^ peak power clearly increased the accumulated damage to the DNA’s secondary structure (P<0.00001). However, changing the pulse repetition rate to 1, 5, or 6 Hz, as well as changing the peak power to 0.4 or 0.7 mW/cm^2^, eliminated the effect entirely. Thus, the effect occurred only within narrow “windows” of the peak intensities and modulation frequencies*.**

***Franzen Air Force Report^3^: Physics of Pulsed Microwaves Impinging on Organic Tissue, Brillouin Precursors*.**

***This author investigated the propagation of a 1 GHz wave train of 10 ns duration by Fourier integral transform after impinging on a dielectric media, such as bio-tissue. The study confirmed the creation of so-called Brillouin precursors, which are secondary bursts of energy generated when a microwave pulse of fast rise time enters tissue. Importantly, the Brillouin precursor microwave energy experiences much less absorption than the primary microwave radiation, which is absorbed exponentially. As a result, microwave radiation with fast pulses, such as in digital cellular radiation, travels much deeper into the body than predicted by conventional models. This effect becomes more pronounced with higher frequencies and with faster pulse rise times, both of which are typically proportional to the transmitted data rate. Therefore, the argument that 5G millimeter wave radiation is absorbed within two millimeters of the skin is false*.**

***Albanese, et al.^4^, showed similar Brillouin precursor generation from periodically pulsed microwave radiation*.**

**
*Extremely Low Frequency (ELF) Components of Digital Cellular Microwave Radiation*
**

***Biological tissue is mostly comprised of water. As such, the tissue presents a lossy dielectric medium for impinging microwave radiation. The complicated waveform of a digitally modulated mobile phone signal can mathematically be viewed as the linear superposition of a myriad of sine waves with varying frequencies and amplitudes, called Fourier analysis. When a wave comprised of different frequency components propagates through a lossy dielectric medium, it experiences dispersion, and the wave decomposes into its Fourier components. (In optics this is the familiar separation of white light into its constituent rainbow of colors.) As a result, numerous frequencies are impacting biological tissue when a modulated microwave is absorbed*.**

***Thus, a complex 5G signal interacting with biological tissue may yield Fourier frequency components in the ELF region. This is the net result of the carrier’s digital modulation in combination with fast-sweeping steered beams and the rapid multiplexing of a base station’s RF energy to quasi-simultaneously serve multiple users*.**

**
*Summary and Conclusion*
**

***The following was added to the manuscript in the section, Overview on Bioeffects of Radiofrequency Radiation (RFR) Exposure. References 1-4 shown here were also cited and added to the Reference Section*.**

*“**Pulsed RFR exhibit substantially different**
**bioeffects, both qualitatively and quantitatively (generally more pronounced)**
**compared to continuous waves at similar time-averaged power densities. (Pakhomov and Murphy, 2011; Belyaev, 2010; Franzen, 1999; Albanese et al., 1989) The specific interaction mechanisms are not well understood.”***

*References*

***^1^A. Pakhomov and M. Murphy, A Comprehensive Review of the Research on Biological Effects of Pulsed Radiofrequency Radiation in Russia and the Former Soviet Union. DOI: 10.1007/987-1-4615-4203-2_7, July 2011*.**

**
*^2^I. Belyaev, Dependence of non-thermal biological effects of microwaves on physical and biological variables: implications for reproducibility and safety standards. European Journal of Oncology – Library Non-Thermal Effects and Mechanisms of Interaction Between Electromangnetic Fieldds and Living Matter, Vol. 5: 187-218 (2010). An ICEMS Monograph. L. Giuliani and M. Soffritti. Bologna, Italy, Ramazzini Institute. Available at:*
**

**
*http://www.icems.eu/papers.htm?f=/c/a/2009/12/15/MNHJ1B49KH.DTL*
**

***^3^J. Franzen, Wideband Pulse Propagation in Linear Dispersive Bio-Dielectrics Using Fourier Transforms. United States Air Force Research Laboratory, Report No. AFRL-HE-BR-TR-1999-0149, February 1999*.**

**
*^4^R. Albanese, J.Penn and R. Medina, “Short-rise-time Microwave Pulse Propagation Through Dispersive Biological Media,” J. Opt. Soc. Am. A, vol.6, no.9, pp 1441-1446 (1989)*
**

***^5^Semin, Iu. A., Shvartsburg, L. K., and Dubovik, B. V., Changes in the secondary structure of DNA under the influence of an external low-intensity electromagnetic field, Radiats Biol Radioecol, 35/1: 36-41 (1995)*.**

2. Describe in a few words the “areas of airspace opacification … in patients with COVID-19 pneumonia”

***The airspace opacities in COVID-19 pneumonia have been characterized as being multi-focal and appearing similar to ground glass, i.e., they are abnormally aerated areas of lung that are not completely filled with fluid or inflammatory exudates and so one can partially see through them*.**

3. For power densities significantly smaller than 1mW/cm^2^ (as in descriptions of Magras and Xenos 1997 and others, change the unit to μW/cm^2^. For example, change 0.0005 – 0.001 mW/cm^2^ to 0.5-1 μW/cm^2^. The antenna park in the Magras and Xenos study was not for mobile telephony antennas. Such antennas are not placed in antenna parks but everywhere. Please check and correct. Also check e^-13^ mW/cm^2^ (Belyaev et al., 1996)

***The Editor-in-Chief requested that we use mW/cm^2^ as the unit of power density throughout the manuscript in the first revision that we already made*.**

***We removed “mobile phone” in connection with the antenna park in our discussion on the Magras and Xenos study*.**

***We changed the value to 10^-13^, which we checked, and this is correct (Belyaev et al., 1996)*.**

5. Page 8, left, lines 45-49. In the description of Walleczek (1992) and anywhere else, correct ELF from (<300 Hz) to (<3000 Hz). Even though most ELFs of anthropogenic EMFs are indeed <300 Hz, the correct upper limit of this band is 3000 Hz.

***We changed it to <3000 Hz*.**

6. “The irregular gating of electrosensitive cell membrane ion channels by coherent, pulsed, oscillating electromagnetic fields was first presented by Panagopoulos, et al., in 2002”. Please describe more accurately: “The mechanism of irregular gating of voltage-gated ion channels in cell membranes by polarized and coherent, oscillating electric or magnetic fields was first presented by Panagopoulos, et al., in 2000 and 2002”, and add also the citation [Panagopoulos DJ, Messini N, Karabarbounis A, Filippetis AL, and Margaritis LH, (2000): A Mechanism for Action of Oscillating Electric Fields on Cells, *Biochemical and Biophysical Research Communications*, 272(3), 634-640.]

**
*We changed this sentence accordingly and added the citation and reference that you provided: “The mechanism of irregular gating of voltage-gated ion channels in cell membranes by polarized and coherent, oscillating electric or magnetic fields was first presented in 2000 and 2002 (Panagopoulos et al., 2000; 2002).”*
**

7. It is not accurate to write: “Pall combined these two observations to propose that low frequency RFR may be causing increased intracellular Ca2+ via the activation of voltage-gated calcium channels (Pall, 2013)”.

Pall did not “propose” something that was proposed long ago by others. Moreover you cannot say he “combined the observations” since he did not refer to Panagopoulos et al in his paper. Therefore it is accurate to write: “Pall (2013) in his review of EMF-induced bioeffect studies combined with use of calcium channel blockers, observed/noted that calcium channels play a major role in EMF bioeffects”.

**
*We revised these sentences as follows: “Pall (2013) in his review of RFR-induced bioeffects combined with use of calcium channel blockers noted that voltage-gated calcium channels play a major role in RFR bioeffects. Increased intracellular Ca^+2^ results from the activation of voltage-gated calcium channels, and this may be one of the primary mechanisms of action of RFR on organisms.”*
**

8. “Research has shown that the interaction between a virus and voltage-gated calcium channels promote virus entry at the virus-host cell fusion step. Thus….”. Provide reference.

**
*We added this citation (Chen et al., 2019) to the manuscript where this sentence appeared. The reference was already in our Reference Section:*
**

**
*Chen, X., R. Cao R and W., Zhong W. 2019. Host Calcium Channels and Pumps in Viral Infections. Cells, 9 (1): 94. DOI:10.3390/cells9010094*
**

9. The “Potekhina *et al*. (1992) whole study cannot be found. Without reading the whole paper one cannot conclude whether the described effects were due to the GHz frequency or to ELF pulsations. Does the document state clearly that no modulation or pulsation of the GHz field was present? It is unlikely that any microwave EMF does not contain such components, even in the form of on/off for energy saving reasons. Referring to microwave exposure studies without knowing this information might be very misleading.

***The Potekhina et al. (1992) paper is available online only in Russian. Pakhomov et al., 1998, who wrote a review of bioeffects and reviewed the Russian paper, indicated no modulations were used, and we had already cited and provided a reference to this review paper, too. Recently the Potekhina et al. paper was reviewed by Pall (2021), who also reviewed a number of other studies published in Russian. It was clear from both Pakhomov’s and Pall’s reviews that the Potekhina et al. study involved unmodulated millimeter waves. Moreover, we were able to translate the Russian paper of Potekhina et al. using Google Translate, and we determined that this statement is accurate. We added Pall’s 2021 citation and reference to the manuscript*.**

**
*We modified the sentence summarizing the Potekhina et al. study, adding the word “unmodulated”: “Since then many other researchers have concluded that RFR exposure can affect the cardiovascular system. Potekhina et al. (1992) found that certain unmodulated frequencies (55 GHz; 73 GHz) caused pronounced arrhythmia.”*
**

**
*We also added the following sentence at the end of the section on cardiac effects:*
**

**
*“Pall’s recent review suggests that millimeter waves may act directly on the pacemaker cells of the sinoatrial node of the heart to change the beat frequency, which may underlie arrhythmias and other heart problems (Pall, 2021).”*
**

**
*Pall M.L. 2021. Millimeter wave and microwave frequency radiation produce deeply penetrating effects: the biology and the physics. Rev Environmental Health, May 26, 2021: 1-12. https://doi.org/10.1515/reveh-2020-0165*
**

The absorption of microwaves by water molecules is no different than the absorption of infrared (heat). The heating effect of microwaves is their only established effect. If a non-thermal mechanism of microwaves exists, it is to be discovered. Now some papers report “non-thermal effects of microwaves” without addressing the issue whether the effects were indeed due to the “microwaves” or to the inevitably co-existing ELFs. This may be very misleading.

***We believe that the statement, “The absorption of microwaves by water molecules is no different than the absorption of heat,” is too simplistic. It may apply to pure water, but not to living organisms where the structure and dynamics of intracellular water is intimately connected with the structure and dynamics of biomolecules. Although the heating effect of microwaves is considered by some as their only established bioeffect, numerous researchers have found nonthermal bioeffects of exposure at lower power densities. This is a main thesis of our manuscript, and we have cited and discussed numerous research reports on nonthermal bioeffects. Consider that organisms are approximately 70% water. Moreover, water absorbs broadly in the GHz spectral region and also displays GHz resonant frequencies. Irradiation at water resonant frequencies, of which there are several in the GHz spectral region, including 2.45 GHz, a Wi-Fi router frequency (pulse-modulated at 10 Hz), may elicit bioeffects due to changes in hydration of biomolecules. The dynamics of water in the hydration layer around proteins and other biomolecules play a crucial role in biomolecular structure and function. In fact, dielectric spectroscopy at GHz frequencies is used to investigate the dynamics and structure of hydration water of biomolecules. Consider also that a paper reported that low-intensity radiation of 70.6 and 73 GHz affects E. coli bacterial growth and changes the properties of water, which we discussed in our previously revised manuscript draft [Torgomyan H, Kalantaryan V, Trchounian A. Low intensity electromagnetic irradiation with 70.6 and 73 GHz frequencies affects Escherichia coli growth and changes water properties. 2011. Cell biochemistry and biophysics. 60(3):275-81]. https://doi.org/10.1007/s12013-010-9150-8 ] It is hypothesized that intracellular water thus affected by absorption of GHz radiation affects the hydration of protein molecules in organisms that may alter protein structure and rates of biochemical reactions (Betskii and Lebedeva, 2004)*.**

***Thus, continuous wave GHz radiation at low power densities, by making subtle changes in intracellular water structure and protein hydration, could subsequently alter biochemistry and physiology and lead to a variety of bioeffects*.**

***In addition, let us clarify that we stated the RFR signal modulations for the data discussed in our manuscript*.**

A careful look in the Pakhomov et al 1998 review paper shows that at least in some of the reviewed studies ELFs were present. Yet, this is not reflected in the title or the abstract of the paper. For example, in page 3 they write: “The Fourier spectrum of these oscillations included two strong peaks, at 5.25 and 46.8 Hz, and these peaks did not change during at least 2 h of experimentation”. Yet, the study is described as investigating mm wave effects… In most of the studies reviewed in this paper, information on possible existence of ELFs is lacking. Thus, their presence is not excluded. The reported in abstract “10 mW/cm^2^ and less” power density for the reviewed studies is huge, and it is unlikely that thermal effects (which start from ~0.1 mW/cm^2^) were absent. In the Betskii and Lebedeva 2004 review, such information is lacking throughout the whole paper. It is reported that many of the studies were performed by use of a “wideband oscillator with an electric tuning of oscillation frequency developed and brought into lot production in the U.S.S.R.”. It is unlikely that this oscillator did not include on/off pulsations, even only for energy saving reasons. When we refer to such studies, knowledge on microwave electronics provided by specialized physicists/engineers is necessary in order to prevent misleading conclusions. Since this information is lacking we cannot conclude that the reported non-thermal effects were due to the microwave frequencies. I suggest such issues should be carefully investigated before statements are made which can possibly be very misleading.

Thus, revise accordingly.

**
*Once again, for the reports that we reviewed in this manuscript, we stated the signal modulations, if any, which were used and reported. We are aware that especially in the early studies, decades ago, the signal parameters were not always fully reported. We also wrote the following revised statement mentioning the role of modulation, among other parameters, in our Discussion Section:*
**

**
*“RFR bioeffects depend upon specific values of wave parameters including frequency; power density; polarization; exposure duration;*
**

**
*modulation characteristics; as well as the cumulative history of exposure and background levels of electromagnetic, electric and magnetic fields. In laboratory studies, bioeffects observed also depend upon genetic parameters and physiological parameters such as oxygen concentration (Belyaev et al., 2000). The reproducibility of bioeffects of RFR exposure has sometimes been difficult due to failure to report and/or control all of these parameters.”*
**

***However, as we wrote in other responses in this Second Rebuttal, there is evidence for nonthermal bioeffects of continuous microwaves, unmodulated, for example, on water and the hydration of biomolecules that can subsequently affect their structure and function, and lead to microwave bioeffects, as well as evidence for cardiac arrhythmias due to exposure to continuous millimeter waves*.**

10. “Saili *et al.*, (2015) found that exposure to Wi-Fi (2.45 GHz)….”. Please report that Wi-Fi radiation includes 10 Hz pulsations (see Table 3, Belyaev et al, 2016).

**
*Yes; 10 Hz is the polling frequency of Wi-Fi routers. Earlier in this Second Rebuttal we defined polling frequency. We changed this sentence accordingly: “Saili et al., (2015) found that exposure to Wi-Fi (2.45 GHz pulsed at 10 Hz) affects heart rhythm, blood pressure, and the efficacy of catecholamines on the cardiovascular system, indicating that RFR can act directly and/or indirectly on the cardiovascular system.”*
**

11. Page 11, DECT is Digitally Enhanced Cordless Telecommunications. Please check and correct.

***Thank you; we changed this sentence accordingly*.**

3^rd^ editorial decision

03-Aug-2021

Ref.: Ms. No. JCTRes-D-21-00034R2

Evidence for a Connection between COVID-19 and Exposure to Radiofrequency Radiation from Wireless Communications Including 5G

Journal of Clinical and Translational Research

Dear Dr. Rubik,

Reviewers have now commented on your paper. You will see that they are advising that you revise your manuscript. If you are prepared to undertake the work required, I would be pleased to reconsider my decision.

For your guidance, reviewers’ comments are appended below.

If you decide to revise the work, please submit a list of changes or a rebuttal against each point which is being raised when you submit the revised manuscript. Also, please ensure that the track changes function is switched on when implementing the revisions. This enables the reviewers to rapidly verify all changes made.

Your revision is due by Sep 02, 2021.

To submit a revision, go to https://www.editorialmanager.com/jctres/ and log in as an Author. You will see a menu item call Submission Needing Revision. You will find your submission record there.

Yours sincerely

Michal Heger

Editor-in-Chief

Journal of Clinical and Translational Research

Reviewers’ comments:

Reviewer #1: The authors have sufficiently addressed my comments and I would recommend the paper for publishing.

In my opinion, comprehensive reviewing the issue of 5G exposure in response of authors to the Reviewer 10 deserves to be published in a separate paper.

Reviewer #5: Review of the revised paper titled “Evidence for a Connection between COVID-19 and Exposure to Radiofrequency Radiation from Wireless Communications Including 5G”

Comments on 2nd revision

Although the authors are clearly no experts on the physical parameters of these radiation types, for reasons unknown and despite my repeating comments, they insist on calling this complex radiation “RFR”. This is misinformation. The whole technique of multiplexing (different tasks from different users accommodated simultaneously by 2G/3G/4G/5G antennas) is based on ELF pulsations. Moreover there is no microwave generator that is not turned on and off for energy saving reasons and this represents additional pulsations. Thus, it is not only the periodic checking of a device by base antennas (’polling’), but several more types of ELF pulsations, plus ELF/VLF modulation, plus random variability of the signal mainly in the ULF (0-3 Hz) band. The authors completely ignored the wi-fi pulsations before my comments. Now they provided recordings from wi-fi and DECT which (of course) show nothing but pulsations. The carrier RF wave is within those pulses. It is exactly the same with any form of modern digital wireless communications. The DECT pulsations are not known to be 50 Hz but 100 Hz and 200 Hz (see Pedersen 1997). Since the authors claim that there are no “per se” pulsations in 4G, 5G, why don’t they take similar recordings from corresponding cell phones to see whether there are pulsations or not, plus intense ULF/ELF/VLF variability of the signals. These types of radiation do not exist without pulsations/ELF components (see Pirard and Vatovez for recordings of such signals). A continuous wave RF signal alone has no applications, in other words, it is useless. It is the modulation/pulsations that make it perform its task to convey information. The information is ALWAYS within the ELF modulation which is ALWAYS combined with pulsations in digital telecom signals. Therefore the experiments in the Russian studies that the authors repeatedly refer to, obviously, for unknown reasons, they have not reported the ELF components which existed, otherwise they tested signals which have no applications. In other words these studies are of dubious quality which we cannot know for sure since the originals are in Russian and most of the available information is from reviews of other Russian scientists. The authors should not refer to studies that are not fully available in English, like Potekhina et al. (1992) and others. There is a huge amount of work in the western world having tested the bioactivity of the RF and ELF parameters of complex RF signals which is much more reliable than the mistranslated Russian studies. Most (if not all) of these studies have shown that the ELF pulsations/modulation are by far the bioactive factor and not the (non-modulated) carrier RF signals alone (Blackman et al 1980; Frei et al 1988; Huber et al 2002). Moreover there have been series of replication studies on Russian reports which have reported inability to reproduce the reported effects of non-modulated microwave/mm wave signals (one example is Furia et al 1986).

The authors must replace “Radiofrequency Radiation” (RFR) by “Radiation from Wireless Communications” throughout their paper, otherwise their paper is misleading. On the one hand they speak of a possible connection between the pandemic and the extreme levels of the telecom signals which may be important for public health protection, and on the other hand they give a false impression that the adverse effects are due to the RF carrier, pulling away the attention from the true bioactive components which are the ELFs.

Most studies reviewed by Pall 2013 are ELF studies. Still the authors describe his review as referring to “RFR”. It seems that the authors do not see anything else but RFR.

Now the authors referred to Pall (2021). This is a deeply flawed paper as described in Panagopoulos (2021). I suggest they exclude it completely otherwise they must also refer to the criticism on this paper. Referring to a paper which is officially criticized by a peer-reviewed letter to the editor, without referring to the criticism is inappropriate.

In conclusion, I insist that the authors revise their paper addressing every point I reported in my previous comments, plus the above points. Otherwise their paper is misleading and I cannot suggest acceptance. Of course it is for the editor to decide.

References Blackman CF., Benane SG, Elder JA, House DE, Lampe JA, and Faulk JM, (1980): “Induction of calcium - ion efflux from brain tissue by radiofrequency radiation: Effect of sample number and modulation frequency on the power - density window”. Bioelectromagnetics, 1, 35 - 43.

Frei M., Jauchem J, Heinmets F, (1988): Physiological Effects of 2.8 GHz Radio-Frequency Radiation: A Comparison of Pulsed and Continuous-Wave Radiation, Journal of Microwave Power and Electromagnetic Energy, 23, 2.

Furia L, Hill DW, and Gandhi OP, (1986): Effect of Millimeter Wave Irradiation on Growth of Saccaromyces Cerevisiae”, IEEE Transactions on Biomedical Engineering, 33, 993-999.

Huber R, Treyer V, Borbely AA, Schuderer J, Gottselig JM, Landolt HP, Werth E, Berthold T, Kuster N, Buck A, Achermann P. (2002): Electromagnetic fields, such as those from mobile phones, alter regional cerebral blood flow and sleep and waking EEG. J Sleep Res. 11(4):289-95.

Panagopoulos DJ, (2021): Comments on Pall’s “Millimeter (MM) wave and microwave frequency radiation produce deeply penetrating effects: the biology and the physics”, Rev Environ Health, doi: 10.1515/REVEH-2021-0090. Online ahead of print.

Pedersen GF, (1997): Amplitude modulated RF fields stemming from a GSM/DCS-1800 phone, Wireless Networks 3, 489-498

Pirard W, and Vatovez B, Study of Pulsed Character of Radiation Emitted by Wireless Telecommunication Systems, Institut scientifique de service public, Liège, Belgium. (available in: https://www.issep.be/wp-content/uploads/7IWSBEEMF_B-Vatovez_W-Pirard.pdf)

Reviewer #10: No further comment.

There is additional documentation related to this decision letter. To access the file(s), please click the link below. You may also login to the system and click the ’View Attachments’ link in the Action column.

Author’s response

Reviewers’ Comments

Review of the revised paper titled “**Evidence for a Connection between COVID-19 and Exposure to Radiofrequency Radiation from Wireless Communications Including 5G**”

**Comments on 2^nd^ revision**

Although the authors are clearly no experts on the physical parameters of these radiation types, for reasons unknown and despite my repeating comments, they insist on calling this complex radiation “RFR”. This is misinformation. The whole technique of multiplexing (different tasks from different users accommodated simultaneously by 2G/3G/4G/5G antennas) is based on ELF pulsations. Moreover there is no microwave generator that is not turned on and off for energy saving reasons and this represents additional pulsations. Thus, it is not only the periodic checking of a device by base antennas (’polling’), but several more types of ELF pulsations, plus ELF/VLF modulation, plus random variability of the signal mainly in the ULF (0-3 Hz) band. The authors completely ignored the wi-fi pulsations before my comments. Now they provided recordings from wi-fi and DECT which (of course) show nothing but pulsations. The carrier RF wave is within those pulses. It is exactly the same with any form of modern digital wireless communications. The DECT pulsations are not known to be 50 Hz but 100 Hz and 200 Hz (see Pedersen 1997).

***We greatly appreciate the reviewer’s constructive criticism, and we will address all of the remaining issues in this Third Rebuttal*.**

***The oscillogram that we showed in Figure 2 of the Second Rebuttal depicted the actual measurement of a DECT 6.0 base station showing 50 Hz pulses. In response to the reviewer’s critique of the pulsation rate that we found, we repeated the measurement using a different DECT phone base, because the original DECT phone base that we had tested was no longer available to us. This time we used a Uniden DECT 6.0 base, model no. DECT 1588, manufactured in 2008. The pulsation rate was indeed 100 Hz, as the reviewer stated is the DECT pulsation rate. We are puzzled as to why we originally measured 50 Hz with the previous DECT phone base*.**

Since the authors claim that there are no “per se” pulsations in 4G, 5G, why don’t they take similar recordings from corresponding cell phones to see whether there are pulsations or not, plus intense ULF/ELF/VLF variability of the signals. These types of radiation do not exist without pulsations/ELF components (see Pirard and Vatovez for recordings of such signals). A continuous wave RF signal alone has no applications, in other words, it is useless. It is the modulation/pulsations that make it perform its task to convey information. The information is ALWAYS within the ELF modulation which is ALWAYS combined with pulsations in digital telecom signals.

***The reviewer wrote, “The authors claim that there are no ‘per se’ pulsations in 4G, 5G.” Please let us clarify our viewpoint and understanding. Previously the reviewer requested a “technical description of 5G (that) should include the frequencies of the ELF pulsations which play by far the most important role in the bioeffects”*.**
***However, after investigating the standards documentation on 5G, we found no mention of these ELF in the 5G protocol. In other words, ELF pulsations or other modulations are apparently not part of the 5G protocol itself, or else this may be proprietary and not in the public domain. Nonetheless, we fully agree with the reviewer that the information in wireless communications radiation is carried by pulsing or other modulations. The various multiplexing schemes employed in telecommunication signalling give rise to ELF components, even though the carrier frequency may be in the GHz range*.**

***We added the following material to the manuscript that starts at the bottom of page 3 - 4*.**

**
*“…organisms lack the ability to adapt to heightened levels of unnatural radiation of wireless communications technology with digital modulations that include short intense pulses (bursts).”*
**

***The following was added to the manuscript on page 4, along with 4 new references added to the manuscript for these citations (Lin-Liu and Adey, 1982; Penafiel et al., 1997; Huber et al. 2002; Panagopoulos, 2021)*.**

**
*“All types of wireless communications employ ELFs in the modulation of the radiofrequency carrier signals, typically pulses to increase the capacity of information transmitted. This combination of radiofrequency radiation with ELF modulation(s), is generally more bioactive, as it is surmised that organisms cannot readily adapt to such rapidly changing wave forms (Lin-Liu & Adey, 1982; Penafiel et al., 1997; Huber et al., 2002; Panagopoulos et al., 2002). Therefore, the presence of ELF components of radiofrequency waves from pulsing or other modulations must be considered in studies on the bioeffects of wireless communications radiation. Unfortunately, the reporting of such modulations has been unreliable, especially in older studies (Panagopoulos, 2021).”*
**

Therefore the experiments in the Russian studies that the authors repeatedly refer to, obviously, for unknown reasons, they have not reported the ELF components which existed, otherwise they tested signals which have no applications. In other words these studies are of dubious quality which we cannot know for sure since the originals are in Russian and most of the available information is from reviews of other Russian scientists. The authors should not refer to studies that are not fully available in English, like Potekhina *et al*. (1992) and others.

**
*We removed this reference, Potekhina et al. (1992), and this sentence from the manuscript that was previously on page 10:*
**

**
*“Potekhina et al. (1992) found that certain unmodulated frequencies (55 GHz; 73 GHz) caused pronounced arrhythmia.”*
**

There is a huge amount of work in the western world having tested the bioactivity of the RF and ELF parameters of complex RF signals which is much more reliable than the mistranslated Russian studies. Most (if not all) of these studies have shown that the ELF pulsations/modulation are by far the bioactive factor and not the (non-modulated) carrier RF signals alone (Blackman et al 1980; Frei et al 1988; Huber et al 2002). Moreover there have been series of replication studies on Russian reports which have reported inability to reproduce the reported effects of non-modulated microwave/mm wave signals (one example is Furia et al 1986).

***We understand that studies on the bioeffects of invariant signals (continuous radiofrequency waves) are not equivalent to studies on bioeffects of modulated wireless communication radiation. We agree with you about the failure in some early Russian reports to reveal these important modulation parameters*.**

**
*On page 4, we included this sentence in the paragraph where we discuss ELF modulations:*
**

**
*“Unfortunately, the reporting of such (ELF) modulations has been unreliable, especially in older studies (Panagopoulos, 2021).”*
**

**
*On page 13 in the Discussion Section, we added this paragraph, because it points to the complexity of real world wireless communications signals, even from a single wireless device, and the Panagopoulos, 2016 reference was also added:*
**

**
*“Finally, there is an inherent complexity to the WCR (Wireless communications radiation) that makes it very difficult to fully characterize wireless signals in the real world that may be associated with adverse bioeffects. Real world digital communication signals, even from single wireless devices, have highly variable signals: variable power density, frequency, modulation, phase, and other parameters changing constantly and unpredictably each moment, as associated with the short, rapid pulsations used in digital wireless communication (Panagopoulos et al., 2016). For example, in using a mobile phone during a typical phone conversation, the intensity of emitted radiation varies significantly each moment depending on signal reception, number of subscribers sharing the frequency band, location within the wireless infrastructure, presence of objects and metallic surfaces, “speaking” versus “non-speaking” mode, among others. Such variations may reach 100% of the average signal intensity. The carrier radiofrequency constantly changes between different values within the available frequency band. The greater the amount of information (text, speech, internet, video, etc.), the more complex the communication signals become. Therefore, we cannot estimate accurately the values of these signal parameters including ELF components or predict their variability over time. Thus, studies on the bioeffects of wireless communications radiation in the laboratory can only be representative of real world exposures (Panagopoulos et al., 2016).”*
**

***The following discussion goes beyond the scope of our manuscript, but we wish to mention it here, since we enjoyed reading the references provided by the reviewer and thinking about possible mechanisms of action*.**

***Since modulations including short intense pulsations are involved in wireless communications radiation, one hypothesis for a mechanism is that organisms act as dispersive and lossy media for this radiation. As a result, the impinging complex waveform may decompose into its various frequency components and thus partially demodulate the signals, such that ELF may emerge and act as a key bioactive component (Pirard and Vatovez, 2012, page 27) (NB: this reference was provided by the reviewer). An alternative hypothesis, also proposed by Pirard and Vatovez, is that there is a “cumulated influence of all the (frequency) components because the human body could possibly not be able to discriminate within the spectrum of the signal*,**
***so that the effects of the envelope shape (e.g. pulsed variations of the amplitude) would need to be defined (Pirard and Vatovez, 2012, page 27). Further research is needed to test comparatively these hypotheses*.**

***If organisms can indeed demodulate wireless communication radiation signals such that bioeffects may be due to component frequencies, the rapid digitally pulsed waves of wireless communications radiation would yield numerous Fourier frequency components. The faster the rise and fall times of a “pulse,” the greater the number of Fourier frequency components. These would include a large number of ELF component waves that may be bioactive*.**

***This discussion on the possible mechanisms of action of the bioeffects goes beyond the scope of our manuscript, so we did not add any material on this topic. Nonetheless, the interested reader will find it here in the Peer Review section of the publication*.**

The authors must replace “Radiofrequency Radiation” (RFR) by “Radiation from Wireless Communications” throughout their paper, otherwise their paper is misleading. On the one hand they speak of a possible connection between the pandemic and the extreme levels of the telecom signals which may be important for public health protection, and on the other hand they give a false impression that the adverse effects are due to the RF carrier, pulling away the attention from the true bioactive components which are the ELFs.

**
*We are aware that each scientific and engineering discipline has a different language and jargon to describe physical phenomena. Engineers, among others, might find it misleading to call it “RFR” (radiofrequency radiation). Our paper aimed for JTCR is interdisciplinary and largely for a medical audience. As we explained in our Second Rebuttal, the health and medical science literature has typically utilized “RFR” or simply “RF” as an “umbrella” term for wireless communications radiation. Nonetheless, we changed our terminology throughout the manuscript, where appropriate, removing the term “RFR” and substituting “WCR” (wireless communications radiation). Here is the revised paragraph on page 2 where we first introduce the new terminology, and softened the language a bit, adding the words, “possible,” “may be” and “potentially,” which renders the thesis of our manuscript more hypothetical: “We explore the scientific evidence suggesting a possible relationship between COVID-19 and radiofrequency radiation related to wireless communications technology including 5G (fifth generation of wireless communication technology), henceforth referred to as WCR (wireless communications radiation). WCR has already been recognized as a form of environmental pollution and physiological stressor (Balmori, 2009). Assessing the potentially detrimental health effects of WCR may be crucial to develop an effective, rational public health policy that may help expedite eradication of the COVID-19 pandemic. In addition, because we are on the verge of worldwide 5G deployment, it is critical to consider the possible damaging health effects of WCR before the public is potentially harmed.”*
**

Most studies reviewed by Pall 2013 are ELF studies. Still the authors describe his review as referring to “RFR”. It seems that the authors do not see anything else but RFR.

***We acknowledge that most studies reviewed by Pall (2013) involve ELF. However, fourteen of the 116 references in Pall’s 2013 review involve radiofrequency, microwave, and millimeter wave bioeffects*.**

**
*We modified the manuscript on page 10 accordingly: “CCBs also block the increase of intracellular Ca^2+^ caused by WCR (wireless communications radiation) exposure as well as exposure to other electromagnetic fields (Pall, 2013).”*
**

Now the authors referred to Pall (2021). This is a deeply flawed paper as described in Panagopoulos (2021). I suggest they exclude it completely otherwise they must also refer to the criticism on this paper. Referring to a paper which is officially criticized by a peer-reviewed letter to the editor, without referring to the criticism is inappropriate.

***Thank you for this information. We were previously unaware of the Panagopoulos (2021) critique. We have removed all citations to Pall (2021) and also removed the reference from our manuscript*.**

In conclusion, I insist that the authors revise their paper addressing every point I reported in my previous comments, plus the above points. Otherwise their paper is misleading and I cannot suggest acceptance. Of course it is for the editor to decide.

**
*We also added more detailed information to this paragraph in the manuscript about 5G (number of phased array antennas, 64 - 256 and performance up to 10x that of 4G) that appears on page 2, as follows:*
**

**
*“The 5G standard specifies all key aspects of the technology, including frequency spectrum allocation, beam-forming, beam steering, multiplexing MIMO (multiple in, multiple out) schemes, as well as modulation schemes, among others. 5G will utilize from 64 to 256 antennas at short distances to serve virtually simultaneously a large number of devices within a cell. The latest finalized 5G standard, Release 16, is codified in the 3GPP published Technical Report TR 21.916 and may be downloaded from the 3GPP server at https://www.3gpp.org/specifications. Engineers claim that 5G will offer performance up to 10 times that of current 4G networks (Lin, 2020).”*
**

***We have addressed every point of the reviewer in three rebuttals. Should we inadvertently have missed a point, please inform us, and we shall correct it accordingly. We appreciate the reviewer’s critique, especially the reference to relevant papers and acknowledge the significant effort it must have taken*.**

**
*References:*
**

***Huber R, Treyer V, Borbely AA, Schuderer J, Gottselig JM, Landolt HP, Werth E, Berthold T, Kuster N, Buck A, Achermann P. (2002): Electromagnetic fields, such as those from mobile phones, alter regional cerebral blood flow and sleep and waking EEG. J Sleep Res. 11(4):289-95*.**

***Lin, J.C. 2020. 5G Communications Technology and Coronavirus Disease. IEEE Microwave 21(9): 16-19*.**

***Lin-Liu, S., and Adey, W.R. (1982). Low frequency amplitude modulated microwave fields change calcium efflux rates from synaptosomes. Bioelectromagnetics, 3(3), 309–322*.**

**
*Panagopoulos DJ, (2021): Comments on Pall’s “Millimeter (MM) wave and microwave frequency radiation produce deeply penetrating effects: the biology and the physics”, Rev Environ Health. doi: 10.1515/REVEH-2021-0090. Epub ahead of print. PMID: 34246201*
**

**
*Panagopoulos, D.J., Cammaerts, M.C., Favre, D., & Balmori, A. (2016) Comments on environmental impact of radiofrequency fields from mobile phone base stations, Critical Reviews in Environmental Science and Technology, 46:9, 885-903. DOI: 10.1080/10643389.2016.1182107*
**

***Panagopoulos, D.J., Karabarbounis, A., and Margaritis, L.H. (2002). Mechanism for action of electromagnetic fields on cells. Biochemical and Biophysical Research Communications, 298 (1), 95–102*.**

***Penafiel, L.M., Litovitz, T., Krause, D., Desta, A., and Mullins, M.J. (1997). Role of modulation on the effect of microwaves on ornithine decarboxylase activity in L929 cells. Bioelectromagnetics, 18(2), 132–141*.**

**
*Pirard, W., and Vatovez, B. 2012. Study of Pulsed Character of Radiation Emitted by Wireless Telecommunication Systems, Institut scientifique de service public, Liège, Belgium. In: Proceedings of the 7^th^ International Workshop on Biological Effects of Electromagnetic Fields 8-12 October, 2012: 1-27. https://www.issep.be/wp-content/uploads/7IWSBEEMF_B-Vatovez_W-Pirard.pdf*
**

**References**

Blackman CF., Benane SG, Elder JA, House DE, Lampe JA, and Faulk JM, (1980): “Induction of calcium - ion efflux from brain tissue by radiofrequency radiation: Effect of sample number and modulation frequency on the power - density window”. *Bioelectromagnetics*, 1, 35 - 43.

Frei.

Furia L, Hill DW, and Gandhi OP, (1986): Effect of Millimeter Wave Irradiation on Growth of Saccaromyces Cerevisiae”, *IEEE Transactions on Biomedical Engineering*, 33, 993-999.

Huber R, Treyer V, Borbely AA, Schuderer J, Gottselig JM, Landolt HP, Werth E, Berthold T, Kuster N, Buck A, Achermann P. (2002): Electromagnetic fields, such as those from mobile phones, alter regional cerebral blood flow and sleep and waking EEG. *J Sleep Res*. 11(4):289-95.

Panagopoulos DJ, (2021): Comments on Pall’s “Millimeter (MM) wave and microwave frequency radiation produce deeply penetrating effects: the biology and the physics”, *Rev Environ Health*, doi: 10.1515/REVEH-2021-0090. Online ahead of print.

Pedersen GF, (1997): Amplitude modulated RF fields stemming from a GSM/DCS-1800 phone, *Wireless Networks* 3, 489–498

Pirard W, and Vatovez B, Study of Pulsed Character of Radiation Emitted by Wireless Telecommunication Systems, *Institut scientifique de service public*, Liège, Belgium. (available in: https://www.issep.be/wp-content/uploads/7IWSBEEMF_B-Vatovez_W-Pirard.pdf)

4^th^ editorial decision

25-Aug-2021

Ref.: Ms. No. JCTRes-D-21-00034R3

Evidence for a Connection between COVID-19 and Exposure to Radiofrequency Radiation from Wireless Communications Including 5G

Journal of Clinical and Translational Research

Dear authors,

I am pleased to inform you that your manuscript has been accepted for publication in the Journal of Clinical and Translational Research.

You will receive the proofs of your article shortly, which we kindly ask you to thoroughly review for any errors.

Thank you for submitting your work to JCTR.

Kindest regards,

Michal Heger

Editor-in-Chief

Journal of Clinical and Translational Research

Comments from the editors and reviewers: